# Sublinear Spectral Clustering Oracle with Little Memory

**Ranran Shen**[1]**, Xiaoyi Zhu**[2]**, Pan Peng**[1*]**, Zengfeng Huang**[23*]
[1]School of Computer Science and Technology, University of Science and Technology of China, Hefei, China
[2]School of Data Science, Fudan University, Shanghai, China
[3]Shanghai Innovation Institute, Shanghai, China
`ranranshen@mail.ustc.edu.cn,zhuxy22@m.fudan.edu.cn,`
`ppeng@ustc.edu.cn, huangzf@fudan.edu.cn`

## Abstract

We study the problem of designing *sublinear spectral clustering oracles* for well-clusterable graphs. Such an oracle is an algorithm that, given query access to the adjacency list of a graph $G$, first constructs a compact data structure $\mathcal{D}$ that captures the clustering structure of $G$. Once built, $\mathcal{D}$ enables sublinear time responses to WhichCluster$(G, x)$ queries for any vertex $x$. A major limitation of existing oracles is that constructing $\mathcal{D}$ requires $\Omega(\sqrt{n})$ memory, which becomes a bottleneck for massive graphs and memory-limited settings. In this paper, we break this barrier and establish a memory-time trade-off for sublinear spectral clustering oracles. Specifically, for well-clusterable graphs, we present oracles that construct $\mathcal{D}$ using much smaller than $O(\sqrt{n})$ memory (e.g., $O(n^{0.01})$) while still answering membership queries in sublinear time. We also characterize the trade-off frontier between memory usage $S$ and query time $T$, showing, for example, that $S \cdot T = \widetilde{O}(n)$ for clusterable graphs with a logarithmic conductance gap, and we show that this trade-off is nearly optimal (up to logarithmic factors) for a natural class of approaches. Finally, to complement our theory, we validate the performance of our oracles through experiments on synthetic networks.

## 1 Introduction

A central task in graph analysis is to uncover communities, which are groups of vertices that are more densely connected internally than externally. This problem, known as *graph clustering*, has long been a cornerstone of graph theory and algorithms (Hagen & Kahng, 1992; Chan et al., 1993; Ng et al., 2001; Czumaj et al., 2015; Peng, 2020). Beyond its theoretical significance, graph clustering underlies diverse applications, ranging from community detection in networks (Van Gennip et al., 2013; Bedi & Sharma, 2016; Li et al., 2024) to bioinformatics (Paccanaro et al., 2006) and image segmentation (Shi & Malik, 2000; Felzenszwalb & Huttenlocher, 2004).

Despite their importance, most graph clustering algorithms are impractical for large graphs, as they require reading the entire input, spending $\Omega(n)$ time, and/or building data structures of size $\Omega(n)$, where $n$ is the number of vertices. Even when only a few cluster memberships are needed, these methods still carry out full global computations, making them unsuitable for massive graphs where both time and memory (or space) matter – but memory is the primary bottleneck.

From a systems perspective, this memory bottleneck is especially pressing. Many realistic environments severely restrict available working memory: streaming models limit algorithms to a single pass with sublinear space; cloud-based platforms often impose high storage and data-transfer costs, making it infeasible to materialize the entire graph; and GPUs and TPUs offer massive compute but only modest on-chip memory relative to dataset size. In all these settings, the primary challenge is to fit a compact representation of the clustering structure into limited fast memory. Thus, developing memory-efficient clustering algorithms is not only a theoretical pursuit but also a practical necessity for analyzing trillion-edge graphs in modern computing environments.

---

*Corresponding authors.

These considerations have motivated the study of *local* clustering oracles that run in sublinear time and space. Our focus is on *sublinear spectral clustering oracles* (Peng, 2020; Gluch et al., 2021; Shen & Peng, 2023), which construct a compact data structure $\mathcal{D}$ from query access to the adjacency list of the graph. Once built, $\mathcal{D}$ enables efficient evaluation of WHICHCLUSTER$(G, x)$ queries, that is, determining the cluster assignment of any vertex $x$ without incurring the global $\Omega(n)$ costs. Importantly, these oracles return consistent assignments (with a fixed random seed) and closely approximate the ground-truth clustering, thereby making local access to clustering information both theoretically sound and practically useful.

Several recent works (Peng, 2020; Gluch et al., 2021; Shen & Peng, 2023) demonstrate that such oracles are possible under planted clustering assumptions, supporting cluster membership queries in both sublinear time and sublinear space. However, all existing sublinear spectral clustering oracles require at least $\Omega(\sqrt{n})$ space. In particular, Peng (Peng, 2020) constructs an oracle using $\tilde{\Theta}(\sqrt{n})$ space, while both Gluch et al. (Gluch et al., 2021) and Shen et al. (Shen & Peng, 2023) require $\Omega(n^{1-\delta})$ space for any $\delta \leq \frac{1}{2}$, which is again at least $\sqrt{n}$. We refer to Table 1 and Section 1.3 for more details. For truly massive graphs, this requirement is prohibitive, as limited working memory and frequent main-memory access quickly dominate the overall cost. This raises the central question:

*Is it possible to design a spectral clustering oracle that breaks the $\Omega(\sqrt{n})$ space barrier – can we use substantially less memory while still achieving sublinear query time? If so, what kinds of trade-offs between space and query efficiency can be realized?*

To the best of our knowledge, the question of establishing a space-time trade-off for sublinear spectral clustering oracle has not been explicitly studied in the prior literature. This challenge is reminiscent of recent work on space-time trade-offs in learning, beginning with Raz (2017)'s result on parity learning and later extended to tasks such as linear regression (Sharan et al., 2019) and noisy parity (Garg et al., 2021). In the area of distribution testing, a series of works (Diakonikolas et al., 2019; Berg et al., 2022; Roy & Vasudev, 2023; Canonne & Yang, 2024) have established sharp space-time trade-offs for fundamental problems such as uniformity testing and closeness testing. Much like in these learning problems and in recent advances on distribution testing, the central question for sublinear spectral clustering is how far memory usage can be reduced without making query times impractically large.

In this paper, we give the first sublinear spectral clustering oracles with little memory (i.e., much less than $O(\sqrt{n})$) and a trade-off between memory usage $S$ and query time $T$ satisfying $S \cdot T \approx \widetilde{O}(n)$ (for a class of well clusterable graphs). We show that this trade-off is nearly optimal (up to logarithmic factors) for a natural class of approaches. In the following, we first present some basic definitions.

**Basic definitions** We measure cluster connectivity using conductance, a widely studied metric (e.g., (Chiplunkar et al., 2018; Dey et al., 2019; Manghiuc & Sun, 2021; Shen & Peng, 2023)). Let $G = (V, E)$ be an undirected graph. For any vertex $v \in V$, let $d_v$ denote the degree of $v$ in $G$. For any subset $C \subseteq V$, let $\text{vol}(C) = \sum_{v \in C} d_v$ denote the volume of $C$. For any two subsets $S, C \subseteq V$, let $E(S, C)$ denote the set of edges between $S$ and $C$.

**Definition 1.1** (Outer and inner conductance). For any non-empty subset $C \subseteq V$, the *outer conductance* and *inner conductance* of $C$ is defined to be

$$\phi_{\text{out}}(C, V) = |E(C, V \backslash C)|/\text{vol}(C), \quad \phi_{\text{in}}(C) = \min_{S \subseteq C, 0 < \text{vol}(S) \leq \text{vol}(C)/2} \phi_{\text{out}}(S, C).$$

Specially, the *conductance* of graph $G$ is defined to be $\phi(G) = \min_{C \subseteq V, 0 < \text{vol}(C) \leq \text{vol}(G)/2} \phi_{\text{out}}(C, V)$.

Intuitively, inner (resp. outer) conductance captures the internal (resp. external) connectivity of a cluster. A "good" cluster exhibits both large inner conductance and small outer conductance. Based on the definition of conductance, we give the formal definition of the input graph which is assumed to have a planted clustering structure (see Definition 1.3).

**Definition 1.2** ($k$-partition). Let $G = (V, E)$ be a graph. A $k$-partition of $V$ is a collection of $k$ disjoint subsets $C_1, \ldots, C_k$ such that $\bigcup_{i=1}^{k} C_i = V$.

**Definition 1.3** (($k, \varphi, \varepsilon$)-clusterable graph). Let $k \geq 2$ be an integer and let $\varphi \in (0, 1)$ and $\varepsilon \in [0, 1)$. Let $G = (V, E)$ be a graph. If there exists a $k$-partition of $V$, denoted by $C_1, \ldots, C_k$, such that for all $i \in [k]$, $\phi_{\text{in}}(C_i) \geq \varphi$, $\phi_{\text{out}}(C_i, V) \leq \varepsilon$ and for all $i, j \in [k]$, one has $\frac{|C_i|}{|C_j|} \in O(1)$, then we call $G$ is a ($k, \varphi, \varepsilon$)-*clusterable graph*.

We work in the *adjacency list model*, where the algorithm can query any neighbor of a specified vertex in constant time.

## 1.1 MAIN RESULTS

**Sublinear spectral clustering oracle**    A key contribution of this work is a spectral clustering oracle that operates with very little memory and provides an explicit trade-off between memory and query time. Given a $(k, \varphi, \varepsilon)$-clusterable graph, the goal of a clustering oracle is to build a data structure $\mathcal{D}$ in sublinear time such that, for any vertex $x$, the oracle can answer WHICHCLUSTER$(G, x)$ in sublinear time. Moreover, the clustering induced by answering WHICHCLUSTER$(G, x)$ for all $x$ should have a small misclassification error, that is, only a small fraction of vertices are assigned to the wrong clusters compared to the ground truth.

In what follows, we state our main theorem in the simplified setting where $\varphi = \Omega(1)$ and $d, k = O(1)$. While we state our results for $d$-regular graphs, they naturally extend to **$d$-bounded** graphs, i.e., graphs in which every vertex has degree at most $d$ (see Appendix D).

**Theorem 1.1** (Informal main result; full statement in Theorem 3.1). *Suppose $\varphi = \Omega(1)$, $d, k = O(1)$, and $\varepsilon \leq h(d, k, \varphi)$ for some function $h$. Let $G = (V, E)$ be a $d$-regular $(k, \varphi, \varepsilon)$-clusterable graph with clusters $C_1, \ldots, C_k$. Let $1 \leq M \leq O\left(n^{1/2 - O(\varepsilon)}\right)$ be a trade-off parameter. Then there exists a sublinear spectral clustering oracle that:*

- *constructs a data structure $\mathcal{D}$ using $\widetilde{O}\left(n^{O(\varepsilon)} \cdot M\right)$ bits of space,*
- *answers any WHICHCLUSTER query in $\widetilde{O}\left(n^{1 + O(\varepsilon)} / M\right)$ time,*
- *misclassifies at most $O(\varepsilon^{1/3})|C_i|$ vertices in each cluster $C_i$, $i \in [k]$.*

Note that the space $S$ used to build $\mathcal{D}$ and the query time $T$ satisfy the trade-off $S \cdot T = \widetilde{O}\left(n^{1 + O(\varepsilon)}\right)$. The oracle is built upon a new subroutine ESTCOLLIPROB (Alg. 2) for estimating the collision probability of two random walk distributions with asymptotically space-time trade-off. In particular, when $\varepsilon \ll 1/\log n$, this simplifies to $S \cdot T = \widetilde{O}(n)$. The theorem establishes a trade-off: larger space $S$ yields faster queries, while smaller $S$ slows them down. Unlike prior oracles that require at least $\Omega(\sqrt{n})$ space, our method operates with substantially less space, often far below $\sqrt{n}$, thereby breaking the $\sqrt{n}$ space barrier.

We provide a more detailed comparison between our main algorithmic result (Theorem 3.1) and prior work in Table 1. Note that there are two types of results, one with $O(\log k \cdot \varepsilon)|C_i|$ misclassification error at the cost of larger space usage and query time (e.g., Gluch et al. (2021)) and the other with $O(\text{poly}(k) \cdot \varepsilon^{1/3})|C_i|$ misclassification error and slightly smaller space usage and query time (e.g., Shen & Peng (2023)).

Table 1: Comparison of our results (Theorem 3.1) with previous work in terms of space usage, query time and misclassification error. We use $O_\varphi$ to suppress dependence on $\varphi$ and $\widetilde{O}$ to hide all poly$(\log n)$ factors. Here $\delta \in (0, \frac{1}{2}]$ is a constant and $1 \leq M \leq O(\frac{n^{1/2 - O(\varepsilon/\varphi^2)}}{k})$ is a trade-off parameter.

| work | space usage | query time | misclassification error |
|---|---|---|---|
| Peng (2020) | $\widetilde{O}_\varphi(\sqrt{n} \cdot \text{poly}(\frac{k}{\varepsilon}))$ | $\widetilde{O}_\varphi(\sqrt{n} \cdot \text{poly}(\frac{k}{\varepsilon}))$ | $O(kn\sqrt{\varepsilon})$ |
| Gluch et al. (2021) | $\widetilde{O}_\varphi(n^{1 - \delta + O(\varepsilon)} \cdot \text{poly}(\frac{k}{\varepsilon}))$ | $\widetilde{O}_\varphi(n^{\delta + O(\varepsilon)} \cdot \text{poly}(\frac{k}{\varepsilon}))$ | $O(\log k \cdot \varepsilon)|C_i|$ [†] |
| **our (Item 1)** | $\widetilde{O}_\varphi(n^{O(\varepsilon)} \cdot M \cdot \text{poly}(\frac{k}{\varepsilon}))$ | $\widetilde{O}_\varphi(n^{1 + O(\varepsilon)} \cdot \frac{1}{M} \cdot \text{poly}(\frac{k}{\varepsilon}))$ | $O(\log k \cdot \varepsilon)|C_i|$ [†] |
| Shen & Peng (2023) | $\widetilde{O}_\varphi(n^{1 - \delta + O(\varepsilon)} \cdot \text{poly}(k))$ | $\widetilde{O}_\varphi(n^{\delta + O(\varepsilon)} \cdot \text{poly}(k))$ | $O(\text{poly}(k) \cdot \varepsilon^{1/3})|C_i|$ [†] |
| **our (Item 2)** | $\widetilde{O}_\varphi(n^{O(\varepsilon)} \cdot M \cdot \text{poly}(k))$ | $\widetilde{O}_\varphi(n^{1 + O(\varepsilon)} \cdot \frac{1}{M} \cdot \text{poly}(k))$ | $O(\text{poly}(k) \cdot \varepsilon^{1/3})|C_i|$ [†] |

   †    for each cluster $C_i, i \in [k]$.

Previous oracles require at least $\Omega(\sqrt{n})$ space usage while our oracle operates within much less space. Moreover, we stress that, our new clustering algorithms (Item 1 and Item 2), although they introduce a space constraint, affect only the space usage and query time; all other guarantees (e.g., the conductance gap and the misclassification error) remain unchanged.

**Distinguishing 1-cluster vs. 2-cluster**    As a corollary of our main result, we obtain a sublinear algorithm for distinguishing between a single-cluster expander and a graph consisting of two disjoint

clusters. Formally, let $\varphi = \Omega(1)$ and $d = O(1)$. Consider the following promise problem: the input is a $d$-regular graph $G = (V, E)$ that is guaranteed to be in one of two cases: (i) $G$ is a $\varphi$-expander on $n$ vertices (i.e., $(1, \varphi, 0)$-clusterable); or (ii) $G$ is the disjoint union of two identical $\varphi$-expanders, each on $n/2$ vertices (i.e., $(2, \varphi, 0)$-clusterable). The goal of the 1-cluster vs. 2-cluster problem is to determine which case holds.

We address this problem with an ESTCOLLIPROB-based algorithm, yielding the following result.

**Theorem 1.2** (Upper bound). *For any trade-off parameter $1 \leq M \leq O(\sqrt{n})$, there exists an algorithm (Alg. 5) that, with probability at least $1 - 2n^{-100}$, solves the 1-cluster vs. 2-cluster problem. Moreover, the algorithm:*

- *uses $\widetilde{O}(M)$ bits of space,*
- *runs in $\widetilde{O}\left(\frac{n}{M}\right)$ time.*

We complement this with a lower bound for distinguishing between the two cases when the graph can only be accessed through random walk queries.

**Definition 1.4** (Random walk queries). For any specified starting vertex $x$, a random walk query returns the endpoint of an $O(\log n)$-step random walk starting from $x$.

**Theorem 1.3** (Lower bound). *Any algorithm that correctly solves the 1-cluster vs. 2-cluster problem with error at most $1/3$ using only random walk oracles must satisfy $S \cdot T \geq \Omega(n)$, where $S$ and $T$ denote the space complexity and time complexity of the algorithm, respectively.*

Note that a random walk query can be simulated with $O(\log n)$ adjacency-list queries, so our upper bound matches the lower bound up to $\mathrm{poly}(\log n)$ factors. Since the ESTCOLLIPROB-based approach solves the 1-cluster vs. 2-cluster problem, our lower bound indicates that its trade-off is nearly tight. This, in turn, suggests that the space-time trade-off of our clustering oracle is essentially tight, at least for approaches based on collision probability estimation.

## 1.2 TECHNICAL OVERVIEW

**Sublinear spectral clustering oracle** To obtain sublinear spectral clustering oracles that rely on a $\log(k)$ or $\mathrm{poly}(k)$ conductance gap, a key primitive is the estimation of the dot product $\langle \boldsymbol{f}_x, \boldsymbol{f}_y \rangle$, where $\boldsymbol{f}_x$ is the spectral embedding of $x \in V$ (see Definition 2.1). Suppose there exists an algorithm that estimates such dot products using $S$ space and $T$ time. We can then design a clustering oracle based on this primitive, which uses $\widetilde{O}(\mathrm{poly}(k) \cdot S)$ space to construct a data structure $\mathcal{D}$ and answers WHICHCLUSTER queries in $\widetilde{O}(\mathrm{poly}(k) \cdot T)$ time (see Section 3.2). Thus, the central task is to understand the space-time trade-off for dot product estimation, as it directly determines the efficiency of the resulting clustering oracle.

Indeed, the previous $\Omega(\sqrt{n})$ space bottleneck in constructing $\mathcal{D}$ arises precisely from this dot product estimation step, rather than from the clustering procedure itself. This observation motivates our technical improvements. In particular, the dot product estimation algorithm of Gluch et al. (2021) does not directly compute $\langle \boldsymbol{f}_x, \boldsymbol{f}_y \rangle$ for arbitrary vertex pairs. Instead, it applies a sequence of transformations and shows that estimating $\langle \boldsymbol{f}_x, \boldsymbol{f}_y \rangle$ can be reduced to computing the collision probability $(\boldsymbol{M}^t \mathbb{1}_x)^T (\boldsymbol{M}^t \mathbb{1}_y) = \langle \boldsymbol{M}^t \mathbb{1}_x, \boldsymbol{M}^t \mathbb{1}_y \rangle$, where $\boldsymbol{M}$ is the random walk transition matrix of $G$ and $\mathbb{1}_s$ is the indicator vector of vertex $s$.

Previous dot product oracle estimates $\langle \boldsymbol{M}^t \mathbb{1}_x, \boldsymbol{M}^t \mathbb{1}_y \rangle$ by performing $R \approx \sqrt{n}$ independent random walks of length $t = O(\frac{\log n}{\varphi^2})$ from each vertex $x$ and $y$, respectively. The endpoints of these walks are stored to construct empirical distributions, whose dot product is then computed. This approach requires $O(R)$ words of space and $O(Rt)$ time, tightly coupling space usage with computation time. In particular, to ensure sufficient accuracy, $R$ must be at least $\Omega(\sqrt{n})$, which implies that the space usage cannot be reduced below $O(\sqrt{n})$.

To reduce the memory requirement below $O(\sqrt{n})$ and achieve a more flexible trade-off between space and time, we propose a batch-based estimation strategy. The idea behind this approach is inspired by Canonne & Yang (2024), where a similar batching technique is used to design memory-efficient algorithms for uniformity testing under memory constraints. While the underlying technique is inspired by prior work, we are the first to apply this idea in the graph setting to rigorously analyze

random walks. Specifically, we partition the total of $R$ random walks into $B = R/M$ batches. In each batch, $M$ walks of length $t$ are performed from each vertex, and only the endpoints within the batch are stored to construct empirical distributions. The batch-level dot product is computed, and the final estimate is obtained by averaging over all batches. This approach reduces the space requirement to $O(M)$ words while keeping the total number of walks. By choosing $M$ smaller than $O(\sqrt{n})$, we can achieve a space-time trade-off satisfies $M \cdot R \approx n$. This allows for efficient estimation of the dot product even under memory constraints.

**Distinguishing 1-cluster vs. 2-cluster** The core idea of our algorithm (Alg. 5) for distinguishing the 1-cluster vs. 2-cluster is to reduce the task to detecting a spectral gap in the random walk operator. Specifically, we set $t = O(\log n / \varphi^2)$ so that in the 1-cluster case, the second largest eigenvalue of $\boldsymbol{M}^t$ becomes negligibly small, while in the 2-cluster case it remains exactly 1. To capture this behavior within bounded space, we avoid storing $\boldsymbol{M}^t$ explicitly and instead construct a compact surrogate matrix $\mathcal{G}$ using the batch-based strategy described above. This surrogate preserves the essential spectral information of $\boldsymbol{M}^t$, so that the separation between the two cases is faithfully reflected in the spectrum of $\mathcal{G}$. Consequently, analyzing $\mathcal{G}$ suffices to distinguish between the 1-cluster and 2-cluster cases using only $O(M)$ space.

To prove the lower bound, we note that analyzing the distribution of random walks of the two cases reveals a fundamental discrepancy: in the 1-cluster case, this distribution converges to uniformity over the entire set of points; whereas in the 2-cluster case, it decomposes into two separate uniform distributions, each concentrated over half of the points. Under a sublinear space constraint, the algorithm cannot store enough indices to reliably identify which cluster a given sample belongs to. We formalize this via the information-theoretic framework for distribution-testing lower bounds of Diakonikolas et al. (2019), showing that each observation provides only limited distinguishing information. Consequently, any algorithm requires a sufficient number of observations to achieve statistical confidence, implying the stated space-time trade-off lower bound.

A key novelty of our approach is a new reduction that connects random-walk-based graph clustering with space-bounded distribution testing. We construct paired hard instances and show how any random-walk algorithm for distinguishing 1-cluster vs. 2-cluster instances can be simulated in the distribution-testing setting. The key technical contribution is an inductive coupling argument ensuring that the random-walk histories remain indistinguishable in total-variation distance. This reduction is new and is what enables our space-time lower bound.

### 1.3 RELATED WORK

Peng (2020) (see also (Czumaj et al., 2015)) provided a robust sublinear spectral clustering oracle that constructs a data structure using $O(\sqrt{n} \cdot \text{poly}(\frac{k \log n}{\varepsilon}))$ bits of space[1] and answers any WHICH-CLUSTER$(G, x)$ in $O(\sqrt{n} \cdot \text{poly}(\frac{k \log n}{\varepsilon}))$ time. This oracle relies on a $\text{poly}(k) \log n$ conductance gap between inner and outer conductance and misclassifies at most $O(kn\sqrt{\varepsilon})$ vertices. Gluch et al. (2021) (resp. Shen & Peng (2023)[2]) gave a sublinear spectral clustering oracle that constructs a data structure using $O(n^{1-\delta+O(\varepsilon)} \cdot \text{poly}(\frac{k \log n}{\varepsilon}))$ (resp. $O(n^{1-\delta+O(\varepsilon)} \cdot \text{poly}(k \log n))$) bits of space and answers any WHICHCLUSTER$(G, x)$ in $O(n^{\delta+O(\varepsilon)} \cdot \text{poly}(\frac{k \log n}{\varepsilon}))$ (resp. $O(n^{\delta+O(\varepsilon)} \cdot \text{poly}(k \log n))$) time, where $\delta \in (0, \frac{1}{2}]$. These two oracles have different conductance gap and misclassification error.

Recently, Neumann & Peng (2022) studied designing sublinear spectral clustering oracles for signed graph. Kapralov et al. (2023) studied designing sublinear hierarchical clustering oracle for graphs exhibiting hierarchical structure. We defer other related works to Appendix B due to page constraint. Moreover, all omitted proofs are provided in the appendix.

### 2 PRELIMINARIES

Let $G = (V, E)$ denote an unweighted, undirected $d$-regular graph with $n$ vertices, where $V = \{1, 2, \ldots, n\}$. Let $i \in [n]$ denote $1 \leq i \leq n$. For a graph $G = (V, E)$, let $\boldsymbol{A} \in \mathbb{R}^{n \times n}$ denote the

---

[1]Although the paper does not explicitly state the space complexity, it can be directly inferred from the algorithm description.

[2]Shen & Peng (2023) stated their result for $\delta = 1/2$. Since their algorithm relies on the dot product oracle in Gluch et al. (2021), the guarantee extends naturally to any $\delta \in (0, \frac{1}{2}]$.

adjacency matrix of $G$, where $\boldsymbol{A}(i,j) = 1$ if $(i,j) \in E$, and $\boldsymbol{A}(i,j) = 0$ otherwise, $i,j \in [n]$. Let $\boldsymbol{D} \in \mathbb{R}^{n \times n}$ denote a diagonal matrix, where $\boldsymbol{D}(i,i) = d_i, i \in [n]$. Let $\boldsymbol{L} = \boldsymbol{D}^{-1}(\boldsymbol{D} - \boldsymbol{A})\boldsymbol{D}^{-1} = \boldsymbol{I} - \frac{\boldsymbol{A}}{d}$ denote the normalized Laplacian matrix of $G$, where $\boldsymbol{I} \in \mathbb{R}^{n \times n}$ is the identity matrix. For $\boldsymbol{L}$, we use $0 = \lambda_1 \leq \cdots \leq \lambda_n \leq 2$ to denote its eigenvalues and $\boldsymbol{u}_1, \ldots, \boldsymbol{u}_n \in \mathbb{R}^n$ to denote the corresponding eigenvectors. Without loss of generality, we assume $\{\boldsymbol{u}_1, \ldots, \boldsymbol{u}_n\}$ forms an orthonormal basis of $\mathbb{R}^n$. Let $\boldsymbol{U} = (\boldsymbol{u}_1, \ldots, \boldsymbol{u}_n) \in \mathbb{R}^{n \times n}$. Based on $\boldsymbol{U}$, we give the definition of spectral embedding (see Definition 2.1). Moreover, let $\boldsymbol{M} = \frac{1}{2}(\boldsymbol{I} + \frac{\boldsymbol{A}}{d}) = \boldsymbol{I} - \frac{\boldsymbol{L}}{2}$ denote the transition matrix of lazy random walk on $G$. That is, if the walker is currently at a vertex $x \in V$, then in the next step it stays at $x$ with probability $\frac{1}{2}$, or moves to each neighbor of $x$ with probability $\frac{1}{2d_x}$.

Let $\boldsymbol{a} \in \mathbb{R}^n$ denote a column vector (unless otherwise stated). For any two vectors $\boldsymbol{a}, \boldsymbol{b} \in \mathbb{R}^n$, we use $\langle \boldsymbol{a}, \boldsymbol{b} \rangle = \boldsymbol{a}^T \boldsymbol{b}$ to denote the dot product of $\boldsymbol{a}$ and $\boldsymbol{b}$. For any $x \in V$, let $\mathbb{1}_x \in \mathbb{R}^n$ denote the indicator vector of $x$, where $\mathbb{1}_x(i) = 1$ if $i = x$ and $0$ otherwise. For any symmetric matrix $\boldsymbol{B} \in \mathbb{R}^{n \times n}$, we use $v_i(\boldsymbol{B})$ to denote the $i$-th largest eigenvalues of $\boldsymbol{B}$.

**Definition 2.1** (spectral embedding)**.** Let $G = (V, E)$ be a graph. For any vertex $x \in V$, we use $\boldsymbol{f}_x \in \mathbb{R}^k$ to denote the *spectral embedding* of $x$, where $\boldsymbol{f}_x = \boldsymbol{U}_{[k]}^T \mathbb{1}_x = (\boldsymbol{u}_1(x), \ldots, \boldsymbol{u}_k(x))^T$.

**Definition 2.2** ($\varphi$-expander)**.** Let $G = (V, E)$ be a graph. Let $\varphi \in (0, 1)$. Let $\phi(G)$ denote the conductance of $G$ (see Definition 1.1). If $\phi(G) \geq \varphi$, then we call $G$ a *$\varphi$-expander*.

The supplementary preliminaries are deferred to Appendix C.

## 3 SPECTRAL CLUSTERING ORACLES WITH LITTLE MEMORY

In this section, we present and prove our main algorithmic result, stated in the theorem below. We emphasize that the resulting algorithms exhibit different trade-offs between the conductance gap ($\varphi$ vs. $\varepsilon$), the misclassification ratio, and the corresponding space-time bounds, depending on the clustering algorithms employed, either that of Gluch et al. (2021) or Shen & Peng (2023).

**Theorem 3.1.** *Let $k \geq 2$ be an integer, $\varphi, \varepsilon \in (0, 1)$ and $h_1(k, \varphi), h_2(k, \varepsilon)$ and $h_3(k, \varphi, \varepsilon)$ be three functions. Let $\varepsilon \ll h_1(k, \varphi)$. Let $G = (V, E)$ be a $d$-regular and $(k, \varphi, \varepsilon)$-clusterable graph with $C_1, \ldots, C_k$. Let $1 \leq M \leq O(\frac{n^{1/2 - O(\varepsilon/\varphi^2)}}{k})$ be a trade-off parameter, where $c$ is a large enough constant. There exists a sublinear spectral clustering oracle that, with probability at least $0.9$:*

- *constructs a data structure $\mathcal{D}$ using $\widetilde{O}_\varphi(h_2(k) \cdot n^{O(\varepsilon/\varphi^2)} \cdot M)$ bits of space,*
- *answers any WHICHCLUSTER query using $\mathcal{D}$ in $\widetilde{O}_\varphi(h_2(k) \cdot n^{1 + O(\varepsilon/\varphi^2)} \cdot \frac{1}{M})$ time[3],*
- *has $O(h_3(k, \varphi, \varepsilon))|C_i|$ misclassification error for each $i \in [k]$,*

*where we use $O_\varphi$ to suppress dependence on $\varphi$ and $\widetilde{O}$ to hide all $\mathrm{poly}(\log n)$ factors and:*

1. *if $h_1(k, \varphi) = \frac{\varphi^3}{\log k}$, then $h_2(k, \varepsilon) = (\frac{k}{\varepsilon})^{O(1)}$ and $h_3(k, \varphi, \varepsilon) = \frac{\varepsilon}{\varphi^3} \cdot \log k$;*
2. *if $h_1(k, \varphi) = \frac{\varphi^2 \cdot \gamma^3}{k^{\frac{9}{2}} \cdot \log^3 k}$, then $h_2(k) = (\frac{k}{\gamma})^{O(1)}$ and $h_3(k, \varphi, \varepsilon) = (\frac{\varepsilon}{\varphi^2})^{\frac{1}{3}} \cdot k^{\frac{3}{2}}$, where $\gamma \in (0.001, 1]$ is a constant such that for all $i \in [k]$, $\gamma \frac{n}{k} \leq |C_i| \leq \frac{n}{\gamma k}$.*

This section is organized as follows. In Section 3.1, we present our dot product oracle with little memory and the corresponding algorithms. In Section 3.2, we provide the proof of Item 2 of Theorem 3.1. The proof of the remaining case, Item 1, is deferred to Appendix F.

### 3.1 DOT PRODUCT ORACLE WITH LITTLE MEMORY

Recall that $\boldsymbol{f}_x$ denotes the spectral embedding of vertex $x$ (see Definition 2.1). Our objective in this section is to design a dot product oracle that approximates $\langle \boldsymbol{f}_x, \boldsymbol{f}_y \rangle$ while achieving a favorable space-time trade-off and ensuring small approximation error. The following theorem states the performance guarantees of our oracle. Proof is deferred to Appendix E.

**Theorem 3.2.** *Let $k \geq 2$ be an integer. Let $\varepsilon, \varphi \in (0, 1)$ with $\frac{\varepsilon}{\varphi^2} \leq \frac{1}{10^5}$. Let $G = (V, E)$ be a $d$-regular and $(k, \varphi, \varepsilon)$-clusterable graph. Let $\frac{1}{n^5} < \xi < 1$. Let $1 \leq M_{\mathrm{init}}, M_{\mathrm{query}} \leq O(\frac{n^{1/2 - 20\varepsilon/\varphi^2}}{k})$.*

---

[3]In order for the query time to be sublinear, $M$ must satisfy $M \geq n^{c \cdot \varepsilon/\varphi^2}$, where $c$ is a constant that is larger than the constant hidden in $O(\cdot)$-term of $n^{1 + O(\varepsilon/\varphi^2)}$.

*Then, with probability at least $1 - 2n^{-100}$, $\textsc{InitOracle}(G, k, \xi, M_{\mathrm{init}})$ (Alg. 3) computes a sublinear space matrix $\Psi$ of size $n^{O(\varepsilon/\varphi^2)} \cdot \log^2 n \cdot (\frac{k}{\xi})^{O(1)}$, such that the following property is satisfied:*

*for every pair of vertices $x, y \in V$, $\textsc{QueryDot}(G, x, y, \xi, \Psi, M_{\mathrm{query}})$ (Alg. 4) computes an output value $\langle \boldsymbol{f}_x, \boldsymbol{f}_y \rangle_{\mathrm{apx}}$ such that with probability at least $1 - 6n^{-100}$:*

$$|\langle \boldsymbol{f}_x, \boldsymbol{f}_y \rangle_{\mathrm{apx}} - \langle \boldsymbol{f}_x, \boldsymbol{f}_y \rangle| \leq \frac{\xi}{n}.$$

*Moreover, let $S_{\mathrm{init}}, T_{\mathrm{init}}$ be the space and time costs of $\textsc{InitOracle}(G, k, \xi, M_{\mathrm{init}})$ (Alg.3), and let $S_{\mathrm{query}}, T_{\mathrm{query}}$ be those of a single $\textsc{QueryDot}(G, x, y, \xi, \Psi, M_{\mathrm{query}})$ query (Alg.4). Then we have*

- $S_{\mathrm{init}} = (\frac{k}{\xi})^{O(1)} \cdot n^{O(\varepsilon/\varphi^2)} \cdot M_{\mathrm{init}} \cdot \log^4 n,$    $T_{\mathrm{init}} = (\frac{k}{\xi})^{O(1)} \cdot n^{1+O(\varepsilon/\varphi^2)} \cdot \frac{\log^4 n}{M_{\mathrm{init}}} \cdot \frac{1}{\varphi^2},$
- $S_{\mathrm{query}} = (\frac{k}{\xi})^{O(1)} \cdot n^{O(\varepsilon/\varphi^2)} \cdot M_{\mathrm{query}} \cdot \log^3 n,$    $T_{\mathrm{query}} = (\frac{k}{\xi})^{O(1)} \cdot n^{1+O(\varepsilon/\varphi^2)} \cdot \frac{\log^3 n}{M_{\mathrm{query}}} \cdot \frac{1}{\varphi^2}.$

Note that to ensure that $\textsc{InitOracle}(G, k, \xi, M_{\mathrm{init}})$ (Alg. 3) and $\textsc{QueryDot}(G, x, y, \xi, \Psi, M_{\mathrm{query}})$ (Alg. 4) run in sublinear time, it is required that $M_{\mathrm{init}}, M_{\mathrm{query}} \geq n^{c \cdot \varepsilon/\varphi^2}$, where $c$ is a constant that is larger than the constant hidden in $O(\cdot)$-term of $n^{1+O(\varepsilon/\varphi^2)}$ in both $T_{\mathrm{init}}$ and $T_{\mathrm{query}}$.

For initializing the dot product oracle, the previous dot product oracle in Gluch et al. (2021) requires at least $\widetilde{\Omega}(\sqrt{n})$ bits of space, whereas our proposed oracle can perform accurate estimation using at most $\widetilde{O}(\sqrt{n})$ bits of space, thus breaking the $\sqrt{n}$ barrier.

**The algorithm**   Algorithm 1 estimates the collision probability (i.e., $\langle \boldsymbol{M}^t \mathbb{1}_x, \boldsymbol{M}^t \mathbb{1}_x \rangle$) of the random walk distributions from two given vertices within a bounded space $\widetilde{O}(M)$. This bounded-space guarantee is achieved through our batch technique, and we are the first to apply this idea in the graph setting for analyzing random walks. Algorithm 2 computes an estimate of the Gram matrix $(\boldsymbol{M}^t \boldsymbol{S})^T (\boldsymbol{M}^t \boldsymbol{S})$ corresponding to the random walk distributions from a set $S$ of vertices, where $\boldsymbol{S} \in \mathbb{R}^{n \times |S|}$ is a matrix whose $i$-th column is an indicator vector $\mathbb{1}_v$ for $v \in S$, while operating within a bounded space $\widetilde{O}(M \cdot |S|^2)$. The formal guarantees of these two procedures are stated in Lemma 3.1 and Lemma E.5, respectively.

**Lemma 3.1.** *Let $k \geq 2$ be an integer and $\varphi, \varepsilon \in (0, 1)$. Let $G = (V, E)$ be a $d$-regular and $(k, \varphi, \varepsilon)$-clusterable graph. Let $\boldsymbol{M}$ be the random walk transition matrix of $G$. Let $Z$ be the output of $\textsc{EstRWDot}(G, R, t, M, x, y)$ (Alg. 1). Let $\sigma_{\mathrm{err}} > 0$. Let $c > 1$ be a large enough constant. For any $t \geq \frac{20 \log n}{\varphi^2}$ and any $x, y \in V$, if $R \geq \frac{c \cdot k^2 n^{-1+40\varepsilon/\varphi^2}}{\sigma_{\mathrm{err}}^2 M}$ and $1 \leq M \leq O(\frac{n^{1/2 - 20\varepsilon/\varphi^2}}{k})$, then with probability at least $0.99$, we have*

$$|Z - \langle \boldsymbol{M}^t \mathbb{1}_x, \boldsymbol{M}^t \mathbb{1}_y \rangle| \leq \sigma_{\mathrm{err}}.$$

*Moreover, $\textsc{EstRWDot}(G, R, t, M, x, y)$ runs in $O(Rt)$ time and uses $O(M \cdot \log n)$ bits of space.*

---

**Algorithm 1: EstRWDot**
$(G, R, t, M, x, y)$

---

1  $Z := 0, B := \frac{R}{M}$   ▷ $B$: number of batch
2  **for** $b = 1$ *to* $B$ **do**
3      Run $M$ independent random walks of length $t$ starting from $x$ (resp. from $y$)
4      Define $\widehat{\boldsymbol{p}}_x(i)$ (resp. $\widehat{\boldsymbol{p}}_y(i)$) as the fraction of randoms walks from $x$ (resp. from $y$) that end at $i$
5      $Z_b := \langle \widehat{\boldsymbol{p}}_x, \widehat{\boldsymbol{p}}_y \rangle, Z := Z + Z_b$
6  $Z := \frac{Z}{B}$
7  **return** $Z$

---

**Algorithm 2: EstColliProb**
$(G, R, t, M, I_S)$

---

1  $s := |I_S| = |\{s_1, \ldots, s_s\}|$
2  **for** $l = 1$ *to* $O(\log n)$ **do**
3      **for** $i = 1$ *to* $s$ **do**
4          **for** $j = i$ *to* $s$ **do**
5              $\mathcal{G}_l(j, i) := \mathcal{G}_l(i, j) :=$ $\textsc{EstRWDot}(G, R, t, M, s_i, s_j)$

6  Let $\mathcal{G}$ be a matrix obtained by taking the entrywise median of $\mathcal{G}_l$'s
7  **return** $\mathcal{G}$

---

Algorithm 3 initializes the dot product oracle by constructing a compact matrix $\Psi$ within approximately bounded space $\widetilde{O}(M)$. Then Algorithm 4 leverages $\Psi$ to estimate $\langle \boldsymbol{f}_x, \boldsymbol{f}_y \rangle$ while still

operating under the same bounded space. The formal guarantees of these two procedures are stated in Theorem 3.2.

---

**Algorithm 3:** INITORACLE $(G, k, \xi, M_{\text{init}})$

1  $t := \frac{20 \log n}{\varphi^2}$

2  $R_{\text{init}} := \Theta\left(\frac{n^{1+920\varepsilon/\varphi^2}}{M_{\text{init}}} \cdot \frac{k^{14}}{\xi^2}\right)$

3  $s := O(n^{480 \cdot \varepsilon/\varphi^2} \cdot \log n \cdot k^8/\xi^2)$

4  Let $I_S = \{s_1, \ldots, s_s\}$ be the multiset of $s$ indices chosen i.u.r. from $V = \{1, \ldots, n\}$

5  $\mathcal{G} := \text{ESTCOLLIPROB}(G, R_{\text{init}}, t, M_{\text{init}}, I_S)$

6  Let $\frac{n}{s} \cdot \mathcal{G} := \widehat{W}\widehat{\Sigma}\widehat{W}^T$ be the eigendecomposition of $\frac{n}{s} \cdot \mathcal{G}$

7  **if** $\widehat{\Sigma}^{-1}$ *exists* **then**

8  $\quad \Psi := \frac{n}{s} \cdot \widehat{W}_{[k]}\widehat{\Sigma}_{[k]}^{-2}\widehat{W}_{[k]}^T \quad \triangleright \Psi \in \mathbb{R}^{s \times s}$

9  $\quad$ return $\Psi$

---

**Algorithm 4:** QUERYDOT $(G, x, y, \xi, \Psi, M_{\text{query}})$

1  $t := \frac{20 \log n}{\varphi^2}$

2  $R_{\text{query}} := \Theta\left(\frac{n^{1+440\varepsilon/\varphi^2}}{M_{\text{query}}} \cdot \frac{k^6}{\xi^2}\right)$

3  **for** $l = 1$ *to* $O(\log n)$ **do**

4  $\quad$ **for** $i = 1$ *to* $s$ **do**

5  $\quad\quad \boldsymbol{x}_l(i) := \text{ESTRWDOT}(G, R_{\text{query}}, t, M_{\text{query}}, x, s_i)$

6  $\quad\quad \boldsymbol{y}_l(i) := \text{ESTRWDOT}(G, R_{\text{query}}, t, M_{\text{query}}, y, s_i)$

7  Let $\boldsymbol{\alpha}_x$ (resp. $\boldsymbol{\alpha}_y$) be a vector obtained by taking entrywise median of $\boldsymbol{x}_l$'s (resp. $\boldsymbol{y}_l$'s) $\quad \triangleright \boldsymbol{\alpha}_x, \boldsymbol{\alpha}_y \in \mathbb{R}^s$

8  return $\langle \boldsymbol{f}_x, \boldsymbol{f}_y \rangle_{\text{apx}} = \boldsymbol{\alpha}_x^T \Psi \boldsymbol{\alpha}_y$

---

## 3.2 CLUSTERING ORACLE: ITEM 2 OF THEOREM 3.1

We now present the proof of Item 2 of Theorem 3.1 and give a clustering oracle with the corresponding space-time trade-off. Item 2, which addresses a sublinear spectral clustering oracle under a poly$(k)$ conductance gap. Our sublinear spectral clustering oracle closely follows the construction in Shen & Peng (2023), except that we substitute our new dot product oracle from Section 3.1 in place of theirs.

**High-level idea of the algorithm** Now we briefly outline the main idea of the oracle. Shen & Peng (2023) showed that for most vertices in a $(k, \varphi, \varepsilon)$-clusterable graph, if $x, y \in V$ belong to the same cluster, then $\langle \boldsymbol{f}_x, \boldsymbol{f}_y \rangle \approx \frac{k}{n}$, otherwise, $\langle \boldsymbol{f}_x, \boldsymbol{f}_y \rangle \approx 0$. Leveraging this property, we can design a clustering oracle as follows: it first samples $s = \frac{k \log k}{\gamma}$ vertices to form a set $S$, and for each pair $u, v \in S$, it computes the dot product $\langle \boldsymbol{f}_u, \boldsymbol{f}_v \rangle_{\text{apx}}$ using our new dot product oracle. If the value is large, an edge $(u, v)$ is added to the initially empty similarity graph $H = (S, \emptyset)$. At query time, the oracle uses $H$ and its connected components to determine the cluster assignment of vertices. We provide a full description of the clustering oracle in Appendix G. Now we present the proof of Item 2 in Theorem 3.1 as follows.

*Proof of Item 2 in Theorem 3.1.* **Space and runtime.** In the preprocessing phase, CONSTRUCTOR-ACLE$(G, k, \varphi, \varepsilon, \gamma, M)$ (Alg. 12) invokes our INITORACLE$(G, k, \xi, M)$ (Alg. 3) one time to get a matrix $\Psi$ (see line 5 of Alg. 12), then CONSTRUCTORACLE$(G, k, \varphi, \varepsilon, \gamma, M)$ invokes our QUERY-DOT$(G, u, v, \xi, \Psi, M)$ $O((k^2 \log^2 k)/\gamma^2)$ times (see lines $6 \sim 9$ of Alg. 12) to get a similarity graph $H$. Therefore, CONSTRUCTORACLE$(G, k, \varphi, \varepsilon, \gamma, M)$ uses $S_{\text{init}} + O((k^2 \log^2 k)/\gamma^2) \cdot S_{\text{query}}$ bits of space. Using Theorem 3.2, we get that CONSTRUCTORACLE$(G, k, \varphi, \varepsilon, \gamma, M)$ uses $O(n^{O(\varepsilon/\varphi^2)} \cdot M \cdot \text{poly}(\frac{k \log n}{\gamma}))$ bits of space to get matrix $\Psi$ and a similarity graph $H$.

In the query phase, WHICHCLUSTER$(G, x, M)$ (Alg. 14) invokes SEARCH$(H, \ell, x, M)$ (Alg. 13) one time. SEARCH$(H, \ell, x, M)$ invokes our QUERYDOT$(G, u, x, \xi, \Psi, M)$ $O((k \log k)/\gamma)$ times (see lines $1 \sim 2$ of Alg. 13) and relies on the similarity graph $H$ (see lines $3 \sim 6$ of Alg. 13). Therefore, WHICHCLUSTER$(G, x, M)$ uses $O((k \log k)/\gamma) \cdot S_{\text{query}}$ bits of space and runs in $O((k \log k)/\gamma) \cdot T_{\text{query}}$ time. Using Theorem 3.2, we get that WHICHCLUSTER$(G, x, M)$ uses $O(n^{O(\varepsilon/\varphi^2)} \cdot M \cdot \text{poly}(\frac{k \log n}{\gamma}))$ bits of space and runs in $O(n^{1+O(\varepsilon/\varphi^2)} \cdot \frac{1}{M} \cdot \text{poly}(\frac{k \log n}{\gamma\varphi}))$ time.

Thus, the oracle constructs a data structure $\mathcal{D}$ (including $\Psi$, similarity graph $H$ etc) using $O(n^{O(\varepsilon/\varphi^2)} \cdot M \cdot \text{poly}(\frac{k \log n}{\gamma}))$ bits of space. Using $\mathcal{D}$, any WHICHCLUSTER$(G, x)$ query can be answered by Alg. 14 in $O(n^{1+O(\varepsilon/\varphi^2)} \cdot \frac{1}{M} \cdot \text{poly}(\frac{k \log n}{\gamma\varphi}))$ time.

**Correctness.** Since the correctness guarantees (i.e., conductance gap and misclassification error) of the clustering oracle rely on the properties of the dot product oracle, and our dot product oracle satisfies the same correctness guarantees with the previous one, the correctness of the overall clustering oracle follows directly from the correctness of the clustering oracle in Shen & Peng (2023). □

## 4  Distinguishing 1-cluster vs. 2-cluster

**The algorithm and sketch of its analysis**   Now we present Alg. 5 for solving the 1-cluster vs. 2-cluster problem, which is based on estimating the second largest eigenvalue of $\mathbf{M}^t$ using a subroutine ESTCOLLIPROB (Alg. 2) from Section 3.1.

---

**Algorithm 5:** DISTINGUISH$(G, M)$

---

1 $t := \frac{20 \log n}{\varphi^2}$, $R := \Theta(\frac{n}{M})$, $s := O(\log n)$
2 Let $I_S = \{s_1, \ldots, s_s\}$ be the multiset of $s$ indices chosen independently and uniformly at random from $V = \{1, \ldots, n\}$
3 $\mathcal{G} := \text{ESTCOLLIPROB}(G, R, t, M, I_S)$
4 Let $v_2(\frac{n}{s}\mathcal{G})$ be the second largest eigenvalue of matrix $\frac{n}{s}\mathcal{G}$
5 **if** $\left(v_2(\frac{n}{s}\mathcal{G})\right)^2 < 0.6$ **then**
6     return "1-cluster"
7 return "2-cluster"

---

The formal guarantee of this algorithm is given in Theorem 1.2, whose proof is deferred to Appendix H. Here, we provide a proof sketch.

Consider the case when the input graph $G$ is a $\varphi$-expander. By Cheeger's inequality (Lemma H.1), we get that the second smallest eigenvalue of $\mathbf{L}$ satisfies $\lambda_2 \geq \varphi^2/2$. Equivalently, the lazy random walk matrix $\mathbf{M} = \mathbf{I} - \mathbf{L}/2$ has its second largest eigenvalue $v_2(\mathbf{M}) \leq 1 - \varphi^2/4$. In contrast, if $G$ consists of two disjoint $\varphi$-expanders of equal size, then $\lambda_2 = 0$ and hence $v_2(\mathbf{M}) = 1$. Setting $t = O(\log n / \varphi^2)$, we obtain that in the 1-cluster case, the contribution of $v_2(\mathbf{M}) \leq n^{-10}$, while in the 2-cluster case, $v_2(\mathbf{M})$ remains exactly 1. Thus, $\mathbf{M}^t$ exhibits a clear spectral gap between the two cases. Alg. 5 constructs an approximation $\mathcal{G} \approx (\mathbf{M}^t \mathbf{S})^T (\mathbf{M}^t \mathbf{S}) \in \mathbb{R}^{O(\log n) \times O(\log n)}$ within bounded space, where each column of $\mathbf{M}^t \mathbf{S}$ corresponds to the $t$-step lazy random walk distribution starting from a vertex in the sampled set $I_S$. The second largest eigenvalue of $\mathcal{G}$ closely reflects that of $\mathbf{M}^t$, thereby preserving the above separation (see Lemma H.4 for the formal statement). Moreover, since $\mathcal{G}$ is a small matrix, we can afford to perform an eigen-decomposition on it directly. Consequently, examining the spectrum of $\mathcal{G}$ suffices to distinguish between the 1-cluster and 2-cluster cases using $\widetilde{O}(M)$ bits of space and $\widetilde{O}(n/M)$ time.

**The lower bound**   The lower bound for distingushing 1-cluster vs. 2-cluster is summarized in Theorem 1.3. The main proof of Theorem 1.3 is presented in Appendix I and comprises two parts. First, we establish a lower bound for distinguishing between a uniform distribution over all vertices and two separate uniform distributions each over half of the vertex set. We demonstrate that under a space constraint of $S$, the information regarding the underlying case can only increase by $O(S/n)$ per observation. Consequently, the total number of observations $T$ must satisfy $T \cdot O(S/n) = \Omega(1)$, which directly implies the space-time trade-off lower bound $S \cdot T = \Omega(n)$ (see Theorem I.2).

Second, by analyzing the random walk distributions in the 1-cluster and 2-cluster cases, we observe that these distributions closely approximate the two aforementioned reference distributions. To finalize the reduction, it is necessary to demonstrate that deviations from uniformity do not significantly alter the final memory state distribution. The key challenge lies in the cumulative effect of sampling distribution discrepancies at each step, which collectively influence the memory state. To quantify this discrepancy, we adopt the total variation distance as a metric and employ a mathematical induction argument. This approach shows that the discrepancy in the memory state distribution does not substantially amplify after each sampling step. Specifically, the incremental increase in discrepancy is proportional to the difference between the sample distributions and remains controllable. Consequently, the overall discrepancy is bounded by the sum of these incremental increases and remains negligible throughout the process.

## 5 EXPERIMENTS

To evaluate the space-time trade-off of our sublinear spectral clustering oracles, we conducted experiments in Python on graphs generated from the stochastic block model (SBM) with parameters $n$ (num of vertices), $k$ (num of clusters), and edge probabilities $p$ (within-cluster) and $q$ (between-cluster). Experiments were run on a server with an Intel(R) Xeon(R) Platinum 8562Y processor (2.80 GHz) and 768 GB RAM. Each reported data is the average over five independent runs.

We implemented two variants of the poly$(k)$-conductance-gap clustering oracle[4]: the original oracle from Shen & Peng (2023), and our memory-efficient variant that operates within a smaller space. For each, we recorded the number of words stored in each component of the data structure $\mathcal{D}$ as a proxy for space $S$, evaluated accuracy (the fraction of vertices correctly classified), the success rate (i.e., the fraction of successful runs among 5 runs[5]). Both variants used the same number of sampled vertices, random walk length, and median-trick repetitions; differences arose only in space-time-related parameters. We instantiated this setup on an SBM graph with $n = 3000$, $k = 3$, $p = 0.07$, and $q = 0.002$, yielding clusters of 1000 vertices each. Additional implementation details are provided in Appendix J.

**Space efficiency**  Prior sublinear spectral clustering oracles require at least $\Omega(\sqrt{n})$ space to construct data structure $\mathcal{D}$. In contrast, our clustering oracle allows constructing $\mathcal{D}$ using substantially less space, well below $\sqrt{n}$. In this section, we provide experimental evidence to validate this improvement.

Table 2: Comparison of space usage for clustering oracles, with 10400 words used as the baseline.

| clustering oracle | ours | | | previous | | | |
|---|---|---|---|---|---|---|---|
| space (# of words) | 9900 | 10100 | **10400** | 34840 | 43888 | 44383 | 61223 |
| space ($\times$ baseline) | 0.95$\times$ | 0.97$\times$ | 1$\times$ | **3.35$\times$** | 4.22$\times$ | **4.27$\times$** | 5.89$\times$ |
| success rate for constructing $\mathcal{D}$ | 1 | 1 | **1** | **0** | 0.6 | **1** | 1 |
| accuracy | 0.9833 | 0.9900 | **0.9907** | **0** | 0.9860 | **0.9997** | 1.0000 |

Table 2 demonstrate that our clustering oracle achieves high accuracy using substantially less space (10400 words as $1\times$). In contrast, the previous clustering oracle requires $4.27$ times of the baseline space to achieve comparable accuracy, and even when given $3.35$ times the baseline space, it fails to construct $\mathcal{D}$ successfully (i.e., success rate is 0). These results confirm that our approach significantly improves space efficiency without compromising accuracy.

**Space-time trade-off**  As established in Theorem 3.1, there is a trade-off between the space $S$ required to construct $\mathcal{D}$ and the query time $T$, satisfying $S \cdot T \approx \widetilde{O}(n^{1+O(\varepsilon)})$, where $\varepsilon$ is the small constant corresponding to the outer conductance.

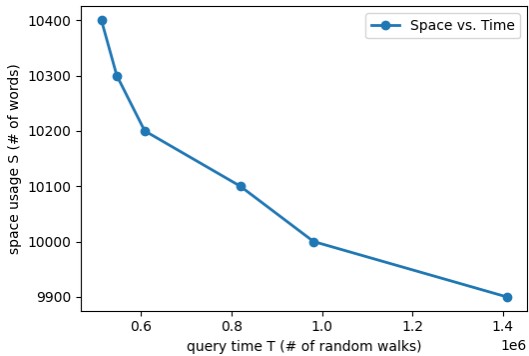

To validate this experimentally, we also measured $S$ as the total number of words stored to construct $\mathcal{D}$. We use the total number of random walks per WHICHCLUSTER query as a proxy for time $T$, since this dominates the query cost. Across all tested parameter settings, the oracle maintains high accuracy ($0.9833 \sim 1$), confirming the practical validity of the configurations used.

Figure 1 plots $S$ (y-axis) versus $T$ (x-axis), illustrating the space-time trade-off: memory usage decreases as query time increases, and vice versa, consistent with the theoretical bound.

Figure 1: Space-time trade-off of the sublinear spectral clustering oracle, showing $S, T$ are inversely proportional.

---

[4]We did not experiment with the $\log(k)$-conductance-gap oracle due to its impractical runtime of $2^{\text{poly}(k)} \cdot n^{1+O(\varepsilon)} \cdot \frac{1}{M}$ for constructing $\mathcal{D}$.

[5]If the available space is too limited, the construction of the similarity graph $H$ may yield either too many or too few connected components, in which case the construction of $\mathcal{D}$ fails.

ACKNOWLEDGMENTS

Ranran Shen and Pan Peng are supported in part by NSFC Grant 62272431 and Quantum Science and Technology - National Science and Technology Major Project (Grant No. 2021ZD0302901). Xiaoyi Zhu and Zengfeng Huang are supported in part by National Natural Science Foundation of China No. 62276066.

ETHICS STATEMENT

This work is purely theoretical and algorithmic in nature. Our experimental evaluation is conducted solely on synthetic datasets generated from the stochastic block model (SBM). The research does not involve human subjects, personal data, or other sensitive information. We do not anticipate any immediate ethical, societal, or environmental risks arising from our methods or results.

REPRODUCIBILITY STATEMENT

We have taken several steps to ensure the reproducibility of our work. All theoretical results are stated formally in the main text and accompanied by complete proofs in the appendix. The assumptions underlying our results are explicitly described. For the experimental evaluation, we used standard stochastic block model (SBM) graphs to ensure reproducibility. Implementation details and parameter settings are included in Appendix J.

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

# Appendix

The appendix is organized as follows.

## A   THE USE OF LARGE LANGUAGE MODELS (LLMS)

During the preparation of this manuscript, we mainly used ChatGPT to assist with English writing. Specifically, the model was employed to improve the fluency of sentences, check grammar, and suggest stylistic refinements. We emphasize that all theoretical contributions, proofs, and experimental results (including code implementation, simulations, and results collection) were developed and verified solely by the authors without the involvement of LLMs. The use of LLMs did not influence the research process, methodology, or the originality of the results presented in this paper.

## B   OTHER RELATED WORK

**Property testing**   Besides the above most directly related work on sublinear spectral clustering oracles, several other research directions are also relevant to our study. One line of work is property testing (i.e., *testing graph clusterability*), where the goal is to quickly distinguish whether a graph can be partitioned into $k$ clusters with high inner conductance, or whether it is far from having such clustering. For example, Czumaj et al. (2015) studied testing whether a graph admits a good cluster structure in the adjacency list query model, providing algorithms with sublinear query time. This direction was later advanced by Chiplunkar et al. (2018). While property testing algorithms do not provide explicit cluster assignments, they capture the feasibility of clustering in sublinear resources and thus serve as an important precursor to oracle-based approaches like ours. For example, Czumaj et al. (2015) implicitly yields a sublinear spectral clustering oracle under a $\log n$ conductance gap. This was later extended by Peng (2020), who developed a robust oracle capable of handling noise.

**Local graph clustering**   Another line of related work is *local graph clustering* (Andersen et al., 2006; Spielman & Teng, 2013; Zhu et al., 2013; Gharan & Trevisan, 2014; Andersen et al., 2016). The goal of this category is to identify a cluster associated with a given vertex. In this setting, the algorithm outputs a set of vertices related to the input vertex, and its running time and memory usage are bounded by the size of the output cluster, up to a weak dependence on $n$. In particular, when the graph contains $k$ clusters and $n$ vertices, the complexity can be as large as $\Omega(n/k)$.

**Grapah problems under limited memory**   Recently, there has been a surge of work on understanding learning under limited memory. Graph problems inherently require substantial space and time to compute, and have attracted increasing attention. One line of research focuses on the semi-streaming model where the algorithm is permitted $O(n \cdot \text{poly}(\log n))$ space. Both upper bound algorithms and lower bound results are proposed for various graph problems, including Maximal Independent Set (Assadi et al., 2024) and Matching (Kapralov, 2013). There is also significant work on the Massively Parallel Computation model, where machines have sublinear memory to solve the graph problems

(Behnezhad et al., 2019; Łącki et al., 2020; Nowicki & Onak, 2021; Assadi et al., 2019; Ghaffari & Nowicki, 2020).

## C    SUPPLEMENTARY PRELIMINARIES

For a vector $\boldsymbol{a} = (\boldsymbol{a}(1), \ldots, \boldsymbol{a}(n))^T$, the $p$-norm ($p \geq 1$) of $\boldsymbol{a}$ is defined to be $\|\boldsymbol{a}\|_p = (\sum_{i=1}^{n} |\boldsymbol{a}(i)|^p)^{\frac{1}{p}}$. For any matrix $\boldsymbol{B} \in \mathbb{R}^{n \times n}$, we use $\|\boldsymbol{B}\|_F = \sqrt{\sum_{i=1}^{n} \sum_{j=1}^{n} \boldsymbol{B}^2(i, j)}$ to denote the Frobenius norm of $\boldsymbol{B}$, $\|\boldsymbol{B}\|_2 = \max_{\boldsymbol{x} \in \mathbb{R}^n, \|\boldsymbol{x}\|_2 = 1} \|\boldsymbol{B}\boldsymbol{x}\|_2$ to denote the spectral norm of $\boldsymbol{B}$ and $\boldsymbol{B}_{[i]}$ to denote the first $i$ columns of $\boldsymbol{B}$, $1 \leq i \leq n$.

**Definition C.1** (TV distance). For two probability distributions $\boldsymbol{p}, \boldsymbol{q}$ over $[n]$, the *total variance distance* (i.e., TV distance) of $\boldsymbol{p}, \boldsymbol{q}$ is defined to be

$$d_{\text{TV}}(\boldsymbol{p}, \boldsymbol{q}) = \frac{1}{2} \|\boldsymbol{p} - \boldsymbol{q}\|_1.$$

**Fact C.1.** *For any vector $\boldsymbol{p} \in \mathbb{R}^n$, we have $\|\boldsymbol{p}\|_4^2 \leq \|\boldsymbol{p}\|_2^2$.*

*Proof.* Let $\|\boldsymbol{p}\|_\infty = \max_{i=1}^{n} |\boldsymbol{p}(i)|$. Then, we have

$$\begin{aligned}
\|\boldsymbol{p}\|_4^2 &= \sqrt{\sum_{i=1}^{n} \boldsymbol{p}^4(i)} \leq \sqrt{\sum_{i=1}^{n} \boldsymbol{p}^2(i) \cdot \|\boldsymbol{p}\|_\infty^2} \\
&= \sqrt{\|\boldsymbol{p}\|_\infty^2} \sqrt{\sum_{i=1}^{n} \boldsymbol{p}^2(i)} \\
&\leq \sqrt{\sum_{i=1}^{n} \boldsymbol{p}^2(i)} \sqrt{\sum_{i=1}^{n} \boldsymbol{p}^2(i)} \\
&= \|\boldsymbol{p}\|_2^2.
\end{aligned}$$

$\square$

## D    FROM $d$-BOUNDED GRAPHS TO $d$-REGULAR GRAPHS

Although we state our results for $d$-regular graphs, they extend naturally to $d$-bounded graphs, i.e., graphs in which every vertex has degree at most $d$. The extension is straightforward: for a $d$-bounded graph $G' = (V, E')$, for every $x \in V$, we can add $d - d_x$ self-loops with weight $\frac{1}{2}$ to $x$ to get a $d$-regular graph $G = (V, E)$. Note that the lazy random walk on $G$ is equivalent to the random walk on $G'$, with the random walk satisfying that if the walker is currently at $x \in V$, then in the next step it stays at $x$ with probability $1 - \frac{d_x}{2d}$, or moves to each neighbor of $x$ with probability $\frac{1}{2d_x}$.

## E    PROOF OF THEOREM 3.2

**Theorem E.1** (Restate of Theorem 3.2). *Let $k \geq 2$ be an integer. Let $\varepsilon, \varphi \in (0, 1)$ with $\frac{\varepsilon}{\varphi^2} \leq \frac{1}{10^5}$. Let $G = (V, E)$ be a $d$-regular and $(k, \varphi, \varepsilon)$-clusterable graph. Let $\frac{1}{n^5} < \xi < 1$. Let $1 \leq M_{\text{init}}, M_{\text{query}} \leq O(\frac{n^{1/2 - 20\varepsilon/\varphi^2}}{k})$. Then, with probability at least $1 - 2n^{-100}$, $\text{INITORACLE}(G, k, \xi, M_{\text{init}})$ (Alg. 3) computes a sublinear space matrix $\Psi$ of size $n^{O(\varepsilon/\varphi^2)} \cdot \log^2 n \cdot (\frac{k}{\xi})^{O(1)}$, such that the following property is satisfied:*

*for every pair of vertices $x, y \in V$, $\text{QUERYDOT}(G, x, y, \xi, \Psi, M_{\text{query}})$ (Alg. 4) computes an output value $\langle \boldsymbol{f}_x, \boldsymbol{f}_y \rangle_{\text{apx}}$ such that with probability at least $1 - 6n^{-100}$:*

$$|\langle \boldsymbol{f}_x, \boldsymbol{f}_y \rangle_{\text{apx}} - \langle \boldsymbol{f}_x, \boldsymbol{f}_y \rangle| \leq \frac{\xi}{n}.$$

*Moreover, let $T_{\text{init}}, S_{\text{init}}$ be the time and space costs of $\text{INITORACLE}(G, k, \xi, M_{\text{init}})$ (Alg.3), and let $T_{\text{query}}, S_{\text{query}}$ be those of a single $\text{QUERYDOT}(G, x, y, \xi, \Psi, M_{\text{query}})$ query (Alg.4). Then we have*

- $T_{\text{init}} = \left(\frac{k}{\xi}\right)^{O(1)} \cdot n^{1+O(\varepsilon/\varphi^2)} \cdot \frac{\log^4 n}{M_{\text{init}}} \cdot \frac{1}{\varphi^2}$,
- $S_{\text{init}} = \left(\frac{k}{\xi}\right)^{O(1)} \cdot n^{O(\varepsilon/\varphi^2)} \cdot M_{\text{init}} \cdot \log^4 n$
- $T_{\text{query}} = \left(\frac{k}{\xi}\right)^{O(1)} \cdot n^{1+O(\varepsilon/\varphi^2)} \cdot \frac{\log^3 n}{M_{\text{query}}} \cdot \frac{1}{\varphi^2}$,
- $S_{\text{query}} = \left(\frac{k}{\xi}\right)^{O(1)} \cdot n^{O(\varepsilon/\varphi^2)} \cdot M_{\text{query}} \cdot \log^3 n$.

To prove Theorem 3.2, we begin by analyzing $Z_b$ defined in Alg. 1. The following lemma shows that $Z_b$ is an unbiased estimator of $\langle \boldsymbol{M}^t \mathbb{1}_x, \boldsymbol{M}^t \mathbb{1}_x \rangle$ and quantifies its variance.

**Lemma E.1.** *Let $G = (V, E)$ be a graph. Let $R, t, M$ be integers, where $1 \le M \le R$. Let $x, y \in V$ be two vertices. Let $\boldsymbol{M}$ be the random walk transition matrix of $G$. Let $Z_b$ ($1 \le b \le \frac{R}{M}$) be the random variable defined in $\text{ESTRWDOT}(G, R, t, M, x, y)$ (see line 6 of Alg. 1). Then, we have*

$$\mathbb{E}[Z_b] = \langle \boldsymbol{M}^t \mathbb{1}_x, \boldsymbol{M}^t \mathbb{1}_y \rangle,$$

$$\text{Var}[Z_b] \le \frac{1}{M^2} \|\boldsymbol{M}^t \mathbb{1}_x\|_2 \cdot \|\boldsymbol{M}^t \mathbb{1}_y\|_2 + \frac{1}{M} \left( \|\boldsymbol{M}^t \mathbb{1}_x\|_2 \cdot \|\boldsymbol{M}^t \mathbb{1}_y\|_2^2 + \|\boldsymbol{M}^t \mathbb{1}_x\|_2^2 \cdot \|\boldsymbol{M}^t \mathbb{1}_y\|_2 \right).$$

*Proof.* Run $M$ random walks of length $t$ from $x$ (resp. from $y$). Let $\boldsymbol{c}_x(i)$ (resp. $\boldsymbol{c}_y(i)$) denote the number of random walks from $x$ (resp. from $y$) that end at vertex $i$. It's clear that we have $\widehat{\boldsymbol{p}}_x(i) = \frac{\boldsymbol{c}_x(i)}{M}$ and $\widehat{\boldsymbol{p}}_y(i) = \frac{\boldsymbol{c}_y(i)}{M}$ (see lines $4 \sim 5$ of Alg. 1). Let $\boldsymbol{p}_x = \boldsymbol{M}^t \mathbb{1}_x$ (resp. $\boldsymbol{p}_y = \boldsymbol{M}^t \mathbb{1}_y$) be the probability distribution of a length $t$ random walk starting from $x$ (resp. from $y$). Note that $\boldsymbol{c}_x(i) \sim \text{Binomial}(M, \boldsymbol{p}_x(i))$ and $\boldsymbol{c}_y(i) \sim \text{Binomial}(M, \boldsymbol{p}_y(i))$. According to line 6 of Alg. 1, we have $Z_b = \langle \widehat{\boldsymbol{p}}_x, \widehat{\boldsymbol{p}}_y \rangle$. Therefore, about $\mathbb{E}[Z_b]$, we have

$$\begin{aligned}
\mathbb{E}[Z_b] &= \langle \widehat{\boldsymbol{p}}_x, \widehat{\boldsymbol{p}}_y \rangle \\
&= \mathbb{E}\left[ \sum_{i=1}^n \widehat{\boldsymbol{p}}_x(i) \widehat{\boldsymbol{p}}_y(i) \right] \\
&= \frac{1}{M^2} \cdot \sum_{i=1}^n \mathbb{E}[\boldsymbol{c}_x(i) \boldsymbol{c}_y(i)] \\
&= \frac{1}{M^2} \cdot \sum_{i=1}^n \mathbb{E}[\boldsymbol{c}_x(i)] \mathbb{E}[\boldsymbol{c}_y(i)] \\
&= \frac{1}{M^2} \cdot \sum_{i=1}^n M \boldsymbol{p}_x(i) M \boldsymbol{p}_y(i) \\
&= \sum_{i=1}^n \boldsymbol{p}_x(i) \boldsymbol{p}_y(i) \\
&= \langle \boldsymbol{p}_x, \boldsymbol{p}_y \rangle = \langle \boldsymbol{M}^t \mathbb{1}_x, \boldsymbol{M}^t \mathbb{1}_y \rangle.
\end{aligned}$$

About $\text{Var}[Z_b]$, since $\text{Var}[Z_b] = \mathbb{E}[Z_b^2] - (\mathbb{E}[Z_b])^2$, it suffices to calculate $\mathbb{E}[Z_b^2]$ to get $\text{Var}[Z_b]$.

$$\begin{aligned}
\mathbb{E}[Z_b^2] &= \mathbb{E}\left[ \langle \widehat{\boldsymbol{p}}_x, \widehat{\boldsymbol{p}}_y \rangle^2 \right] \\
&= \mathbb{E}\left[ \left( \sum_{i=1}^n \widehat{\boldsymbol{p}}_x(i) \widehat{\boldsymbol{p}}_y(i) \right)^2 \right] \\
&= \mathbb{E}\left[ \sum_{i=1}^n \sum_{j=1}^n \widehat{\boldsymbol{p}}_x(i) \widehat{\boldsymbol{p}}_y(i) \widehat{\boldsymbol{p}}_x(j) \widehat{\boldsymbol{p}}_y(j) \right]
\end{aligned}$$

$$= \frac{1}{M^4} \sum_{i=1}^{n} \sum_{j=1}^{n} \mathbb{E}\left[\boldsymbol{c}_x(i)\boldsymbol{c}_y(i)\boldsymbol{c}_x(j)\boldsymbol{c}_y(j)\right]$$

$$= \frac{1}{M^4} \sum_{i=1}^{n} \sum_{j=1}^{n} \mathbb{E}\left[\boldsymbol{c}_x(i)\boldsymbol{c}_x(j)\right] \cdot \mathbb{E}\left[\boldsymbol{c}_y(i)\boldsymbol{c}_y(j)\right]$$

$$= \frac{1}{M^4} \sum_{i=1}^{n} \mathbb{E}\left[\boldsymbol{c}_x^2(i)\right] \cdot \mathbb{E}\left[\boldsymbol{c}_y^2(i)\right] + \frac{1}{M^4} \sum_{i=1}^{n} \sum_{j=1,j \neq i}^{n} \mathbb{E}\left[\boldsymbol{c}_x(i)\boldsymbol{c}_x(j)\right] \cdot \mathbb{E}\left[\boldsymbol{c}_y(i)\boldsymbol{c}_y(j)\right].$$

For convenience, we use $A_1$ to denote $\frac{1}{M^4} \sum_{i=1}^{n} \mathbb{E}\left[\boldsymbol{c}_x^2(i)\right] \cdot \mathbb{E}\left[\boldsymbol{c}_y^2(i)\right]$ and $A_2$ to denote $\frac{1}{M^4} \sum_{i=1}^{n} \sum_{j=1,j \neq i}^{n} \mathbb{E}\left[\boldsymbol{c}_x(i)\boldsymbol{c}_x(j)\right] \cdot \mathbb{E}\left[\boldsymbol{c}_y(i)\boldsymbol{c}_y(j)\right]$.

Since $\boldsymbol{c}_x(i) \sim \text{Binomial}(M, \boldsymbol{p}_x(i))$, we have $\mathbb{E}[\boldsymbol{c}_x(i)] = M\boldsymbol{p}_x(i)$ and $\mathbb{E}[\boldsymbol{c}_x^2(i)] = \text{Var}[\boldsymbol{c}_x(i)] + (\mathbb{E}[\boldsymbol{c}_x(i)])^2 = M\boldsymbol{p}_x(i)(1 - \boldsymbol{p}_x(i)) + M^2\boldsymbol{p}_x^2(i) = M[\boldsymbol{p}_x(i) + (M-1)\boldsymbol{p}_x^2(i)]$. Therefore, we have

$$A_1 = \frac{1}{M^4} \sum_{i=1}^{n} \mathbb{E}\left[\boldsymbol{c}_x^2(i)\right] \cdot \mathbb{E}\left[\boldsymbol{c}_y^2(i)\right]$$

$$= \frac{1}{M^4} \sum_{i=1}^{n} M\left[\boldsymbol{p}_x(i) + (M-1)\boldsymbol{p}_x^2(i)\right] \cdot M\left[\boldsymbol{p}_y(i) + (M-1)\boldsymbol{p}_y^2(i)\right]$$

$$= \frac{1}{M^2} \sum_{i=1}^{n} \boldsymbol{p}_x(i)\boldsymbol{p}_y(i) + (M-1)\left(\boldsymbol{p}_x\boldsymbol{p}_y^2(i) + \boldsymbol{p}_x^2(i)\boldsymbol{p}_y(i)\right) + (M-1)^2\boldsymbol{p}_x^2(i)\boldsymbol{p}_y^2(i)$$

$$= \frac{1}{M^2}\langle\boldsymbol{p}_x, \boldsymbol{p}_y\rangle + \frac{M-1}{M^2}\left(\langle\boldsymbol{p}_x, \boldsymbol{p}_y^2\rangle + \langle\boldsymbol{p}_x^2, \boldsymbol{p}_y\rangle\right) + \frac{(M-1)^2}{M^2}\langle\boldsymbol{p}_x^2, \boldsymbol{p}_y^2\rangle,$$

where with a slight abuse of notation, we use $\langle p_x, p_y^2\rangle$ to denote $\sum_{i=1}^{n} p_x(i)p_y^2(i)$, and we use $\langle p_x^2, p_y^2\rangle$ to denote $\sum_{i=1}^{n} p_x^2(i)p_y^2(i)$.

To calculate $A_2$, we need to calculate $\mathbb{E}[\boldsymbol{c}_x(i)\boldsymbol{c}_x(j)]$ where $i \neq j$. We define $X_a^i$ as follows:

$$X_a^i = \begin{cases} 1, & \text{The } a\text{-th random walk from } x \text{ ends at } i \\ 0, & \text{otherwise} \end{cases}.$$

So we have $\mathbb{E}[\boldsymbol{c}_x(i)\boldsymbol{c}_x(j)] = \mathbb{E}\left[\sum_{a=1}^{M} X_a^i \sum_{a=1}^{M} X_a^j\right] = \sum_{a=1}^{M} \sum_{b=1}^{M} \mathbb{E}[X_a^i X_b^j]$. For all $a = b$ and $i \neq j$, we have $\mathbb{E}[X_a^i X_b^j = 0]$, since for a single random walk, it cannot ends at $i$ and $j$ the same time. For all $a \neq b$ and $i \neq j$, we have $\mathbb{E}[X_a^i X_b^j] = \boldsymbol{p}_x(i)\boldsymbol{p}_x(j)$. So we can get $\mathbb{E}[\boldsymbol{c}_x(i)\boldsymbol{c}_x(j)] = M(M-1)\boldsymbol{p}_x(i)\boldsymbol{p}_x(j)$. By the same augment, we get that for all $i \neq j$, $\mathbb{E}[\boldsymbol{c}_y(i)\boldsymbol{c}_y(j)] = M(M-1)\boldsymbol{p}_y(i)\boldsymbol{p}_y(j)$. Therefore,

$$A_2 = \frac{1}{M^4} \sum_{i=1}^{n} \sum_{j=1,j \neq i}^{n} \mathbb{E}\left[\boldsymbol{c}_x(i)\boldsymbol{c}_x(j)\right] \cdot \mathbb{E}\left[\boldsymbol{c}_y(i)\boldsymbol{c}_y(j)\right]$$

$$= \frac{1}{M^4} \sum_{i=1}^{n} \sum_{j=1,j \neq i}^{n} M(M-1)\boldsymbol{p}_x(i)\boldsymbol{p}_x(j) \cdot M(M-1)\boldsymbol{p}_y(i)\boldsymbol{p}_y(j)$$

$$= \frac{(M-1)^2}{M^2} \sum_{i=1}^{m} \sum_{j=1,j \neq i}^{n} \boldsymbol{p}_x(i)\boldsymbol{p}_y(i) \cdot \boldsymbol{p}_x(j)\boldsymbol{p}_y(j)$$

$$= \frac{(M-1)^2}{M^2} \left(\sum_{i=1}^{n} \sum_{j=1}^{n} \boldsymbol{p}_x(i)\boldsymbol{p}_y(i) \cdot \boldsymbol{p}_x(j)\boldsymbol{p}_y(j) - \sum_{i=1}^{n} \boldsymbol{p}_x^2(i)\boldsymbol{p}_y^2(i)\right)$$

$$= \frac{(M-1)^2}{M^2} \left(\sum_{i=1}^{n} \boldsymbol{p}_x(i)\boldsymbol{p}_y(i) \sum_{j=1}^{n} \boldsymbol{p}_x(j)\boldsymbol{p}_y(j) - \langle\boldsymbol{p}_x^2, \boldsymbol{p}_y^2\rangle\right)$$

$$= \frac{(M-1)^2}{M^2} \left( \langle \boldsymbol{p}_x, \boldsymbol{p}_y \rangle^2 - \langle \boldsymbol{p}_x^2, \boldsymbol{p}_y^2 \rangle \right).$$

Put them together, we get

$$\mathbb{E}[Z_b^2] = A_1 + A_2$$

$$= \frac{1}{M^2} \langle \boldsymbol{p}_x, \boldsymbol{p}_y \rangle + \frac{M-1}{M^2} \left( \langle \boldsymbol{p}_x, \boldsymbol{p}_y^2 \rangle + \langle \boldsymbol{p}_x^2, \boldsymbol{p}_y \rangle \right) + \frac{(M-1)^2}{M^2} \langle \boldsymbol{p}_x^2, \boldsymbol{p}_y^2 \rangle$$

$$+ \frac{(M-1)^2}{M^2} \left( \langle \boldsymbol{p}_x, \boldsymbol{p}_y \rangle^2 - \langle \boldsymbol{p}_x^2, \boldsymbol{p}_y^2 \rangle \right)$$

$$= \frac{1}{M^2} \langle \boldsymbol{p}_x, \boldsymbol{p}_y \rangle + \frac{M-1}{M^2} \left( \langle \boldsymbol{p}_x, \boldsymbol{p}_y^2 \rangle + \langle \boldsymbol{p}_x^2, \boldsymbol{p}_y \rangle \right) + \frac{(M-1)^2}{M^2} \langle \boldsymbol{p}_x, \boldsymbol{p}_y \rangle^2.$$

Therefore, we have

$$\mathrm{Var}[Z_b] = \mathbb{E}[Z_b^2] - (\mathbb{E}[Z_b])^2$$

$$= \frac{1}{M^2} \langle \boldsymbol{p}_x, \boldsymbol{p}_y \rangle + \frac{M-1}{M^2} \left( \langle \boldsymbol{p}_x, \boldsymbol{p}_y^2 \rangle + \langle \boldsymbol{p}_x^2, \boldsymbol{p}_y \rangle \right) + \frac{(M-1)^2}{M^2} \langle \boldsymbol{p}_x, \boldsymbol{p}_y \rangle^2 - \langle \boldsymbol{p}_x, \boldsymbol{p}_y \rangle^2$$

$$= \frac{1}{M^2} \langle \boldsymbol{p}_x, \boldsymbol{p}_y \rangle + \frac{M-1}{M^2} \left( \langle \boldsymbol{p}_x, \boldsymbol{p}_y^2 \rangle + \langle \boldsymbol{p}_x^2, \boldsymbol{p}_y \rangle \right) + \frac{1-2M}{M^2} \langle \boldsymbol{p}_x, \boldsymbol{p}_y \rangle^2$$

$$\leq \frac{1}{M^2} \langle \boldsymbol{p}_x, \boldsymbol{p}_y \rangle + \frac{1}{M} \left( \langle \boldsymbol{p}_x, \boldsymbol{p}_y^2 \rangle + \langle \boldsymbol{p}_x^2, \boldsymbol{p}_y \rangle \right)$$

$$= \frac{1}{M^2} \sum_{i=1}^{n} \boldsymbol{p}_x(i) \boldsymbol{p}_y(i) + \frac{1}{M} \left( \sum_{i=1}^{n} \boldsymbol{p}_x(i) \boldsymbol{p}_y^2(i) + \sum_{i=1}^{n} \boldsymbol{p}_x^2(i) \boldsymbol{p}_y(i) \right)$$

$$\leq \frac{1}{M^2} \|\boldsymbol{p}_x\|_2 \cdot \|\boldsymbol{p}_y\|_2 + \frac{1}{M} \left( \|\boldsymbol{p}_x\|_2 \cdot \|\boldsymbol{p}_y\|_4^2 + \|\boldsymbol{p}_x\|_4^2 \cdot \|\boldsymbol{p}_y\|_2 \right)$$

$$\leq \frac{1}{M^2} \|\boldsymbol{p}_x\|_2 \cdot \|\boldsymbol{p}_y\|_2 + \frac{1}{M} \left( \|\boldsymbol{p}_x\|_2 \cdot \|\boldsymbol{p}_y\|_2^2 + \|\boldsymbol{p}_x\|_2^2 \cdot \|\boldsymbol{p}_y\|_2 \right),$$

where the second-to-last inequality uses the Cauchy–Schwarz inequality and the last one follows from Fact C.1. □

Building on Lemma E.1, we now consider the estimator $Z$ obtained by averaging $B = R/M$ independent copies of $Z_b$. The following lemma shows that $Z$ remains an unbiased estimator with variance reduced by a factor of $B = R/M$.

**Lemma E.2.** *Let $G = (V, E)$ be a graph. Let $R, t, M$ be integers, where $1 \leq M \leq R$. Let $x, y \in V$ be two vertices. Let $\boldsymbol{M}$ be the random walk transition matrix of $G$. Let $Z$ be the output of* ESTRWDOT$(G, R, t, M, x, y)$ *(Alg. 1). Then, we have*

$$\mathbb{E}[Z] = \langle \boldsymbol{M}^t \mathbb{1}_x, \boldsymbol{M}^t \mathbb{1}_y \rangle,$$

$$\mathrm{Var}[Z] \leq \frac{1}{R} \left[ \frac{1}{M} \|\boldsymbol{M}^t \mathbb{1}_x\|_2 \cdot \|\boldsymbol{M}^t \mathbb{1}_y\|_2 + \left( \|\boldsymbol{M}^t \mathbb{1}_x\|_2 \cdot \|\boldsymbol{M}^t \mathbb{1}_y\|_2^2 + \|\boldsymbol{M}^t \mathbb{1}_x\|_2^2 \cdot \|\boldsymbol{M}^t \mathbb{1}_y\|_2 \right) \right].$$

*Proof.* According to Alg. 1, we know that $Z = \frac{1}{B} \sum_{b=1}^{B} Z_b$, where $B = \frac{R}{M}$. Therefore, using Lemma E.1, we have $\mathbb{E}[Z] = \frac{1}{B} \sum_{b=1}^{B} \mathbb{E}[Z_b] = \langle \boldsymbol{M}^t \mathbb{1}_x, \boldsymbol{M}^t \mathbb{1}_y \rangle$ and

$$\mathrm{Var}[Z] = \frac{1}{B^2} \sum_{b=1}^{B} \mathrm{Var}[Z_b]$$

$$= \frac{1}{B} \mathrm{Var}[Z_b]$$

$$= \frac{M}{R} \mathrm{Var}[Z_b]$$

$$\leq \frac{M}{R}\left[\frac{1}{M^2}\|\boldsymbol{M}^t\mathbb{1}_x\|_2 \cdot \|\boldsymbol{M}^t\mathbb{1}_y\|_2 + \frac{1}{M}\left(\|\boldsymbol{M}^t\mathbb{1}_x\|_2 \cdot \|\boldsymbol{M}^t\mathbb{1}_y\|_2^2 + \|\boldsymbol{M}^t\mathbb{1}_x\|_2^2 \cdot \|\boldsymbol{M}^t\mathbb{1}_y\|_2\right)\right]$$

$$= \frac{1}{R}\left[\frac{1}{M}\|\boldsymbol{M}^t\mathbb{1}_x\|_2 \cdot \|\boldsymbol{M}^t\mathbb{1}_y\|_2 + \left(\|\boldsymbol{M}^t\mathbb{1}_x\|_2 \cdot \|\boldsymbol{M}^t\mathbb{1}_y\|_2^2 + \|\boldsymbol{M}^t\mathbb{1}_x\|_2^2 \cdot \|\boldsymbol{M}^t\mathbb{1}_y\|_2\right)\right].$$

$\square$

Lemma 3.1 shows that, with suitable input parameters, EstRWDot$(G, R, t, M, x, y)$ (Alg. 1) approximates the dot product of the random walk distributions from any two vertices $x, y \in V$ within an error of $\sigma_{\text{err}}$.

**Lemma E.3** (Restatement of Lemma 3.1). *Let $k \geq 2$ be an integer and $\varphi, \varepsilon \in (0, 1)$. Let $G = (V, E)$ be a $d$-regular and $(k, \varphi, \varepsilon)$-clusterable graph. Let $\boldsymbol{M}$ be the random walk transition matrix of $G$. Let $Z$ be the output of EstRWDot$(G, R, t, M, x, y)$ (Alg. 1). Let $\sigma_{\text{err}} > 0$. Let $c > 1$ be a large enough constant. For any $t \geq \frac{20 \log n}{\varphi^2}$ and any $x, y \in V$, if $R \geq \frac{c \cdot k^2 n^{-1+40\varepsilon/\varphi^2}}{\sigma_{\text{err}}^2 M}$ and $1 \leq M \leq O(\frac{n^{1/2-20\varepsilon/\varphi^2}}{k})$, then with probability at least $0.99$, we have*

$$|Z - \langle \boldsymbol{M}^t\mathbb{1}_x, \boldsymbol{M}^t\mathbb{1}_y\rangle| \leq \sigma_{\text{err}}.$$

*Moreover, EstRWDot$(G, R, t, M, x, y)$ runs in $O(Rt)$ time and uses $O(M \cdot \log n)$ bits of space.*

**Remark E.1.** *The success probability of Lemma 3.1 can be boosted up to $1 - n^{-100}$ using median trick, i.e., by taking the median of $O(\log n)$ independent runs.*

To prove Lemma 3.1, we need the following lemma in Gluch et al. (2021).

**Lemma E.4** (Lemma 22 in Gluch et al. (2021)). *Let $k \geq 2$ be an integer and $\varphi, \varepsilon \in (0, 1)$. Let $G = (V, E)$ be a $d$-regular and $(k, \varphi, \varepsilon)$-clusterable graph. Let $\boldsymbol{M}$ be the random walk transition matrix of $G$. For any $t \geq \frac{20 \log n}{\varphi^2}$ and any $x \in V$ we have*

$$\|\boldsymbol{M}^t\mathbb{1}_x\|_2 \leq O(k \cdot n^{-1/2+(20\varepsilon/\varphi^2)}).$$

Now we are ready to prove Lemma 3.1.

*Proof of Lemma 3.1.* **Correctness.** By Lemma E.2 and Lemma E.4, we can get that

$$\text{Var}[Z] \leq \frac{1}{R}\left[\frac{1}{M}\|\boldsymbol{M}^t\mathbb{1}_x\|_2 \cdot \|\boldsymbol{M}^t\mathbb{1}_y\|_2 + \left(\|\boldsymbol{M}^t\mathbb{1}_x\|_2 \cdot \|\boldsymbol{M}^t\mathbb{1}_y\|_2^2 + \|\boldsymbol{M}^t\mathbb{1}_x\|_2^2 \cdot \|\boldsymbol{M}^t\mathbb{1}_y\|_2\right)\right]$$

$$= \frac{1}{R}\left(\frac{O(k^2 \cdot n^{-1+40\varepsilon/\varphi^2})}{M} + O(k^3 \cdot n^{-3/2+60\varepsilon/\varphi^2})\right).$$

Using Chebyshev's inequality, we have

$$\begin{aligned} \Pr[|Z - \langle \boldsymbol{M}^t\mathbb{1}_x, \boldsymbol{M}^t\mathbb{1}_y\rangle| \geq \sigma_{\text{err}}] &= \Pr[|Z - \mathbb{E}[Z]| \geq \sigma_{\text{err}}] \\ &\leq \frac{\text{Var}[Z]}{\sigma_{\text{err}}^2} \\ &\leq \frac{1}{\sigma_{\text{err}}^2} \cdot \frac{1}{R}\left(\frac{O(k^2 \cdot n^{-1+40\varepsilon/\varphi^2})}{M} + O(k^3 \cdot n^{-3/2+60\varepsilon/\varphi^2})\right) \\ &\leq \frac{1}{\sigma_{\text{err}}^2} \cdot \frac{1}{R} \cdot O\left(\frac{k^2 \cdot n^{-1+40\varepsilon/\varphi^2}}{M}\right) \\ &\leq \frac{1}{100}, \end{aligned}$$

where the second-to-last inequality holds by $M \leq O\left(\frac{n^{1/2-20\varepsilon/\varphi^2}}{k}\right)$. And the last inequality holds by our choice of

$$R \geq \frac{c \cdot k^2 n^{-1+40\varepsilon/\varphi^2}}{\sigma_{\text{err}}^2 M},$$

where $c$ is a large enough constant that cancels the constant hidden in $O\left(\frac{k^2 \cdot n^{-1+40\varepsilon/\varphi^2}}{M}\right)$.

**Runtime and space.** Algorithm EstRWDot$(G, R, t, M, x, y)$ (Alg. 1) performs $B = \frac{R}{M}$ batches (i.e., $B = \frac{R}{M}$ iterations of the for-loop). In each batch, it runs $M$ random walks of length $t$, which requires $O(Mt)$ time and $O(M)$ words of space to store the $O(M)$ endpoints of the walks. Computing the dot product of two probability distributions takes $O(M)$ time, since each distribution has at most $M$ nonzero entries. Therefore, the runtime and space per batch are $O(Mt+M) = O(Mt)$ time and $O(M)$ words, respectively. Moreover, the space used within each batch can be reused across batches. Consequently, the overall runtime and space complexity of EstRWDot$(G, R, t, M, x, y)$ (Alg. 1) are $B \cdot O(Mt) = \frac{R}{M} \cdot O(Mt) = O(Rt)$ and $O(M)$ words (i.e., $O(M \cdot \log n)$ bits of space, since each endpoint can be stored in $\log n$ bits), respectively. $\qquad\square$

Lemma E.5 states that, under appropriate input parameters, the output $\mathcal{G}$ of our algorithm EstColliProb $(G, R, t, M, I_S)$ (Alg. 2) is close to $(M^t S)^T (M^t S)$ in spectral norm, where $(M^t S)^T (M^t S)$ is the Gram matrix of the random walk distributions from vertices in the sample set.

**Lemma E.5.** *Let $k \geq 2$ be an integer and $\varphi, \varepsilon \in (0, 1)$. Let $G = (V, E)$ be a d-regular and $(k, \varphi, \varepsilon)$-clusterable graph. Let $M$ be the random walk transition matrix of $G$. Let $I_S = \{s_1, \ldots, s_s\}$ be a multiset of s indices chosen from $\{1, \ldots, n\}$. Let $S \in \mathbb{R}^{n \times s}$ be the matrix whose i-th column equals $\mathbb{1}_{s_i}$. Let $\mathcal{G} \in \mathbb{R}^{s \times s}$ be the output of EstColliProb $(G, R, t, M, I_S)$ (Alg. 2). Let $\sigma_{\text{err}} > 0$. Let $c > 1$ be a large enough constant. For any $t \geq \frac{20 \log n}{\varphi^2}$, if $R \geq \frac{c \cdot k^2 n^{-1+40\varepsilon/\varphi^2}}{\sigma_{\text{err}}^2 M}$ and $1 \leq M \leq O\left(\frac{n^{1/2-20\varepsilon/\varphi^2}}{k}\right)$, then with probability at least $1 - n^{-100}$, we have*

$$\|\mathcal{G} - (M^t S)^T (M^t S)\|_2 \leq s \cdot \sigma_{\text{err}}.$$

*Moreover, EstColliProb $(G, R, t, M, I_S)$ runs in $O(Rt \cdot \log n \cdot s^2)$ time and uses $O(M \cdot \log^2 n \cdot s^2)$ bits of space.*

*Proof.* **Correctness.** Note that in line 5 of Alg. 2, we get $\mathcal{G}_l(i, j) :=$ EstRWDot$(G, R, t, M, s_i, s_j)$ (Alg. 1). Since $t \geq \frac{20 \log n}{\varphi^2}$, $R \geq \frac{c \cdot k^2 n^{-1+40\varepsilon/\varphi^2}}{\sigma_{\text{err}}^2 M}$ and $1 \leq M \leq O\left(\frac{n^{1/2-20\varepsilon/\varphi^2}}{k}\right)$, then by Lemma 3.1, with probability at least 0.99, for all $i, j \in [s]$, we have

$$|\mathcal{G}_l(i, j) - \langle M^t \mathbb{1}_{s_i}, M^t \mathbb{1}_{s_j}\rangle| = |\mathcal{G}_l(i, j) - (M^t \mathbb{1}_{s_i})^T (M^t \mathbb{1}_{s_j})| \leq \sigma_{\text{err}}.$$

Note that in line 6 of Alg. 2, we define $\mathcal{G}$ as a matrix obtained by taking the entrywises median of $\mathcal{G}_l$'s over $O(\log n)$ runs. Thus with probability at least $1 - n^{-100}$ (see Remark E.1), for all $i, j \in [s]$, we have

$$|\mathcal{G}(i, j) - (M^t \mathbb{1}_{s_i})^T (M^t \mathbb{1}_{s_j})| \leq \sigma_{\text{err}},$$

which implies

$$\|\mathcal{G} - (M^t S)^T (M^t S)\|_F \leq s \cdot \sigma_{\text{err}}.$$

Moreover, we have

$$\|\mathcal{G} - (M^t S)^T (M^t S)\|_2 \leq \|\mathcal{G} - (M^t S)^T (M^t S)\|_F \leq s \cdot \sigma_{\text{err}}.$$

**Runtime and space.** In Alg. 2, Alg. 1 is called $\log n \cdot s^2$ times. Since the runtime and space of Alg. 1 are $O(Rt)$ and $O(M \log n)$ bits, respectively, the runtime and space of Alg. 2 are $O(Rt \cdot \log n \cdot s^2)$ and $O(M \cdot \log^2 n \cdot s^2)$ bits, respectively. $\qquad\square$

Recall that we use $(\boldsymbol{M}^t \mathbb{1}_x)^T (\boldsymbol{M}^t \boldsymbol{S})(\frac{n}{s} \cdot \widetilde{W}_{[k]} \widetilde{\Sigma}_{[k]}^{-4} \widetilde{W}_{[k]}^T)(\boldsymbol{M}^t \boldsymbol{S})^T (\boldsymbol{M}^t \mathbb{1}_y)$ to estimate $\langle \boldsymbol{f}_x, \boldsymbol{f}_y \rangle$. Lemma E.6 states that under appropriate parameters, Alg. 3 outputs a matrix $\Psi = \frac{n}{s} \cdot \widehat{W}_{[k]} \widehat{\Sigma}_{[k]}^{-2} \widehat{W}_{[k]}^T$ which, with high probability, is spectrally close to $\frac{n}{s} \cdot \widetilde{W}_{[k]} \widetilde{\Sigma}_{[k]}^{-4} \widetilde{W}_{[k]}^T$. The proof of Lemma E.6 is analogous to that of Lemma 24 in Gluch et al. (2021). Nevertheless, for completeness, we provide a concise proof here.

**Lemma E.6.** *Let $k \geq 2$ be an integer and $\varphi, \varepsilon \in (0, 1)$. Let $G = (V, E)$ be a $d$-regular and $(k, \varphi, \varepsilon)$-clusterable graph. Let $\boldsymbol{M}$ be the random walk transition matrix of $G$. Let $I_S = \{s_1, \ldots, s_s\}$ be a multiset of $s$ indices chosen independently and uniformly at random form $V = \{1, \ldots, n\}$. Let $\boldsymbol{S} \in \mathbb{R}^{n \times s}$ be the matrix whose $i$-th column equals $\mathbb{1}_{s_i}$. Let $\mathcal{G} \in \mathbb{R}^{s \times s}$ be the output of ESTCOLLIPROB $(G, R, t, M, I_S)$ (Alg. 2). Let $\sqrt{\frac{n}{s}} \cdot \boldsymbol{M}^t \boldsymbol{S} = \widetilde{U} \widetilde{\Sigma} \widetilde{W}^T$ be an SVD of $\sqrt{\frac{n}{s}} \cdot \boldsymbol{M}^t \boldsymbol{S}$ where $\widetilde{U} \in \mathbb{R}^{n \times n}, \widetilde{\Sigma} \in \mathbb{R}^{n \times s}, \widetilde{W} \in \mathbb{R}^{s \times n}$. Let $\frac{n}{s} \cdot \mathcal{G} = \widehat{W} \widehat{\Sigma} \widehat{W}^T$ be an eigendecomposition of $\frac{n}{s} \cdot \mathcal{G}$. Let $\frac{1}{n^8} < \xi < 1$. Let $c_1 > 1$ and $c_2 > 1$ be two large enough constants. For any $t \geq \frac{20 \log n}{\varphi^2}$, if $\frac{\varepsilon}{\varphi^2} \leq \frac{1}{10^5}$, $s \geq c_1 \cdot n^{240\varepsilon/\varphi^2} \cdot \log n \cdot k^4$, $R \geq \frac{c_2 \cdot k^6 \cdot n^{1+760\varepsilon/\varphi^2}}{M \cdot \xi^2}$ and $1 \leq M \leq O\left(\frac{n^{1/2-20\varepsilon/\varphi^2}}{k}\right)$, then with probability at least $1 - 2 \cdot n^{-100}$, matrices $\widehat{\Sigma}_{[k]}^{-2}$ and $\widetilde{\Sigma}_{[k]}^{-4}$ exist and we have*

$$\|\widetilde{W}_{[k]} \widetilde{\Sigma}_{[k]}^{-4} \widetilde{W}_{[k]}^T - \widehat{W}_{[k]} \widehat{\Sigma}_{[k]}^{-2} \widehat{W}_{[k]}^T\|_2 < \xi.$$

Equipped with Lemma E.5, to prove Lemma E.6, we also need the following lemmas.

**Lemma E.7** (Lemma 18 in Gluch et al. (2021)). *Let $\widetilde{A}, \widehat{A} \in \mathbb{R}^{n \times n}$ be symmetric matrices with eigendecomposition $\widetilde{A} = \widetilde{Y} \widetilde{\Gamma} \widetilde{Y}^T$ and $\widehat{A} = \widehat{Y} \widehat{\Gamma} \widehat{Y}^T$. Let the eigenvalues of $\widetilde{A}$ be $1 \geq \gamma_1 \geq \cdots \geq \gamma_n \geq 0$. Suppose that $\|\widetilde{A} - \widehat{A}\|_2 \leq \frac{\gamma_k}{100}$ and $\gamma_{k+1} < \frac{\gamma_k}{4}$. Then we have*

$$\|\widetilde{Y}_{[k]} \widetilde{\Gamma}_{[k]}^{-1} \widetilde{Y}_{[k]}^T - \widehat{Y}_{[k]} \widehat{\Gamma}_{[k]}^{-1} \widehat{Y}_{[k]}^T\|_2 \leq \frac{16\|\widetilde{A} - \widehat{A}\|_2 + 4\gamma_{k+1}}{\gamma_k^2}.$$

**Lemma E.8** (Lemma 28 in Gluch et al. (2021)). *Let $k \geq 2$ be an integer and $\varphi, \varepsilon \in (0, 1)$. Let $G = (V, E)$ be a $d$-regular and $(k, \varphi, \varepsilon)$-clusterable graph. Let $\boldsymbol{M}$ be the random walk transition matrix of $G$. Let $I_S = \{s_1, \ldots, s_s\}$ be a multiset of $s$ indices chosen independently and uniformly at random form $V = \{1, \ldots, n\}$. Let $\boldsymbol{S} \in \mathbb{R}^{n \times s}$ be the matrix whose $i$-th column equals $\mathbb{1}_{s_i}$. Let $c > 1$ be a large enough constant. For any $t \geq \frac{20 \log n}{\varphi^2}$, if $\frac{\varepsilon}{\varphi^2} \leq \frac{1}{10^5}$ and $s \geq c \cdot n^{240\varepsilon/\varphi^2} \cdot \log n \cdot k^4$, then with probability at least $1 - n^{-100}$, we have*

- $v_k \left(\frac{n}{s} \cdot (\boldsymbol{M}^t \boldsymbol{S})(\boldsymbol{M}^t \boldsymbol{S})^T\right) = v_k \left(\frac{n}{s} \cdot (\boldsymbol{M}^t \boldsymbol{S})^T (\boldsymbol{M}^t \boldsymbol{S})\right) \geq \frac{n^{-80\varepsilon/\varphi^2}}{2}$,
- $v_{k+1} \left(\frac{n}{s} \cdot (\boldsymbol{M}^t \boldsymbol{S})(\boldsymbol{M}^t \boldsymbol{S})^T\right) \leq n^{-9}$.

**Lemma E.9** (Weyl's Inequality). *Let $A, B \in \mathbb{R}^{n \times n}$ be symmetric matrices. Let $\alpha_1, \ldots, \alpha_n$ and $\beta_1, \ldots, \beta_n$ be the eigenvalues of $A$ and $B$ respectively. Then for any $i \in [n]$, we have*

$$|\alpha_i - \beta_i| \leq \|A - B\|_2.$$

Now we are ready to prove Lemma E.6.

*Proof of Lemma E.6.* Let $c_3 > 1$ be a large enough constant and let $\sigma_{\text{err}} = \frac{\xi \cdot n^{-1-360\varepsilon/\varphi^2}}{c_3 \cdot k^2}$. Let $c$ be a constant from Lemma E.5. By the assumption of the lemma for a large enough constant $c_2 > 1$, we have

$$R \geq \frac{c_2 \cdot k^6 \cdot n^{1+760\varepsilon/\varphi^2}}{M \cdot \xi^2} \geq \frac{c \cdot k^2 n^{-1+40\varepsilon/\varphi^2}}{\sigma_{\text{err}}^2 M}.$$

Thus we can apply Lemma E.5. Hence, with probability at least $1 - n^{-100}$, we have

$$\|\mathcal{G} - (\boldsymbol{M}^t \boldsymbol{S})^T (\boldsymbol{M}^t \boldsymbol{S})\|_2 \leq s \cdot \sigma_{\text{err}}.$$

Let $\widetilde{A} = \frac{n}{s} \cdot (\boldsymbol{M}^t \boldsymbol{S})^T (\boldsymbol{M}^t \boldsymbol{S}) = \widetilde{W} \widetilde{\Sigma}^2 \widetilde{W}^T$ and $\widehat{A} = \frac{n}{s} \cdot \mathcal{G}$. Thus, we have $\widetilde{A}^2 = \left(\frac{n}{s} \cdot (\boldsymbol{M}^t \boldsymbol{S})^T (\boldsymbol{M}^t \boldsymbol{S})\right)^2 = \widetilde{W} \widetilde{\Sigma}^4 \widetilde{W}^T$ and $\widehat{A}^2 = \left(\frac{n}{s} \cdot \mathcal{G}\right)^2 = \widehat{W} \widehat{\Sigma}^2 \widehat{W}^T$. To use Lemma E.7, we

have to bound $\|\widetilde{A}^2 - \widehat{A}^2\|_2 = \left(\frac{n}{s}\right)^2 \|((\boldsymbol{M}^t\boldsymbol{S})^T(\boldsymbol{M}^t\boldsymbol{S}))^2 - \mathcal{G}^2\|_2$. Using the triangle inequality and sub-multiplicativity of spectral norm and the above $\|\mathcal{G} - (\boldsymbol{M}^t S)^T(\boldsymbol{M}^t S)\|_2 \leq s \cdot \sigma_{\mathrm{err}}$ bound, we can get that

$$\|((\boldsymbol{M}^t\boldsymbol{S})^T(\boldsymbol{M}^t\boldsymbol{S}))^2 - \mathcal{G}^2\|_2 \leq (s \cdot \sigma_{\mathrm{err}})^2 + 2 \cdot s \cdot \sigma_{\mathrm{err}}\|(\boldsymbol{M}^t\boldsymbol{S})^T(\boldsymbol{M}^t\boldsymbol{S})\|_2.$$

Note that $\|(\boldsymbol{M}^t\boldsymbol{S})^T(\boldsymbol{M}^t\boldsymbol{S})\|_2 \leq \|(\boldsymbol{M}^t\boldsymbol{S})^T(\boldsymbol{M}^t\boldsymbol{S})\|_F = \sqrt{\sum_{i=1}^s \sum_{j=1}^s ((\boldsymbol{M}^t\mathbb{1}_{s_i})^T(\boldsymbol{M}^t\mathbb{1}_{s_j}))^2}$, by Cauchy Schwarz inequality and Lemma E.4, we can get that $\|(\boldsymbol{M}^t\boldsymbol{S})^T(\boldsymbol{M}^t\boldsymbol{S})\|_2 \leq O(s \cdot k^2 \cdot n^{-1+40\varepsilon/\varphi^2})$. Put them together and by the choice of $\sigma_{\mathrm{err}} = \frac{\xi \cdot n^{-1-360\varepsilon/\varphi^2}}{c_3 \cdot k^2}$, we have that

$$\|\widetilde{A}^2 - \widehat{A}^2\|_2 \leq O\left(\frac{\xi \cdot n^{-320\varepsilon/\varphi^2}}{c_3}\right).$$

Moreover, let $c_1$ be the constant from Lemma E.8, since $s \geq c_1 \cdot n^{240\varepsilon/\varphi^2} \cdot \log n \cdot k^4$, by Lemma E.8, with probability at least $1 - n^{-100}$, we have

$$v_k\left(\widetilde{A}^2\right) = v_k\left(\left(\frac{n}{s} \cdot (\boldsymbol{M}^t\boldsymbol{S})^T(\boldsymbol{M}^t\boldsymbol{S})\right)^2\right) \geq \left(\frac{n^{-80\varepsilon/\varphi^2}}{2}\right)^2 = \frac{n^{-160\varepsilon/\varphi^2}}{4},$$

and

$$v_{k+1}\left(\widetilde{A}^2\right) = v_{k+1}\left(\left(\frac{n}{s} \cdot (\boldsymbol{M}^t\boldsymbol{S})^T(\boldsymbol{M}^t\boldsymbol{S})\right)^2\right) \leq (n^{-9})^2 = n^{-18}.$$

By Weyl's inequality, we have that $v_k(\widehat{A}^2) \geq v_k(\widetilde{A}^2) - \|\widetilde{A}^2 - \widehat{A}^2\|_2 \geq \frac{n^{-160\varepsilon/\varphi^2}}{4} - O(\frac{\xi \cdot n^{-320\varepsilon/\varphi^2}}{c_3}) > 0$, so $\widehat{\Sigma}_{[k]}^{-2}$ exists. Moreover, since $\widetilde{A}^2, \widehat{A}^2$ are symmetric matrices, $\|\widetilde{A}^2 - \widehat{A}^2\|_2 \leq \frac{v_k(\widetilde{A}^2)}{100}$ and $v_{k+1}(\widetilde{A}^2) < \frac{v_k(\widetilde{A}^2)}{4}$, by Lemma E.7, we have that

$$\begin{aligned}
\|\widetilde{W}_{[k]}\widetilde{\Sigma}_{[k]}^{-4}\widetilde{W}_{[k]}^T - \widehat{W}_{[k]}\widehat{\Sigma}_{[k]}^{-2}\widehat{W}_{[k]}^T\|_2 &\leq \frac{16\|\widetilde{A}^2 - \widehat{A}^2\|_2 + 4v_{k+1}(\widetilde{A}^2)}{v_k(\widetilde{A}^2)^2} \\
&\leq \frac{O\left(\frac{\xi \cdot n^{-320\varepsilon/\varphi^2}}{c_3}\right) + 4n^{-18}}{\frac{n^{-320\varepsilon/\varphi^2}}{16}} \\
&\leq O\left(\frac{\xi}{c_3}\right) + 64n^{-17} \\
&\leq \xi. \qquad\qquad\qquad \frac{1}{n^8} \leq \xi
\end{aligned}$$

Moreover, both Lemma E.5 and Lemma E.8 fail with probability at most $n^{-100}$, by union bound, we can get that the above inequality holds with probability at least $1 - 2n^{-100}$. $\qquad\square$

The following lemma shows that the output value $\langle \boldsymbol{f}_x, \boldsymbol{f}_y \rangle_{\mathrm{apx}}$ of Alg. 4 is close to $(\boldsymbol{M}^t\mathbb{1}_x)^T(\boldsymbol{M}^t\boldsymbol{S})\left(\frac{n}{s} \cdot \widetilde{W}_{[k]}\widetilde{\Sigma}_{[k]}^{-4}\widetilde{W}_{[k]}^T\right)(\boldsymbol{M}^t\boldsymbol{S})^T(\boldsymbol{M}^t\mathbb{1}_y)$. The proof follows from the proof of Lemma 29 in Gluch et al. (2021). Nevertheless, for completeness, we provide a concise proof here.

**Lemma E.10.** *Let $k \geq 2$ be an integer and $\varphi, \varepsilon \in (0,1)$. Let $G = (V, E)$ be a d-regular and $(k, \varphi, \varepsilon)$-clusterable graph. Let $\boldsymbol{M}$ be the random walk transition matrix of $G$. Let $I_S = \{s_1, \ldots, s_s\}$ be a multiset of s indices chosen independently and uniformly at random form $V = \{1, \ldots, n\}$. Let $\boldsymbol{S} \in \mathbb{R}^{n \times s}$ be the matrix whose i-th column equals $\mathbb{1}_{s_i}$. Let $\sqrt{\frac{n}{s}} \cdot \boldsymbol{M}^t\boldsymbol{S} = \widetilde{U}\widetilde{\Sigma}\widetilde{W}^T$ be an SVD of $\sqrt{\frac{n}{s}} \cdot \boldsymbol{M}^t\boldsymbol{S}$ where $\widetilde{U} \in \mathbb{R}^{n \times n}, \widetilde{\Sigma} \in \mathbb{R}^{n \times n}, \widetilde{W} \in \mathbb{R}^{s \times n}$. Let $\frac{1}{n^6} < \xi < 1$ and $1 \leq M_{\mathrm{init}} \leq O\left(\frac{n^{1/2-20\varepsilon/\varphi^2}}{k}\right)$. Let $t \geq \frac{20\log n}{\varphi^2}$. Let $c > 1$ be a large enough constant. Let $s \geq$*

$c \cdot n^{240\varepsilon/\varphi^2} \cdot \log n \cdot k^4$. *Let $\Psi$ denote the matrix constructed by* INITORACLE $(G, k, \xi, M_{\text{init}})$ *(Alg. 3).*

*Let $x, y \in V$. Let $\langle \boldsymbol{f}_x, \boldsymbol{f}_y \rangle_{\text{apx}} \in \mathbb{R}$ denote the value returned by* QUERYDOT $(G, x, y, \xi, \Psi, M_{\text{query}})$ *(Alg. 4). If $\frac{\varepsilon}{\varphi^2} \leq \frac{1}{10^5}$, Alg. 3 succeeds and $1 \leq M_{\text{query}} \leq O\left(\frac{n^{1/2 - 20\varepsilon/\varphi^2}}{k}\right)$, then with probability at least $1 - 5n^{-100}$ matrix $\widetilde{\Sigma}_{[k]}^{-4}$ exists and we have*

$$\left| \langle \boldsymbol{f}_x, \boldsymbol{f}_y \rangle_{\text{apx}} - (\boldsymbol{M}^t \mathbb{1}_x)^T (\boldsymbol{M}^t \boldsymbol{S}) \left( \frac{n}{s} \cdot \widetilde{W}_{[k]} \widetilde{\Sigma}_{[k]}^{-4} \widetilde{W}_{[k]}^T \right) (\boldsymbol{M}^t \boldsymbol{S})^T (\boldsymbol{M}^t \mathbb{1}_y) \right| < \frac{\xi}{n}.$$

*Proof.* Note that in line 8 of Alg. 4, $\langle \boldsymbol{f}_x, \boldsymbol{f}_y \rangle_{\text{apx}}$ is defined as $\boldsymbol{\alpha}_x^T \Psi \boldsymbol{\alpha}_y$, where in line 8 of Alg. 3, $\Psi \in \mathbb{R}^{s \times s}$ is defined to be $\Psi = \frac{n}{s} \cdot \widehat{W}_{[k]} \widehat{\Sigma}_{[k]}^{-2} \widehat{W}_{[k]}^T$ and $\boldsymbol{\alpha}_x, \boldsymbol{\alpha}_y \in \mathbb{R}^s$ are vectors obtained by taking entriwise median over all $O(\log n)$ runs (see lines $3 \sim 7$ of Alg. 4).

For any vertex $x \in V$, we use $\boldsymbol{p}_x$ to denote $\boldsymbol{p}_x = \boldsymbol{M}^t \mathbb{1}_x$. We then define

$$\mathbf{a}_x = \boldsymbol{p}_x^T(\boldsymbol{M}^t \boldsymbol{S}), A = \frac{n}{s} \cdot \widetilde{W}_{[k]} \widetilde{\Sigma}_{[k]}^{-4} \widetilde{W}_{[k]}^T, \mathbf{a}_y = (\boldsymbol{M}^t \boldsymbol{S})^T \boldsymbol{p}_x,$$

$$\mathbf{e}_x = \boldsymbol{\alpha}_x^T - \mathbf{a}_x, \quad E = \Psi - A, \quad \mathbf{e}_y = \boldsymbol{\alpha}_y - \mathbf{a}_y.$$

Then by triangle inequality, we have

$$\left| \boldsymbol{\alpha}^T \Psi \boldsymbol{\alpha}_y - \boldsymbol{p}_x^T(\boldsymbol{M}^t \boldsymbol{S}) \left( \frac{n}{s} \cdot \widetilde{W}_{[k]} \widetilde{\Sigma}_{[k]}^{-4} \widetilde{W}_{[k]}^T \right) (\boldsymbol{M}^t \boldsymbol{S})^T \boldsymbol{p}_y \right|$$
$$= |(\mathbf{a}_x + \mathbf{e}_x)(A + E)(\mathbf{a}_y + \mathbf{e}_y) - \mathbf{a}_x A \mathbf{a}_y|$$
$$\leq \|\mathbf{e}_x\|_2 \|E\|_2 \|\mathbf{e}_y\|_2 + \|\mathbf{e}_x\|_2 \|A\|_2 \|\mathbf{e}_y\|_2 + \|\mathbf{a}_x\|_2 \|E\|_2 \|\mathbf{e}_y\|_2$$
$$+ \|\mathbf{a}_x\|_2 \|A\|_2 \|\mathbf{e}_y\|_2 + \|\mathbf{a}_x\|_2 \|E\|_2 \|\mathbf{a}_y\|_2 + \|\mathbf{e}_x\|_2 \|A\|_2 \|\mathbf{a}_y\|_2 + \|\mathbf{a}_x\|_2 \|E\|_2 \|\mathbf{a}_y\|_2.$$

In the following, we bound $\|\mathbf{a}_x\|_2, \|\mathbf{a}_y\|_2, \|E\|_2, \|A\|_2, \|\mathbf{e}_x\|_2$ and $\|\mathbf{e}_x\|_2$.

Let $c' > 1$ be a constant and let $\xi' = \frac{\xi}{c' \cdot k^4 \cdot n^{80\varepsilon/\varphi^2}}$. Thus for large enough constant $c$, we have $s \geq c_1 \cdot n^{240\varepsilon/\varphi^2} \cdot \log n \cdot k^4$ and $R_{\text{init}} = \Theta\left(\frac{n^{1+920\varepsilon/\varphi^2}}{M_{\text{init}}} \cdot \frac{k^{14}}{\xi^2}\right) \geq \frac{c_2 k^6 \cdot n^{1+760\varepsilon/\varphi^2}}{M_{\text{init}} \cdot \xi'^2}$ as in line 2 of Alg. 3, hence, by Lemma E.6 applied with $\xi'$ we have that with probability at least $1 - 2n^{-100}$, $\widehat{\Sigma}_{[k]}^{-2}$ and $\widetilde{\Sigma}_{[k]}^{-4}$ exist and we have

$$\|E\|_2 = \frac{n}{s} \cdot \|\widehat{W}_{[k]} \widehat{\Sigma}_{[k]}^{-2} \widehat{W}_{[k]}^T - \widetilde{W}_{[k]} \widetilde{\Sigma}_{[k]}^{-4} \widetilde{W}_{[k]}^T\|_2 < \frac{n}{s} \cdot \xi' = \frac{\xi \cdot n}{c' \cdot k^4 \cdot n^{80\varepsilon/\varphi^2} \cdot s}. \tag{1}$$

Moreover, according to the proof of Lemma 29 in Gluch et al. (2021), we have that, with probability at least $1 - n^{-100}$,

$$\|A\|_2 \leq \frac{4 \cdot n^{1+160\varepsilon/\varphi^2}}{s}. \tag{2}$$

And with probability 1, we have

$$\|\mathbf{a}_x\|_2 \leq O(\sqrt{s} \cdot k^2 \cdot n^{-1+40\varepsilon/\varphi^2}) \tag{3}$$

and

$$\|\mathbf{a}_y\|_2 \leq O(\sqrt{s} \cdot k^2 \cdot n^{-1+40\varepsilon/\varphi^2}). \tag{4}$$

Now we need to bound $\mathbf{e}_x$ and $\mathbf{e}_y$. Recall that $\mathbf{e}_x = \boldsymbol{\alpha}_x^T - \boldsymbol{p}_x^T(\boldsymbol{M}^t \boldsymbol{S})$, where $\boldsymbol{\alpha}_x \in \mathbb{R}^s$ is obtained by taking entrywise median over all $\boldsymbol{x}_l$'s. Note that in line 5 of Alg. 4, $\boldsymbol{x}_l(i)$ is the output of ESTRWDOT $(G, R_{\text{query}}, t, M_{\text{query}}, x, s_i)$ (Alg. 1). Let $c_3$ be a constant infront of $R$ in Lemma 3.1. Let $\sigma_{\text{err}} = \frac{\xi}{c' \cdot k^2 \cdot n^{1+200\varepsilon/\varphi^2}}$. Thus by our choice of $R_{\text{query}} = \Theta\left(\frac{n^{1+440\varepsilon/\varphi^2}}{M_{\text{query}}} \cdot \frac{k^6}{\xi^2}\right)$ in line 2 of Alg. 4, the prerequisites of Lemma 3.1 are satisfied:

$$R_{\text{query}} = \Theta\left( \frac{n^{1+440\varepsilon/\varphi^2}}{M_{\text{query}}} \cdot \frac{k^6}{\xi^2} \right) \geq \frac{c_3 \cdot k^2 n^{-1+40\varepsilon/\varphi^2}}{\sigma_{\text{err}}^2 \cdot M_{\text{query}}}.$$

Thus we can apply Lemma 3.1. Hence, for any $1 \le i \le s$ with probability at least $0.99$, we have

$$|\boldsymbol{x}_l(i) - \boldsymbol{p}_x^T \boldsymbol{p}_{s_i}| \le \sigma_{\text{err}}.$$

Since we are running $O(\log n)$ rounds to compute $\boldsymbol{x}_l$'s and $\boldsymbol{\alpha}_x$ is obtained by taking entrywise median, we can get that with probability at least $1 - n^{-100}$ for all $z \in I_S$ (see Remark E.1), we have

$$|\boldsymbol{\alpha}_x(z) - \boldsymbol{p}_x^T \boldsymbol{p}_z| \le \sigma_{\text{err}}.$$

Therefore, with probability at least $1 - n^{-100}$, we can get

$$\|\mathbf{e}_x\|_2 = \|\boldsymbol{\alpha}_x^T - \boldsymbol{p}_x^T(\boldsymbol{M}^t \boldsymbol{S})\|_2 \le \sqrt{s} \cdot \sigma_{\text{err}} = \frac{\sqrt{s} \cdot \xi}{c' \cdot k^2 \cdot n^{1+200\varepsilon/\varphi^2}}. \tag{5}$$

Using the same analysis, with probability at least $1 - n^{-100}$, we can get that

$$\|\mathbf{e}_y\|_2 = \|\boldsymbol{\alpha}_y - (\boldsymbol{M}^t \boldsymbol{S})^T \boldsymbol{p}_y\|_2 \le \sqrt{s} \cdot \sigma_{\text{err}} = \frac{\sqrt{s} \cdot \xi}{c' \cdot k^2 \cdot n^{1+200\varepsilon/\varphi^2}}. \tag{6}$$

Putting (1),(2),(3),(4),(5),(6) together and for large enough $n$, we can get

$$\left| \boldsymbol{\alpha}^T \Psi \boldsymbol{\alpha}_y - \boldsymbol{p}_x^T(\boldsymbol{M}^t \boldsymbol{S}) \left( \frac{n}{s} \cdot \widetilde{W}_{[k]} \widetilde{\Sigma}_{[k]}^{-4} \widetilde{W}_{[k]}^T \right) (\boldsymbol{M}^t \boldsymbol{S})^T \boldsymbol{p}_y \right|$$

$$\le \|\mathbf{e}_x\|_2 \|E\|_2 \|\mathbf{e}_y\|_2 + \|\mathbf{e}_x\|_2 \|A\|_2 \|\mathbf{e}_y\|_2 + \|\mathbf{a}_x\|_2 \|E\|_2 \|\mathbf{e}_y\|_2$$

$$+ \|\mathbf{a}_x\|_2 \|A\|_2 \|\mathbf{e}_y\|_2 + \|\mathbf{a}_x\|_2 \|E\|_2 \|\mathbf{a}_y\|_2 + \|\mathbf{e}_x\|_2 \|A\|_2 \|\mathbf{a}_y\|_2 + \|\mathbf{a}_x\|_2 \|E\|_2 \|\mathbf{a}_y\|_2$$

$$\le O(\frac{\xi}{c' \cdot n})$$

$$\le \frac{\xi}{n}.$$

The last inequality holds by setting $c'$ be a large enough constant to cancel the hidden constant of $O(\frac{\xi}{c' \cdot n})$.

Using union bound, if Alg. 3 succeeds, then the above inequality holds with probability at least $1 - 2n^{-100} - n^{-100} - 2n^{-100} = 1 - 5n^{-100}$. $\qquad\square$

Having Lemma 3.1 and Lemma E.10, to prove Theorem 3.2, we also need the following lemma.

**Lemma E.11** (Lemma 19 in Gluch et al. (2021)). *Let $k \ge 2$ be an integer and $\varphi, \varepsilon \in (0,1)$. Let $G = (V, E)$ be a $d$-regular and $(k, \varphi, \varepsilon)$-clusterable graph. Let $\boldsymbol{M}$ be the random walk transition matrix of $G$. Let $I_S = \{s_1, \ldots, s_s\}$ be a multiset of $s$ indices chosen independently and uniformly at random form $V = \{1, \ldots, n\}$. Let $\boldsymbol{S} \in \mathbb{R}^{n \times s}$ be the matrix whose $i$-th column equals $\mathbb{1}_{s_i}$. Let $\sqrt{\frac{n}{s}} \cdot \boldsymbol{M}^t \boldsymbol{S} = \widetilde{U} \widetilde{\Sigma} \widetilde{W}^T$ be an SVD of $\sqrt{\frac{n}{s}} \cdot \boldsymbol{M}^t \boldsymbol{S}$ where $\widetilde{U} \in \mathbb{R}^{n \times n}, \widetilde{\Sigma} \in \mathbb{R}^{n \times n}, \widetilde{W} \in \mathbb{R}^{s \times n}$. Let $\frac{1}{n^6} < \xi < 1$ and $t \ge \frac{20 \log n}{\varphi^2}$. Let $c > 1$ be a large enough constant. Let $s \ge c \cdot n^{480\varepsilon/\varphi^2} \cdot \log n \cdot k^8/\xi^2$. If $\frac{\varepsilon}{\varphi^2} \le \frac{1}{10^5}$, then with probability at least $1 - n^{-100}$, matrix $\widetilde{\Sigma}_{[k]}^{-4}$ exists and we have*

$$\left| \mathbb{1}_x^T \boldsymbol{U}_{[k]} \boldsymbol{U}_{[k]}^T \mathbb{1}_y - (\boldsymbol{M}\mathbb{1}_x)^T (\boldsymbol{M}^t \boldsymbol{S}) \left( \frac{n}{s} \cdot \widetilde{W}_{[k]} \widetilde{\Sigma}_{[k]}^{-4} \widetilde{W}_{[k]}^T \right) (\boldsymbol{M}^t \boldsymbol{S})^T (\boldsymbol{M}\mathbb{1}_y) \right| \le \frac{\xi}{n}.$$

Now we are ready to prove Theorem 3.2.

*Poof of Theorem 3.2.* **Correctness.** Equipped with Lemma E.10, based on the correctness proof of Theorem 2 in Gluch et al. (2021), we can directly obtain the correctness.

Note that in line 3 of Alg. 3, we set $s = O(n^{480\varepsilon/\varphi^2} \cdot \log n \cdot k^8/\xi^2)$, and in line 4 of Alg. 3, we sample $s$ indices independently and uniformly at random form $V = \{1, \ldots, n\}$ to get $I_S = \{s_1, \ldots, s_s\}$. Recall that $\boldsymbol{M}$ is the random walk transition matrix of $G$. Let $\boldsymbol{S} \in \mathbb{R}^{n \times s}$ be the matrix whose $i$-th column is $\mathbb{1}_{s_i}$. Let $\sqrt{\frac{n}{s}} \cdot \boldsymbol{M}^t \boldsymbol{S} = \widetilde{U} \widetilde{\Sigma} \widetilde{W}^T$ be an SVD of $\sqrt{\frac{n}{s}} \cdot \boldsymbol{M}^t \boldsymbol{S}$ where $\widetilde{U} \in \mathbb{R}^{n \times n}, \widetilde{\Sigma} \in \mathbb{R}^{n \times n}, \widetilde{W} \in \mathbb{R}^{s \times n}$.

Recall that for any vertex $x \in V$, we define $\boldsymbol{f}_x = \boldsymbol{U}_{[k]}^T \mathbb{1}_x$ (see Definition 2.1), thus we have $\langle \boldsymbol{f}_x, \boldsymbol{f}_y \rangle = \boldsymbol{f}_x^T \boldsymbol{f}_y = (\boldsymbol{U}_{[k]}^T \mathbb{1}_x)^T \boldsymbol{U}_{[k]}^T \mathbb{1}_y = \mathbb{1}_x^T \boldsymbol{U}_{[k]} \boldsymbol{U}_{[k]}^T \mathbb{1}_y$. For convenience, let us denote $\boldsymbol{B} = (\boldsymbol{M}^t \mathbb{1}_x)^T (\boldsymbol{M}^t \boldsymbol{S}) \left( \frac{n}{s} \cdot \widetilde{W}_{[k]} \widetilde{\Sigma}_{[k]}^{-4} \widetilde{W}_{[k]}^T \right) (\boldsymbol{M}^t \boldsymbol{S})^T (\boldsymbol{M}^t \mathbb{1}_y)$. By trangle inequality, we have

$$
\begin{aligned}
|\langle \boldsymbol{f}_x, \boldsymbol{f}_y \rangle_{\text{apx}} - \langle \boldsymbol{f}_x, \boldsymbol{f}_y \rangle| &= |\langle \boldsymbol{f}_x, \boldsymbol{f}_y \rangle_{\text{apx}} - \boldsymbol{B} + \boldsymbol{B} - \langle \boldsymbol{f}_x, \boldsymbol{f}_y \rangle| \\
&\leq |\langle \boldsymbol{f}_x, \boldsymbol{f}_y \rangle_{\text{apx}} - \boldsymbol{B}| + |\boldsymbol{B} - \langle \boldsymbol{f}_x, \boldsymbol{f}_y \rangle| \\
&= |\langle \boldsymbol{f}_x, \boldsymbol{f}_y \rangle_{\text{apx}} - \boldsymbol{B}| + |\boldsymbol{B} - \langle \mathbb{1}_x^T \boldsymbol{U}_{[k]} \boldsymbol{U}_{[k]}^T \mathbb{1}_y \rangle|.
\end{aligned}
$$

Let $\xi' = \frac{\xi}{2}$. Let $c'$ be a constant in front of $s$ form Lemma E.10. Since $s = O(n^{480\varepsilon/\varphi^2} \cdot \log n \cdot k^8/\xi^2) \geq c' \cdot n^{240\varepsilon/\varphi^2} \cdot \log n \cdot k^4$, then by Lemma E.10, with probability at least $1 - 5n^{-100}$, we have $|\langle \boldsymbol{f}_x, \boldsymbol{f}_y \rangle_{\text{apx}} - \boldsymbol{B}| \leq \frac{\xi'}{n} = \frac{\xi}{2n}$.

Let $c$ be a constant in front of $s$ form Lemma E.11. Since $s = O(n^{480\varepsilon/\varphi^2} \cdot \log n \cdot k^8/\xi^2) \geq c \cdot n^{480\varepsilon/\varphi^2} \cdot \log n \cdot k^8/\xi'^2$ and $\frac{\varepsilon}{\varphi^2} \leq \frac{1}{10^5}$, then by Lemma E.11, with probability at least $1 - n^{-100}$, we have $|\boldsymbol{B} - \langle \mathbb{1}_x^T \boldsymbol{U}_{[k]} \boldsymbol{U}_{[k]}^T \mathbb{1}_y \rangle| \leq \frac{\xi'}{n} = \frac{\xi}{2n}$.

Therefore, by union bound, with probability at least $1 - 5n^{-100} - n^{-100} = 1 - 6n^{-100}$, we have $|\langle \boldsymbol{f}_x, \boldsymbol{f}_y \rangle_{\text{apx}} - \langle \boldsymbol{f}_x, \boldsymbol{f}_y \rangle| \leq \frac{\xi}{2n} + \frac{\xi}{2n} = \frac{\xi}{n}$.

**Runtime and space of INITORACLE.** Algorithm INITORACLE($G, k, \xi, M_{\text{init}}$) (Alg. 3) calls EST-COLLIPROB($G, R_{\text{init}}, t, M_{\text{init}}, I_S$) (Alg. 2) to get $\mathcal{G}$ (see line 5 of Alg. 3). According to Lemma E.5, ESTCOLLIPROB($G, R_{\text{init}}, t, M_{\text{init}}, I_S$) runs in $O(R_{\text{init}} \cdot t \cdot \log n \cdot s^2)$ time and uses $O(M_{\text{init}} \cdot \log^2 n \cdot s^2)$ bits of space. Then in line 7 of INITORACLE, it computes the SVD of matrix $\mathcal{G}$ in $s^3$ time and it uses $s^2 \cdot \log n$ bits of space to store $\Psi \in \mathbb{R}^{n \times n}$. Thus overall INITORACLE runs in $O(R_{\text{init}} \cdot t \cdot \log n \cdot s^2 + s^3)$ time and uses $O(M_{\text{init}} \cdot \log^2 n \cdot s^2 + s^2 \cdot \log n)$ bits of space. By the choice of $t := \frac{20 \log n}{\varphi^2}$, $R_{\text{init}} := \Theta(\frac{n^{1+920\varepsilon/\varphi^2}}{M_{\text{init}}} \cdot \frac{k^{14}}{\xi^2})$ and $s := O(n^{480 \cdot \varepsilon/\varphi^2} \cdot \log n \cdot k^8/\xi^2)$ as in INITORACLE, we get that INITORACLE runs in $T_{\text{init}} = (\frac{k}{\xi})^{O(1)} \cdot n^{1+O(\varepsilon/\varphi^2)} \cdot \frac{1}{M_{\text{init}}} \cdot \log^4 n \cdot \frac{1}{\varphi^2}$ time and uses $S_{\text{init}} = (\frac{k}{\xi})^{O(1)} \cdot n^{O(\varepsilon/\varphi^2)} \cdot M_{\text{init}} \cdot \log^4 n$ bits of space.

**Runtime and space of QUERYDOT.** In QUERYDOT (Alg. 4), in lines $3 \sim 6$, it calls ESTRW-DOT($G, R_{\text{query}}, t, M_{\text{query}}, x, s_i$) (Alg. 1) for $O(\log n \cdot s)$ times. According to Lemma 3.1, ESTRW-DOT($G, R_{\text{query}}, t, M_{\text{query}}, x, s_i$) runs in $O(R_{\text{query}} \cdot t)$ time and uses $O(M_{\text{query}} \cdot \log n)$ bits of space. Moreover, in line 9 of QUERYDOT, it returns $\langle \boldsymbol{f}_x, \boldsymbol{f}_y \rangle_{\text{apx}} = \boldsymbol{\alpha}_x^T \Psi \boldsymbol{\alpha}_y$, which can be computed in $O(s^2)$ time, since we can compute $\boldsymbol{a} = \boldsymbol{\alpha}_x^T \Psi$ in $s^2$ time and then we compute $\boldsymbol{a}\boldsymbol{\alpha}_y$ in $s^2$ time. Thus overall QUERYDOT runs in $O(\log n \cdot s \cdot R_{\text{query}} \cdot t + s^2)$ time and $O(\log^2 n \cdot s \cdot M_{\text{query}})$ bits of space. By the choice of $t := \frac{20 \log n}{\varphi^2}$, $R_{\text{query}} := \Theta(\frac{n^{1+440\varepsilon/\varphi^2}}{M_{\text{query}}} \cdot \frac{k^6}{\xi^2})$ and $s := O(n^{480 \cdot \varepsilon/\varphi^2} \cdot \log n \cdot k^8/\xi^2)$ as in QUERYDOT, we get that QUERYDOT runs in $T_{\text{query}} = (\frac{k}{\xi})^{O(1)} \cdot n^{1+O(\varepsilon/\varphi^2)} \cdot \frac{1}{M_{\text{query}}} \cdot \log^3 n \cdot \frac{1}{\varphi^2}$ time and uses $S_{\text{query}} = (\frac{k}{\xi})^{O(1)} \cdot n^{O(\varepsilon/\varphi^2)} \cdot M_{\text{query}} \cdot \log^3 n$ bits of space.

□

# F  PROOF OF ITEM 1 IN THEOREM 3.1

In this section, we first present an algorithm for computing the spectral dot product in a subspace, which will serve as a building block for the sublinear spectral clustering oracle that relies on a $\log(k)$ conductance gap. Next, we introduce the sublinear spectral clustering oracle, originally proposed in Gluch et al. (2021), corresponding to Item 1 in Theorem 3.1. Finally, we provide the proof of Item 1 in Theorem 3.1.

## F.1  DOT PRODUCT ORACLE ON SUBSPACE

Note that the clustering oracle in Gluch et al. (2021) relies on cluster centers:

**Definition F.1** (Cluster center). For a vertex set $C \subset V$, the *cluster center* of $C$ is defined to be

$$\mu_C = \frac{1}{|C|} \sum_{x \in C} \boldsymbol{f}_x.$$

They proved that if $x \in C_i$, then $\boldsymbol{f}_x$ is close to $\mu_{C_i}$, which means $\langle \boldsymbol{f}_x, \mu_C \rangle \geq c \cdot \|\mu_C\|_2^2$, where $c$ is a constant. Therefore, the key idea behind the clustering oracle in Gluch et al. (2021) is to sample a subset of vertices and enumerate possible $k$-partition in order to obtain a good approximation $\widehat{\mu}_1, \ldots, \widehat{\mu}_k$ to the true cluster centers $\mu_1, \ldots, \mu_k$ (see lines $6 \sim 11$ of Alg. 7). When answering an arbitrary WHICHCLUSTER $(G, x)$ query, the oracle assigns the $x$ to the cluster whose center is close to $\boldsymbol{f}_x$ while other cluster centers are not close to $\boldsymbol{f}_x$ (see line 5 of Alg. 11).

In fact, their clustering algorithm uses hyperplane partitioning, which requires computing dot products in the subspace (i.e., $\langle \boldsymbol{f}_x, \Pi\mu \rangle$). Therefore, we first present the algorithm that computes the dot products in the subspace based on our improved version. We highlight that this (i.e., Alg. 6) is not our contribution.

---

**Algorithm 6:** DOTPRODUCTORACLEONSUBSPACE$(G, x, y, \xi, \Psi, M, B_1, \ldots, B_r)$

---

1   Let $\boldsymbol{X} \in \mathbb{R}^{r \times r}, \boldsymbol{h}_x \in \mathbb{R}^r, \boldsymbol{h}_y \in \mathbb{R}^r$

2   Let $\xi' = \Theta(\xi \cdot n^{-80\varepsilon/\varphi^2} \cdot k^{-6})$

3   **for** $i, j \in [r]$ **do**

4      $\boldsymbol{X}(i, j) := \frac{1}{|B_i||B_j|} \cdot \sum_{z_i \in B_i} \sum_{z_j \in B_j} \text{QUERYDOT}(G, z_i, z_j, \xi', \Psi, M)$

5   **for** $i \in [r]$ **do**

6      $\boldsymbol{h}_x(i) := \frac{1}{|B_i|} \cdot \sum_{z_i \in B_i} \text{QUERYDOT}(G, z_i, x, \xi', \Psi, M)$

7      $\boldsymbol{h}_y(i) := \frac{1}{|B_i|} \cdot \sum_{z_i \in B_i} \text{QUERYDOT}(G, z_i, y, \xi', \Psi, M)$

8   return $\langle \boldsymbol{f}_x, \widehat{\Pi} \boldsymbol{f}_y \rangle_{\text{apx}} := \text{QUERYDOT}(G, x, y, \xi', \Psi, M) - \boldsymbol{h}_x^T \boldsymbol{X}^{-1} \boldsymbol{h}_y$

---

In the following, we will give some informal theorem and corollaries about Alg. 6. Note that the only modification we make to Alg. 6 is to replace SPECTRALDOTPRODUCT with our improved version. Since our dot product oracle provides the same correctness guarantees as the original one, the correctness of the theorem and corollaries concerning Alg. 6 follows immediately from the proof of Theorem 6 in Gluch et al. (2021). Therefore, we focus on analyzing the time and space complexities.

**Theorem F.1** (Informal). *Let $k \geq$ be an integer, $\varphi$, $\frac{1}{n^5} < \xi < 1$ and $\frac{\varepsilon}{\varphi^2}$ be smaller than a positive absolute constant. Let $G = (V, E)$ be a $d$-regular and $(k, \varphi, \varepsilon)$-clusterable graph with $C_1, \ldots, C_k$.*

*Let $r \in [k]$. Let $B_1, \ldots, B_r$ denote multisets of vertices. Let $b = \max_{i \in [r]} |B_i|$. Let $\widehat{\mu}_i = \frac{1}{|B_i|} \sum_{x \in B_i} \boldsymbol{f}_x$. Let $\widehat{\Pi}$ is defined as a orthogonal projection onto the span $(\{\widehat{\mu}_1, \ldots, \widehat{\mu}_r\})^{\perp}$. Then for all $x, y \in V$, we have*

     *1 $\left| \langle \boldsymbol{f}_x, \widehat{\Pi} \boldsymbol{f}_y \rangle_{\text{apx}} - \langle \boldsymbol{f}_x, \widehat{\Pi} \boldsymbol{f}_y \rangle \right| \leq \frac{\xi}{n}$, where $\langle \boldsymbol{f}_x, \widehat{\Pi} \boldsymbol{f}_y \rangle_{\text{apx}}$ is the output of Alg. 6,*

     *2 Alg. 6 runs in $b^2 \cdot (\frac{k}{\xi})^{O(1)} \cdot n^{1 + O(\varepsilon/\varphi^2)} \cdot \frac{1}{M} \cdot \log^3 n \cdot \frac{1}{\varphi^2}$ time,*

     *3 Alg. 6 uses $b^2 \cdot (\frac{k}{\xi})^{O(1)} \cdot n^{O(\varepsilon/\varphi^2)} \cdot M \cdot \log^3 n$ bits of space.*

*Proof.* In lines $3 \sim 4$ of Alg. 6, to compute $\boldsymbol{X}$, Alg. 6 calls QUERYDOT for $r^2 \cdot b^2 \leq k^2 \cdot b^2$ times. In lines $5 \sim 7$ of Alg. 6, to compute $\boldsymbol{h}_x, \boldsymbol{h}_y$, Alg. 6 calls QUERYDOT for $r \cdot b \leq k \cdot b$ times. To compute $\boldsymbol{X}^{-1}$, it takes $r^3 \leq k^3$ time. Therefore, Alg. 6 runs in $k^2 \cdot b^2 \cdot T_{\text{query}} + k \cdot b \cdot T_{\text{query}} + k^3$ time and it uses $k^2 \cdot b^2 \cdot S_{\text{query}} + k \cdot b \cdot S_{\text{query}} + k^2$ bits of space. Note that $T_{\text{query}} = (\frac{k}{\xi'})^{O(1)} \cdot n^{1 + O(\varepsilon/\varphi^2)} \cdot \frac{1}{M} \cdot \log^3 n \cdot \frac{1}{\varphi^2}$ and $S_{\text{query}} = (\frac{k}{\xi'})^{O(1)} \cdot n^{O(\varepsilon/\varphi^2)} \cdot M \cdot \log^3 n$, where $\xi' = \Theta(\xi \cdot n^{-80\varepsilon/\varphi^2} \cdot k^{-6})$. Therefore, we get that Alg. 6 runs in $b^2 \cdot (\frac{k}{\xi})^{O(1)} \cdot n^{1 + O(\varepsilon/\varphi^2)} \cdot \frac{1}{M} \cdot \log^3 n \cdot \frac{1}{\varphi^2}$ time and uses $b^2 \cdot (\frac{k}{\xi})^{O(1)} \cdot n^{O(\varepsilon/\varphi^2)} \cdot M \cdot \log^3 n$ bits of space. $\square$

**Corollary F.1.** *There exists an algorithm that*

     *1 returns a value $\langle \boldsymbol{f}_x, \widehat{\Pi}\widehat{\mu} \rangle_{\text{apx}}$ such that $\left| \langle \boldsymbol{f}_x, \widehat{\Pi}\widehat{\mu} \rangle_{\text{apx}} - \langle \boldsymbol{f}_x, \widehat{\Pi}\widehat{\mu} \rangle \right| \leq \frac{\xi}{n}$,*

    *2 runs in $b^3 \cdot (\frac{k}{\xi})^{O(1)} \cdot n^{1+O(\varepsilon/\varphi^2)} \cdot \frac{1}{M} \cdot \log^3 n \cdot \frac{1}{\varphi^2}$ time,*

    *3 uses $b^3 \cdot (\frac{k}{\xi})^{O(1)} \cdot n^{O(\varepsilon/\varphi^2)} \cdot M \cdot \log^3 n$ bits of space.*

*Proof.* One can compute $\langle \boldsymbol{f}_x, \widehat{\Pi}\widehat{\mu} \rangle_{\text{apx}} := \frac{1}{|B|} \cdot \sum_{y \in B}$ DOTPRODUCTORACLEONSUBSPACE$(G, x, y,$ $\xi, \Psi, M, B_1, \ldots, B_r)$ (Alg. 6). Therefore, the algorithm that computes $\langle \boldsymbol{f}_x, \widehat{\Pi}\widehat{\mu} \rangle_{\text{apx}}$ calls Alg. 6 $b$ times, which ends the proof. $\qquad\square$

**Corollary F.2.** *There exists an algorithm that*

    *1 returns a value $\|\widehat{\Pi}\widehat{\mu}\|^2_{\text{apx}}$ such that $\left| \|\widehat{\Pi}\widehat{\mu}\|^2_{\text{apx}} - \|\widehat{\Pi}\widehat{\mu}\|^2 \right| \le \frac{\xi}{n}$,*

    *2 runs in $b^4 \cdot (\frac{k}{\xi})^{O(1)} \cdot n^{1+O(\varepsilon/\varphi^2)} \cdot \frac{1}{M} \cdot \log^3 n \cdot \frac{1}{\varphi^2}$ time,*

    *3 uses $b^4 \cdot (\frac{k}{\xi})^{O(1)} \cdot n^{O(\varepsilon/\varphi^2)} \cdot M \cdot \log^3 n$ bits of space.*

*Proof.* One can compute $\|\widehat{\Pi}\widehat{\mu}\|^2_{\text{apx}} = (\widehat{\Pi}\widehat{\mu})^T(\widehat{\Pi}\widehat{\mu}) = \widehat{\mu}^T\widehat{\Pi}^T\widehat{\Pi}\widehat{\mu} = \widehat{\mu}^T\widehat{\Pi}\widehat{\mu} = \langle \widehat{\mu}, \widehat{\Pi}\widehat{\mu} \rangle = \frac{1}{|B|} \cdot \sum_{x \in B} \langle \boldsymbol{f}_x, \widehat{\Pi}\widehat{\mu} \rangle_{\text{apx}}$. Therefore, the algorithm that computes $\|\widehat{\Pi}\widehat{\mu}\|^2_{\text{apx}}$ calls the algorithm in Corollary F.1 $b$ times, which ends the proof. $\qquad\square$

### F.2 SUBLINEAR SPECTRAL CLUSTERING ORACLE

Now we present the sublinear spectral clustering oracle with a $\log(k)$ gap between inner and outer conductance, originally proposed in Gluch et al. (2021), and adapt it by incorporating our dot product oracle, which operates with very little memory.

Algorithm 7 finds some cluster centers that reflects the clustering structure of the input graph.

---

**Algorithm 7:** FINDCENTERS$(G, M)$

---

1   INITORACLE$(G, k, 10^{-6} \cdot \frac{\sqrt{\varepsilon}}{\varphi}, M)$

2   $s_1 := \Theta\left(\frac{\varphi^2}{\varepsilon} k^5 \log^2 k \log(1/\eta)\right)$, $s_2 := \Theta\left(\frac{\varphi^4}{\varepsilon^2} k^5 \log^2 k \log(1/\eta)\right)$

3   **for** $t \in [1 \ldots \log(2/\eta)]$ **do**

4      $S :=$ Random samples of vertices of $V$ of size $s = \Theta(\frac{\varphi^2}{\varepsilon} k^4 \log k)$

5      **for** $(P_1, P_2, \ldots, P_k) \in$ PARTITION $(S)$ **do**

6         **for** $i = 1$ *to* $k$ **do**

7            $\widehat{\mu}_i := \frac{1}{|P_i|} \sum_{x \in P_i} f_x$

8         $(r, C) :=$ COMPUTEORDEREDPARTITION$(G, (\widehat{\mu}_1, \ldots, \widehat{\mu}_k)), s_1, s_2, M)$

9         **if** $r =$ TRUE **then**

10           return $C$

---

**Algorithm 8:** COMPUTEORDEREDPARTITION$(G, (\widehat{\mu}_1, \ldots, \widehat{\mu}_k), s_1, s_2, M)$

---

1   $S := \{\widehat{\mu}_1, \ldots, \widehat{\mu}_k\}$

2   **for** $i = 1$ *to* $\lceil \log k \rceil$ **do**

3      $T_i := \emptyset$

4      **for** $\widehat{\mu} \in S$ **do**

5         $\psi :=$ OUTERCONDUCTANCE$(G, \widehat{\mu}, (T_1, \ldots, T_{i-1}), S, s_1, s_2, M)$

6         **if** $\psi \le O(\frac{\varepsilon}{\varphi^2} \cdot \log k)$ **then**

7           $T_i := T_i \cup \{\widehat{\mu}\}$

8      $S := S \backslash T_i$

9      **if** $S = \emptyset$ **then**

10         return $(\text{TRUE}, (T_1, \ldots, T_i))$

11   return $(\text{FALSE}, \perp)$

---

**Algorithm 9:** OUTERCONDUCTANCE$(G, \widehat{\mu}, (T_1, \ldots, T_b), S, s_1, s_2, M)$

1   cnt $:= 0$
2   **for** $t = 1$ *to* $s_1$ **do**
3     $x \sim$ UNIFORM$\{1 \ldots n\}$
4     **if** ISINSIDE$(x, \widehat{\mu}, (T_1, \ldots, T_b), S, M)$ **then**
5       cnt $:=$ cnt $+ 1$

6   **if** $\frac{n}{s_1} \cdot$ cnt $< \min_{p \in [k]} |C_p|/2$ **then**
7     **return** $\infty$

8   $e := 0, a := 0$
9   **for** $t = 1$ *to* $s_2$ **do**
10    $x \sim$ UNIFORM$\{1 \ldots n\}$
11    $y \sim$ UNIFORM$\{w \in \mathcal{N}(u)\}$
12    **if** ISINSIDE$(x, \widehat{\mu}, (T_1, \ldots, T_b), S, M)$ **then**
13      $a := a + 1$
14      **if** $\neg$ISINSIDE$(y, \widehat{\mu}, (T_1, \ldots, T_b), S, M)$ **then**
15        $e := e + 1$

16   **return** $\frac{e}{a}$

---

**Algorithm 10:** ISINSIDE$(x, \widehat{\mu}, (T_1, \ldots, T_b), S, M)$

1   **for** $i = 1$ *to* $b$ **do**
2    Let $\Pi$ be the projection onto the span $(\cup_{j<i} T_j)^{\perp}$
3    Let $S_i = (\cup_{j \geq i} T_j) \cup S$
4    **for** $\widehat{\mu}_i \in T_i$ **do**
5      **if** $x \in C^{\text{apx}}_{\Pi \widehat{\mu}_i, 0.93} \setminus \cup_{\widehat{\mu}' \in S_i \setminus \{\widehat{\mu}_i\}} C^{\text{apx}}_{\Pi \widehat{\mu}', 0.93}$ **then**
6        **return** FALSE

7   Let $\Pi$ be the projection onto the span $(\cup_{j \leq b} T_j)^{\perp}$
8   **if** $x \in C^{\text{apx}}_{\Pi \widehat{\mu}, 0.93} \setminus \cup_{\widehat{\mu}' \in S \setminus \{\widehat{\mu}\}} C^{\text{apx}}_{\Pi \widehat{\mu}', 0.93}$ **then**
9    **return** TRUE

10 **return** FALSE

---

Algorithm 11 corresponds to the query phase of the clustering oracle where it is used to assign vertices to clusters based on cluster centers.

**Algorithm 11:** HYPERPLANEPARTITIONING$(x, (T_1, \ldots, T_b), M)$

1   **for** $i = 1$ *to* $b$ **do**
2    Let $\Pi$ be the projection onto the span $(\cup_{j<i} T_j)^{\perp}$
3    Let $S_i = (\cup_{j \geq i} T_j)$
4    **for** $\widehat{\mu} \in T_i$ **do**
5      **if** $x \in C^{\text{apx}}_{\Pi \widehat{\mu}, 0.93} \setminus \cup_{\widehat{\mu}' \in S_i \setminus \{\widehat{\mu}\}} C^{\text{apx}}_{\Pi \widehat{\mu}', 0.93}$ **then**
6        **return** $\widehat{\mu}$

---

### F.3   DEFERRED PROOF

**Theorem F.2** (Restate of Item 1 in Theorem 3.1). *Let $k \geq 2$ be an integer, $\varphi, \varepsilon \in (0, 1)$ and $h_1(k, \varphi), h_2(k, \varepsilon)$ and $h_3(k, \varphi, \varepsilon)$ be three functions. Let $\varepsilon \ll h_1(k, \varphi)$. Let $G = (V, E)$ be a d-regular and $(k, \varphi, \varepsilon)$-clusterable graph with $C_1, \ldots, C_k$. Let $n^{c \cdot \varepsilon / \varphi^2} \leq M \leq O\left(\frac{n^{1/2 - O(\varepsilon/\varphi^2)}}{k}\right)$ be a trade-off parameter, where $c$ is a large enough constant. There exists a sublinear spectral clustering oracle that:*

- *constructs a data structure $\mathcal{D}$ using $\widetilde{O}_{\varphi}\left(h_2(k) \cdot n^{O(\varepsilon/\varphi^2)} \cdot M\right)$ bits of space,*

- *answers any* WHICHCLUSTER *query using $\mathcal{D}$ in* $\widetilde{O}_\varphi \left( h_2(k) \cdot n^{1+O(\varepsilon/\varphi^2)} \cdot \frac{1}{M} \right)$ *time,*
- *has* $O\left( h_3(k,\varphi,\varepsilon) \right) |C_i|$ *misclassification error for each* $i \in [k]$,

*where we use $O_\varphi$ to suppress dependence on $\varphi$ and $\widetilde{O}$ to hide all* $\mathrm{poly}(\log n)$ *factors and:*

*1 if* $h_1(k,\varphi) = \frac{\varphi^3}{\log k}$, *then* $h_2(k,\varepsilon) = \left( \frac{k}{\varepsilon} \right)^{O(1)}$ *and* $h_3(k,\varphi,\varepsilon) = \frac{\varepsilon}{\varphi^3} \cdot \log k$.

*Proof.* **Space and runtime.** In the preprocessing phase, as line 1 of FINDCENTERS (Alg. 7), it invokes INITORACLE$(G, k, \xi, M)$ one time to get a matrix $\Psi$, which takes $S_{\text{init}}$ bits of space according to Theorem 3.2. Then it samples $s = \frac{\varphi^2}{\varepsilon} k^4 \log k$ vertices and tests all the possible $k$-partitions of the sample set. For each partition, it invokes Alg. 8 one time. Each run of Alg. 8 invokes Alg. 9 $k \log k$ times. Each run of Alg. 9 invokes Alg. 10 $(s_1 + s_2)$ times. Each run of Alg. 10 computes $C_{\Pi\widehat{\mu}, 0.93}^{\text{apx}}$ about $k^{O(1)}$ times, where $C_{\Pi\widehat{\mu}, 0.93}^{\text{apx}} = \{x \in V, \frac{\langle \boldsymbol{f}_x, \Pi\widehat{\mu} \rangle_{\text{apx}}}{\|\Pi\widehat{\mu}\|_{\text{apx}}^2} \geq 0.93\}$. According to Corollary F.1 and Corollary F.2, computing $\frac{\langle \boldsymbol{f}_x, \Pi\widehat{\mu} \rangle_{\text{apx}}}{\|\Pi\widehat{\mu}\|_{\text{apx}}^2}$ takes $s^4 \cdot \left( \frac{k\varphi}{\varepsilon} \right)^{O(1)} \cdot n^{O(\varepsilon/\varphi^2)} \cdot M_{\text{query}} \cdot \log^3 n$ bits of space, where we set $\xi = 10^{-6} \cdot \frac{\sqrt{\varepsilon}}{\varphi}$. Therefore, Alg. 7 uses $S_{\text{init}} + k \log k \cdot (s_1 + s_2) \cdot s^4 \cdot \left( \frac{k\varphi}{\varepsilon} \right)^{O(1)} \cdot n^{O(\varepsilon/\varphi^2)} \cdot M_{\text{query}} \cdot \log^3 n$ bits of space. By setting $s_1 := \Theta\left( \frac{\varphi^2}{\varepsilon} k^5 \log^2 k \log(1/\eta) \right)$, $s_2 := \Theta\left( \frac{\varphi^4}{\varepsilon^2} k^5 \log^2 k \log(1/\eta) \right)$, $\eta = O(\log n)$ and $M_{\text{query}} = M$, we get that Alg. 7 uses $\left( \frac{k\varphi}{\varepsilon} \right)^{O(1)} \cdot n^{O(\varepsilon/\varphi^2)} \cdot M \cdot \mathrm{poly}(\log n)$ bits of space to get a matrix $\Psi$ and a collection of vertex sets $C$ that represents the cluster centers.

In the query phase, HYPERPLANEPARTITIONING (Alg. 11) computes $C_{\Pi\widehat{\mu}, 0.93}^{\text{apx}}$ about $k^{O(1)}$ times, where $C_{\Pi\widehat{\mu}, 0.93}^{\text{apx}} = \{x \in V, \frac{\langle \boldsymbol{f}_x, \Pi\widehat{\mu} \rangle_{\text{apx}}}{\|\Pi\widehat{\mu}\|_{\text{apx}}^2} \geq 0.93\}$. According to Corollary F.1 and Corollary F.2, computing $\frac{\langle \boldsymbol{f}_x, \Pi\widehat{\mu} \rangle_{\text{apx}}}{\|\Pi\widehat{\mu}\|_{\text{apx}}^2}$ takes $s^4 \cdot \left( \frac{k\varphi}{\varepsilon} \right)^{O(1)} \cdot n^{O(\varepsilon/\varphi^2)} \cdot M \cdot \log^3 n$ bits of space and $s^4 \cdot \left( \frac{k}{\varepsilon} \right)^{O(1)} \cdot n^{1+O(\varepsilon/\varphi^2)} \cdot \frac{1}{M} \cdot \log^3 n \cdot \frac{1}{\varphi^2}$ time, where we set $\xi = 10^{-6} \cdot \frac{\sqrt{\varepsilon}}{\varphi}$. By setting $s = \frac{\varphi^2}{\varepsilon} k^4 \log k$, we get that Alg. 11 takes $\left( \frac{k\varphi}{\varepsilon} \right)^{O(1)} \cdot n^{O(\varepsilon/\varphi^2)} \cdot M \cdot \mathrm{poly}(\log n)$ bits of space and $\left( \frac{k\varphi}{\varepsilon} \right)^{O(1)} \cdot n^{1+O(\varepsilon/\varphi^2)} \cdot \frac{1}{M} \cdot \mathrm{poly}(\log n)$ time.

Thus, the clustering oracle constructs a data structure $\mathcal{D}$ (including matrix $\Psi$, cluster centers $C$ and other information used by the query phase) using $\left( \frac{k\varphi}{\varepsilon} \right)^{O(1)} \cdot n^{O(\varepsilon/\varphi^2)} \cdot M \cdot \mathrm{poly}(\log n)$ bits of space. Using $\mathcal{D}$, any WHICHCLUSTER query can be answered by Alg. 11 in $\left( \frac{k\varphi}{\varepsilon} \right)^{O(1)} \cdot n^{1+O(\varepsilon/\varphi^2)} \cdot \frac{1}{M} \cdot \mathrm{poly}(\log n)$ time.

**Correctness.** We highlight that the sublinear spectral clustering oracle is not our contribution. Note that the only modification we make to the clustering oracle is to replace the dot product oracle used in the original work (Gluch et al., 2021) with our improved oracle. Since the correctness guarantees (i.e., conductance gap and misclassification error) of the clustering oracle rely on the properties of the dot product oracle, and our dot product oracle satisfies the same correctness guarantees with the previous one, the correctness of the overall clustering oracle follows directly from the correctness of the clustering oracle in Gluch et al. (2021). $\qquad\square$

## G SUBLINEAR CLUSTERING ORACLE RELATED TO ITEM 2 IN THEOREM 3.1

In this section, we present the sublinear spectral clustering oracle with a $\mathrm{poly}(k)$ gap between inner and outer conductance, originally proposed in Shen & Peng (2023), and adapt it by incorporating our dot product oracle, which operates with very little memory.

Algorithm 12 first initializes our dot product oracle to get a matrix $\Psi$ (see line 5). It then leverages our dot product oracle to estimate $\langle \boldsymbol{f}_x, \boldsymbol{f}_y \rangle$ for all pairs of vertices $x, y$ in the sample set $S$, which are subsequently used to construct a similarity graph $H$ (see lines $6 \sim 9$).

---

**Algorithm 12:** CONSTRUCTORACLE$(G, k, \varphi, \varepsilon, \gamma, M)$

---

1  Let $\xi = \frac{\sqrt{\gamma}}{1000}$ and let $s = \frac{10 \cdot k \log k}{\gamma}$
2  Let $\theta = 0.96(1 - \frac{4\sqrt{\varepsilon}}{\varphi})\frac{\gamma k}{n} - \frac{\sqrt{k}}{n}(\frac{\varepsilon}{\varphi^2})^{1/6} - \frac{\xi}{n}$
3  Sample a set S of $s$ vertices independently and uniformly at random from $V$
4  Generate a similarity graph $H = (S, \emptyset)$
5  Let $\Psi = $ INITORACLE$(G, k, \xi, M)$
6  **for** *any $u, v \in S$* **do**
7  |   Let $\langle \boldsymbol{f}_u, \boldsymbol{f}_v \rangle_{\text{apx}} = $ QUERYDOT$(G, u, v, \xi, \Psi, M)$
8  |   **if** $\langle \boldsymbol{f}_u, \boldsymbol{f}_v \rangle_{\text{apx}} \geq \theta$ **then**
9  |   |   Add an edge $(u, v)$ to the similarity graph $H$

10 **if** *$H$ has exactly $k$ connected components* **then**
11 |   Label the connected components with $1, 2, \ldots, k$ (we write them as $S_1, \ldots, S_k$)
12 |   Label $x \in S$ with $i$ if $x \in S_i$
13 |   Return $H$ and the vertex labeling $\ell$
14 **else**
15 |   return **fail**

---

**Algorithm 13:** SEARCHINDEX$(H, \ell, x, M)$

---

1 **for** *any vertex $u \in S$* **do**
2 |   Let $\langle \boldsymbol{f}_u, \boldsymbol{f}_x \rangle_{\text{apx}} = $ QUERYDOT$(G, u, x, \xi, \Psi, M)$
3 **if** *there exists a unique index $1 \leq i \leq k$ such that $\langle \boldsymbol{f}_u, \boldsymbol{f}_x \rangle_{\text{apx}} \geq \theta$ for all $u \in S_i$* **then**
4 |   return index $i$
5 **else**
6 |   return **outlier**

---

Algorithm 14 corresponds to the query phase of the sublinear spectral clustering oracle, where it answers any WHICHCLUSTER query using matrix $\Psi$ and similarity graph $H$.

---

**Algorithm 14:** WHICHCLUSTER$(G, x, M)$

---

1 **if** *preprocessing phase **fails*** **then**
2 |   return **fail**
3 **if** SEARCHINDEX*$(H, \ell, x, M)$ return **outlier*** **then**
4 |   return a random index$\in [k]$
5 **else**
6 |   return SEARCHINDEX$(H, \ell, x, M)$

---

## H  PROOF OF THEOREM 1.2

**Theorem H.1** (Restate of Theorem 1.2)**.** *For any trade-off parameter $1 \leq M \leq O(\sqrt{n})$, there exists an algorithm (Alg. 5) that, with probability at least $1 - 2n^{-100}$, solves the 1-cluster vs. 2-cluster problem. Moreover, the algorithm:*

- *uses $\widetilde{O}(M)$ bits of space,*
- *runs in $\widetilde{O}\left(\frac{n}{M}\right)$ time.*

To prove Theorem 1.2, we need the following lemmas.

**Lemma H.1** (Cheeger's inequality)**.** *In holds for any graph $G$ that*

$$\frac{\lambda_2}{2} \leq \phi(G) \leq \sqrt{2\lambda_2}.$$

Lemma H.2 bounds the $\ell_2$-norm of the $t$-step random walk distribution starting from any vertex $x$ in a $d$-regular graph, distinguishing between the case where the graph is a single $\varphi$-expander and the case where it consists of two disjoint $\varphi$-expanders.

**Lemma H.2** (Expander related version of Lemma E.4). *Let $\varphi \in (0,1)$. Let $G$ be a $d$-regular graph. Let $M$ be the random walk transition matrix of $G$. For any $t \geq \frac{20 \log n}{\varphi^2}$ and any $x \in V$,*

*1 if $G$ is a $\varphi$-expander of size $n$, then $\|M^t \mathbb{1}_x\|_2 \leq \sqrt{\frac{2}{n}}$,*

*2 if $G$ is the disjoint union of two identical $\varphi$-expanders of size $n/2$, then $\|M^t \mathbb{1}_x\|_2 \leq \sqrt{\frac{3}{n}}$.*

*Proof.* **Item 1.** Let $L$ be the normalized Laplacian matrix of $G$. Recall that we use $0 = \lambda_1 \leq \cdots \leq \lambda_n \leq 2$ to denote the eigenvalues of $L$ and we use $u_1, \ldots, u_n$ to denote the corresponding eigenvectors, where $u_1, \ldots, u_n$ form an orthonormal basis of $\mathbb{R}^n$ and $u_1(x) = \frac{1}{\sqrt{n}}$ for any $x \in V$. Note that $M = I - \frac{L}{2}$. Hence, the eigenvalues of $M$ are given by $1 = 1 - \frac{\lambda_1}{2} \geq \cdots \geq 1 - \frac{\lambda_n}{2} \geq 0$, and the corresponding eigenvectors are still $u_1, \ldots, u_n$. For convenience, we relabel the eigenvalues of $M$ as $1 = v_1(M) = (1 - \frac{\lambda_1}{2}) \geq v_2(M) = (1 - \frac{\lambda_2}{2}) \geq \cdots \geq v_n(M) = (1 - \frac{\lambda_n}{2}) \geq 0$. Moreover, we can write that $\mathbb{1}_x = \sum_{i=1}^n \alpha_i u_i$. Note that $u_j^T \mathbb{1}_x = \sum_{i=1}^n \alpha_i u_j^T u_i = \alpha_j$. Therefore, $\alpha_j$ corresponds to $u_j^T \mathbb{1}_x = u_j(x)$. Now, we have

$$M^t \mathbb{1}_x = M^t \sum_{i=1}^n \alpha_i u_i = \sum_{i=1}^n \alpha_i M^t u_i = \sum_{i=1}^n \alpha_i (v_i(M))^t u_i.$$

Thus, we have

$$\|M^t \mathbb{1}_x\|_2^2 = (M^t \mathbb{1}_x)^T (M^t \mathbb{1}_x) = \sum_{i=1}^n \alpha_i^2 (v_i(M))^{2t}$$

$$= \alpha_1^2 (v_1(M))^{2t} + \sum_{i=2}^n \alpha_i^2 (v_i(M))^{2t}$$

$$\leq \frac{1}{n} + (v_2(M))^{2t} \cdot \sum_{i=2}^n \alpha_i^2$$

$$\leq \frac{1}{n} + (v_2(M))^{2t} \cdot (n-1).$$

Since $G$ is a $\varphi$-expander, according to Cheeger's inequality (Lemma H.1), we get that $\lambda_2 \geq \frac{\varphi^2}{2}$. Therefore, for any $t \geq \frac{20 \log n}{\varphi^2}$, we have

$$v_2(M)^{2t} = \left(1 - \frac{\lambda_2}{2}\right)^{2t} \leq \left(1 - \frac{\varphi^2}{4}\right)^{\frac{4}{\varphi^2} \cdot 10 \log n} \leq \frac{1}{n^{10}}.$$

Combine above results together, we get that

$$\|M^t \mathbb{1}_x\|_2^2 \leq \frac{1}{n} + \frac{1}{n^{10}} \cdot (n-1) = \frac{1}{n} + \frac{1}{n^9} \leq \frac{2}{n}.$$

**Item 2.** We use $C_1, C_2$ to denote the two $\varphi$-expanders in $G$. Since $C_1$ and $C_2$ are disconnected, the normalized Laplacian matrix $L$ of $G$ can be written in block-diagonal form as

$$L = \begin{pmatrix} L_{C_1} & 0 \\ 0 & L_{C_2} \end{pmatrix},$$

where $L_{C_1} \in \mathbb{R}^{\frac{n}{2} \times \frac{n}{2}}$ and $L_{C_2} \in \mathbb{R}^{\frac{n}{2} \times \frac{n}{2}}$ are the normalized Laplacian matrix of $C_1$ and $C_2$, respectively. For $L_{C_i}$, we use $0 = \lambda_1^{C_i} \leq \cdots \leq \lambda_{n/2}^{C_i} \leq 2$ to denote the eigenvalues of $L_{C_i}$ and we use $u_1^{C_i}, \ldots, u_{n/2}^{C_i} \in \mathbb{R}^{\frac{n}{2} \times \frac{n}{2}}$ to denote the corresponding eigenvectors, where $u_1^{C_i}, \ldots, u_{n/2}^{C_i}$

from an orthonormal basis of $\mathbb{R}^{\frac{n}{2} \times \frac{n}{2}}$ and $\boldsymbol{u}_1^{C_i}(x) = \sqrt{\frac{2}{n}}$ for any $x \in V$. Therefore, the eigenvalues of $\boldsymbol{L}$ are given by $0 = \lambda_1 \leq \cdots \leq \lambda_{n/2} \leq 2$, each of which has multiplicity two, where $\lambda_i = \lambda_i^{C_1} = \lambda_i^{C_2}$. For $\lambda_i$, we use $\boldsymbol{u}_{2i-1}, \boldsymbol{u}_{2i} \in \mathbb{R}^n$ to denote the corresponding eigenvectors, where $\boldsymbol{u}_{2i-1} = ((\boldsymbol{u}_i^{C_1})^T, 0, \ldots, 0)^T$ and $\boldsymbol{u}_{2i} = (0, \ldots, 0, (\boldsymbol{u}_i^{C_2})^T)^T$. Note that $\boldsymbol{M} = \boldsymbol{I} - \frac{\boldsymbol{L}}{2}$. Hence, the eigenvalues of $\boldsymbol{M}$ are given by $1 = 1 - \frac{\lambda_1}{2} \geq \cdots \geq 1 - \frac{\lambda_{n/2}}{2} \geq 0$, each of which has multiplicity two, and the corresponding eigenvectors are still $\boldsymbol{u}_1, \ldots, \boldsymbol{u}_n$. For convenience, we relabel the eigenvalues of $\boldsymbol{M}$ as $1 = v_1(\boldsymbol{M}) = v_2(\boldsymbol{M}) = (1 - \frac{\lambda_1}{2}) \geq v_3(\boldsymbol{M}) = v_4(\boldsymbol{M}) = (1 - \frac{\lambda_2}{2}) \geq \cdots \geq v_{n-1}(\boldsymbol{M}) = v_n(\boldsymbol{M}) = (1 - \frac{\lambda_{n/2}}{2}) \geq 0$.

Similar to the proof of item 1, we get

$$\|\boldsymbol{M}^t \mathbb{1}_x\|_2^2 = (\boldsymbol{M}^t \mathbb{1}_x)^T (\boldsymbol{M}^t \mathbb{1}_x) = \sum_{i=1}^n \alpha_i^2 (v_i(\boldsymbol{M}))^{2t}$$

$$= \alpha_1^2 + \alpha_2^2 + \sum_{i=3}^n \alpha_i^2 (v_i(\boldsymbol{M}))^{2t}$$

$$\leq \frac{2}{n} + (v_3(\boldsymbol{M}))^{2t} \cdot \sum_{i=3}^n \alpha_i^2$$

$$\leq \frac{2}{n} + (v_3(\boldsymbol{M}))^{2t} \cdot (n-2).$$

Since $C_1$ and $C_2$ both are $\varphi$-expander, according to Cheeger's inequality (Lemma H.1), we get that $\lambda_2^{C_1} = \lambda_2^{C_2} \geq \frac{\varphi^2}{2}$. Therefore, for any $t \geq \frac{20 \log n}{\varphi^2}$, we have

$$(v_3(\boldsymbol{M}))^{2t} = \left(1 - \frac{\lambda_2}{2}\right)^{2t} = \left(1 - \frac{\lambda_2^{C_1}}{2}\right)^{2t} \leq \left(1 - \frac{\varphi^2}{4}\right)^{\frac{4}{\varphi^2} \cdot 10 \log n} \leq \frac{1}{n^{10}}.$$

Combine above results together, we get that

$$\|\boldsymbol{M}^t \mathbb{1}_x\|_2^2 \leq \frac{2}{n} + \frac{1}{n^{10}} \cdot (n-2) = \frac{2}{n} + \frac{1}{n^9} \leq \frac{3}{n}.$$

$\square$

The following lemma shows that, under appropriate parameters, Alg. 1 can estimate the dot product of the random walk distributions from any two vertices up to $\sigma_{\text{err}}$, whether the graph is a single $\varphi$-expander or consists of two disjoint $\varphi$-expanders.

**Lemma H.3** (Expander related version of Lemma 3.1). *Let $\varphi \in (0,1)$. Let $G = (V, E)$ be either a $d$-regular $\varphi$-expander with size $n$ or the disjoint union of two identical $d$-regular $\varphi$-expander of size $n/2$. Let $\boldsymbol{M}$ be the random walk transition matrix of $G$. Let $Z$ be the output of $\text{EstRWDot}(G, R, t, M, x, y)$ (Alg. 1). Let $\sigma_{\text{err}} > 0$. Let $c > 1$ be a large enough constant. For any $t \geq \frac{20 \log n}{\varphi^2}$ and any $x, y \in V$, if $R \geq \frac{c \cdot n^{-1}}{\sigma_{\text{err}}^2 M}$ and $1 \leq M \leq O(n^{1/2})$, then with probability at least $0.99$, we have*

$$|Z - \langle \boldsymbol{M}^t \mathbb{1}_x, \boldsymbol{M}^t \mathbb{1}_y \rangle| \leq \sigma_{\text{err}}.$$

*Moreover, $\text{EstRWDot}(G, R, t, M, x, y)$ runs in $O(Rt)$ time and uses $O(M \cdot \log n)$ bits of space.*

*Proof.* **Runtime and space**. See the proof of Lemma 3.1.

**Correctness.**

By Lemma E.2 and Lemma H.2, we can get that

$$\text{Var}[Z] \leq \frac{1}{R} \left[ \frac{1}{M} \|\boldsymbol{M}^t \mathbb{1}_x\|_2 \cdot \|\boldsymbol{M}^t \mathbb{1}_y\|_2 + \left( \|\boldsymbol{M}^t \mathbb{1}_x\|_2 \cdot \|\boldsymbol{M}^t \mathbb{1}_y\|_2^2 + \|\boldsymbol{M}^t \mathbb{1}_x\|_2^2 \cdot \|\boldsymbol{M}^t \mathbb{1}_y\|_2 \right) \right]$$

$$= \frac{1}{R} \left( \frac{O(n^{-1})}{M} + O(n^{-3/2}) \right).$$

Using Chebyshev's inequality, we have

$$
\begin{aligned}
\Pr[|Z - \langle \boldsymbol{M}^t \mathbb{1}_x, \boldsymbol{M}^t \mathbb{1}_y \rangle| \geq \sigma_{\text{err}}] &= \Pr[|Z - \mathbb{E}[Z]| \geq \sigma_{\text{err}}] \\
&\leq \frac{\text{Var}[Z]}{\sigma_{\text{err}}^2} \\
&\leq \frac{1}{\sigma_{\text{err}}^2} \cdot \frac{1}{R} \left( \frac{O(n^{-1})}{M} + O(n^{-3/2}) \right) \\
&\leq \frac{1}{\sigma_{\text{err}}^2} \cdot \frac{1}{R} \cdot O\left( \frac{n^{-1}}{M} \right) \qquad\qquad M \leq O\left( n^{1/2} \right) \\
&\leq \frac{1}{100}.
\end{aligned}
$$

The last inequality holds by our choice of $R$ as follows, where $c$ is a large enough constant that cancels the constant hidden in $O\left( \frac{n^{-1}}{M} \right)$:

$$R \geq \frac{c \cdot n^{-1}}{\sigma_{\text{err}}^2 M}.$$

$\square$

Lemma H.4 asserts that, under suitable parameters, the output $\mathcal{G}$ of ESTCOLLIPROB (Alg. 2) approximates $(\boldsymbol{M}^t \boldsymbol{S})^T (\boldsymbol{M}^t \boldsymbol{S})$ in spectral norm, where the latter is the Gram matrix of the random walk distributions from sampled vertices, and this holds whether the graph is a single $\varphi$-expander or two disjoint $\varphi$-expanders.

**Lemma H.4** (Expander related version of Lemma E.5). *Let $\varphi \in (0, 1)$. Let $G = (V, E)$ be either a $d$-regular $\varphi$-expander with size $n$ or the disjoint union of two identical $d$-regular $\varphi$-expander of size $n/2$. Let $\boldsymbol{M}$ be the random walk transition matrix of $G$. Let $I_S = \{s_1, \ldots, s_s\}$ be a multiset of $s$ indices chosen from $\{1, \ldots, n\}$. Let $\boldsymbol{S} \in \mathbb{R}^{n \times s}$ be the matrix whose $i$-th column equals $\mathbb{1}_{s_i}$. Let $\mathcal{G} \in \mathbb{R}^{s \times s}$ be the output of ESTCOLLIPROB $(G, R, t, M, I_S)$ (Alg. 2). Let $\sigma_{\text{err}} > 0$. Let $c > 1$ be a large enough constant. For any $t \geq \frac{20 \log n}{\varphi^2}$, if $R \geq \frac{c \cdot n^{-1}}{\sigma_{\text{err}}^2 M}$ and $1 \leq M \leq O\left( n^{1/2} \right)$, then weith probability $1 - n^{-100}$, we have*

$$\|\mathcal{G} - (\boldsymbol{M}^t \boldsymbol{S})^T (\boldsymbol{M}^t \boldsymbol{S})\|_2 \leq s \cdot \sigma_{\text{err}}.$$

*Moreover, ESTCOLLIPROB $(G, R, t, M, I_S)$ runs in $O(Rt \cdot \log n \cdot s^2)$ time and uses $O(M \cdot \log^2 n \cdot s^2)$ bits of space.*

*Proof.* Note that we have established Lemma H.3, which is an analogue of Lemma 3.1 for graph that is either a $\varphi$-expander of size $n$ or the disjoint union of two identical $\varphi$-expanders of size $n/2$. Since the proof of Lemma E.5 relies only on Lemma 3.1, the same augment immediately yields Lemma H.4, the corresponding analogue of Lemma E.5. $\square$

Lemma H.5 demonstrates that $(\boldsymbol{M}^t \boldsymbol{S})(\boldsymbol{M}^t \boldsymbol{S})^T$ has a clear spectral gap between the 1-cluster and 2-cluster cases.

**Lemma H.5** (Expander related version of Lemma E.8). *Let $\varphi \in (0, 1)$. Let $G$ be a $d$-regular graph. Let $\boldsymbol{M}$ be the random walk transition matrix of $G$. Let $I_S = \{s_1, \ldots, s_s\}$ be a multiset of $s$ indices chosen independently and uniformly at random form $V = \{1, \ldots, n\}$. Let $\boldsymbol{S} \in \mathbb{R}^{n \times s}$ be the matrix whose $i$-th column equals $\mathbb{1}_{s_i}$. For any $t \geq \frac{20 \log n}{\varphi^2}$, with probability at least $1 - n^{-100}$, we have*

*1  if $G$ is a $\varphi$-expander of size $n$ and $s \geq 1$, then $v_2 \left( \frac{n}{s} \cdot (\boldsymbol{M}^t \boldsymbol{S})(\boldsymbol{M}^t \boldsymbol{S})^T \right) \leq n^{-9}$,*

    *2 if $G$ is the disjoint union of two identical $\varphi$-expanders of size $n/2$ and $s \geq c \cdot \log n$, where $c > 1$ is a large enough constant, then $v_2\left(\frac{n}{s} \cdot (M^t S)(M^t S)^T\right) \geq 0.99$.*

To prove Lemma H.5, we need the following lemma.

**Lemma H.6** (Lemma 21 in Gluch et al. (2021)). *Let $A \in \mathbb{R}^{n \times n}$ be a matrix. Let $b = \max_{\ell \in \{1,\ldots,n\}} \|(A\mathbb{1}_\ell)(A\mathbb{1}_\ell)^T\|_2$. Let $0 < \xi < 1$. Let $s \geq \frac{40 n^2 b^2 \log n}{\xi^2}$. Let $I_S = \{s_1, \ldots, s_s\}$ be a multiset of $s$ indices chosen independently and uniformly at random form $V = \{1, \ldots, n\}$. Let $S \in \mathbb{R}^{n \times s}$ be the matrix whose $i$-th column equals $\mathbb{1}_{s_i}$. Then we have*

$$\Pr\left[\|AA^T - \frac{n}{s}(AS)(AS)^T\|_2 \geq \xi\right] \leq n^{-100}.$$

*Proof of Lemma H.5.* **Item 1.** The proof follows directly from the proof of item 2 of Lemma 28 in Gluch et al. (2021).

**Item 2.** Let $A = (M^t)(M^t)^T = M^{2t}$, we get $v_2(A) = v_2(M)^{2t}$. Since $G$ is the disjoint union of two identical $\varphi$-expanders, $G$ has two connected components. Therefore, the normalized Laplacian matrix $L$ of $G$ has two smallest eigenvalues equal to $0$. Consequently, since $M = I - \frac{L}{2}$, the two largest eigenvalues of $M$ are $1 - \frac{0}{2} = 1$. Thus, $v_2(A) = 1$.

Let $\widetilde{A} = \frac{n}{s} \cdot (M^t S)(M^t S)^T$. By Item 2 in Lemma H.2, we have $b = \|(M^t \mathbb{1}_x)(M^t \mathbb{1}_x)^T\|_2 \leq \|M^t \mathbb{1}_x\|_2^2 \leq \frac{3}{n}$. Let $\xi = \frac{1}{100}$. Therefore, for a large enough constant $c > 1$, we have $s = c \cdot \log n \geq \frac{40 n^2 b^2 \log n}{(\frac{1}{100})^2}$. Thus, according to Lemma H.6, we get that with probability at least $1 - n^{-100}$,

$$\|A - \widetilde{A}\|_2 \leq \frac{1}{100}.$$

By Weyl's inequality (Lemma E.9), we get that $v_2(\widetilde{A}) \geq v_2(A) - \|\widetilde{A}\|_2 \geq 1 - \frac{1}{100} = 0.99.$    $\square$

The proof of Lemma H.7 follows directly from the proof of Lemma 24 in Gluch et al. (2021). Nevertheless, for the sake of completeness, we provide a concise proof here.

**Lemma H.7** (Expander related version of Lemma E.6). *Let $\varphi \in (0, 1)$. Let $G = (V, E)$ be a $d$-regular graph. Let $I_S = \{s_1, \ldots, s_s\}$ be a multiset of $s$ indices chosen independently and uniformly at random form $V = \{1, \ldots, n\}$. Let $\mathcal{G} \in \mathbb{R}^{s \times s}$ be the output of $\text{ESTCOLLIPROB}\,(G, R, t, M, I_S)$ (Alg. 2). Let $c_1 > 1$ be a large enough constant. For any $t \geq \frac{20 \log n}{\varphi^2}$, if $R \geq \frac{c_1 \cdot n}{M}$ and $1 \leq M \leq O\left(n^{1/2}\right)$, then with probability at least $1 - 2 \cdot n^{-100}$,*

    *1 if $G$ is a $\varphi$-expander of size $n$ and $s \geq 1$, then $v_2\left(\left(\frac{n}{s}\mathcal{G}\right)^2\right) = \left(v_2(\frac{n}{s}\mathcal{G})\right)^2 < 0.001$,*

    *2 if $G$ is the disjoint union of two identical $\varphi$-expanders of size $n/2$ and $s \geq c_2 \cdot \log n$, where $c_2 > 1$ is a large enough constant, then $v_2\left(\left(\frac{n}{s}\mathcal{G}\right)^2\right) = \left(v_2(\frac{n}{s}\mathcal{G})\right)^2 > 0.95$.*

*Proof.* Let $M$ be the random walk transition matrix of $G$. Let $S \in \mathbb{R}^{n \times s}$ be the matrix whose $i$-th column equals $\mathbb{1}_{s_i}$. Let $\sqrt{\frac{n}{s}} \cdot M^t S = \widetilde{U}\widetilde{\Sigma}\widetilde{W}^T$ be an SVD of $\sqrt{\frac{n}{s}} \cdot M^t S$ where $\widetilde{U} \in \mathbb{R}^{n \times n}, \widetilde{\Sigma} \in \mathbb{R}^{n \times n}, \widetilde{W} \in \mathbb{R}^{s \times n}$. Let $\frac{n}{s} \cdot \mathcal{G} = \widehat{W}\widehat{\Sigma}\widehat{W}^T$ be an eigendecomposition of $\frac{n}{s} \cdot \mathcal{G}$.

**Item 1.** Let $\sigma_{\text{err}} = \frac{0.0001}{n}$. Let $c$ be the constant from Lemma H.4. By the assumption of the lemma, we have

$$R = \frac{c_1 \cdot n}{M} \geq \frac{c \cdot 10^8 \cdot n}{M} = \frac{c \cdot n^{-1}}{\sigma_{\text{err}}^2 M}.$$

Thus we can apply Lemma H.4. Hence, with probability at least $1 - n^{-100}$, we have

$$\|\mathcal{G} - (M^t S)^T (M^t S)\|_2 \leq s \cdot \sigma_{\text{err}}.$$

Let $\widetilde{A} = \frac{n}{s} \cdot (M^t S)^T (M^t S) = \widetilde{W}\widetilde{\Sigma}^2\widetilde{W}^T$ and $\widehat{A} = \frac{n}{s} \cdot \mathcal{G}$. Thus, we have $\widetilde{A}^2 = \left(\frac{n}{s} \cdot (M^t S)^T (M^t S)\right)^2 = \widetilde{W}\widetilde{\Sigma}^4\widetilde{W}^T$ and $\widehat{A}^2 = \left(\frac{n}{s} \cdot \mathcal{G}\right)^2 = \widehat{W}\widehat{\Sigma}^2\widehat{W}^T$. Moreover, we have

$\|\widetilde{A}^2 - \widehat{A}^2\|_2 = \left(\frac{n}{s}\right)^2 \|\left((M^t S)^T (M^t S)\right)^2 - \mathcal{G}^2\|_2$. Using the triangle inequality and sub-multiplicativity of spectral norm and the above $\|\mathcal{G} - (M^t S)^T (M^t S)\|_2 \leq s \cdot \sigma_{\text{err}}$ bound, we can get that

$$\|\left((M^t S)^T (M^t S)\right)^2 - \mathcal{G}^2\|_2 \leq (s \cdot \sigma_{\text{err}})^2 + 2 \cdot s \cdot \sigma_{\text{err}}\|(M^t S)^T (M^t S)\|_2.$$

Note that $\|(M^t S)^T (M^t S)\|_2 \leq \|(M^t S)^T (M^t S)\|_F = \sqrt{\sum_{i=1}^s \sum_{j=1}^s ((M^t \mathbb{1}_{s_i})^T (M^t \mathbb{1}_{s_j}))^2}$, by Cauchy Schwarz inequality and Item 1 of Lemma H.2, we can get that $\|(M^t S)^T (M^t S)\|_2 \leq s \cdot \frac{2}{n}$. Put them together and by the choice of $\sigma_{\text{err}} = \frac{0.0001}{n}$, we have that

$$\|\widetilde{A}^2 - \widehat{A}^2\|_2 \leq \left(\frac{n}{s}\right)^2 \cdot \left(s^2 \sigma_{\text{err}}^2 + 2 \cdot s \cdot \sigma_{\text{err}} \cdot s \cdot \frac{2}{n}\right) = n^2 \sigma_{\text{err}}^2 + 4n\sigma_{\text{err}} \leq 0.00005.$$

Moreover, since $s \geq 1$, by Item 1 of Lemma H.5, with probability at least $1 - n^{-100}$, we have

$$v_2\left(\widetilde{A}^2\right) = v_2\left(\left(\frac{n}{s} \cdot (M^t S)^T (M^t S)\right)^2\right) \leq (n^{-9})^2 = n^{-18}.$$

By Weyl's inequality, we have that

$$v_2(\widehat{A}^2) \leq v_2(\widetilde{A}^2) + \|\widetilde{A}^2 - \widehat{A}^2\|_2 \leq n^{-18} + 0.0005 \leq 0.001.$$

**Item 2.** By the same augment of the proof of Item 1 and Item 2 of Lemma H.2, we can get that $\|(M^t S)^T (M^t S)\|_2 \leq s \cdot \frac{3}{n}$. Thus, by the choice of $\sigma_{\text{err}} = \frac{0.0001}{n}$, we have that

$$\|\widetilde{A}^2 - \widehat{A}^2\|_2 \leq \left(\frac{n}{s}\right)^2 \cdot \left(s^2 \sigma_{\text{err}}^2 + 2 \cdot s \cdot \sigma_{\text{err}} \cdot s \cdot \frac{3}{n}\right) = n^2 \sigma_{\text{err}}^2 + 6n\sigma_{\text{err}} \leq 0.0007.$$

Moreover, since $s \geq c_2 \cdot \log n$, by Item 2 of Lemma H.5, with probability at least $1 - n^{-100}$, we have

$$v_2\left(\widetilde{A}^2\right) = v_2\left(\left(\frac{n}{s} \cdot (M^t S)^T (M^t S)\right)^2\right) \geq (0.99)^2 > 0.98.$$

By Weyl's inequality, we have that

$$v_2(\widehat{A}^2) \geq v_2(\widetilde{A}^2) - \|\widetilde{A}^2 - \widehat{A}^2\|_2 \geq 0.98 - 0.0007 > 0.95.$$

$\square$

Now we are ready to prove Theorem 1.2.

*Proof of Theorem 1.2.* **Correctness.** By the promise in the theorem statement, the input $d$-regular graph $G = (V, E)$ is guaranteed to be either a $\varphi$-expander or the disjoint union of two identical $\varphi$-expanders, each of size $n/2$. We run algorithm DISTINGUISH$(G, M)$ (Alg. 5) to distinguish the above two cases. Note that the choices of $t$, $s$, and $R$ are made so that all the assumptions required by Lemma H.7 are satisfied. Therefore, by Lemma H.7, we get that in case (i) (when $G$ is a $\varphi$-expander), with probability at least $1 - 2n^{-100}$, $(v_2(\frac{n}{s}\mathcal{G}))^2 < 0.001 < 0.6$; in case (ii), with probability at least $1 - 2n^{-100}$, $(v_2(\frac{n}{s}\mathcal{G}))^2 > 0.95 > 0.6$. Therefore, we get that, with probability at least $1 - 2n^{-100}$, algorithm DISTINGUISH correctly distinguishes which case holds.

**Space and runtime.** According to Lemma H.4, getting matrix $\mathcal{G}$ requires $O(R \cdot t \cdot \log n \cdot s^2)$ time and $O(M \cdot \log^2 n \cdot s^2)$ bits of space. Computing $(\frac{n}{s}\mathcal{G})^2$ requires $O(s^3)$ time and $O(s^2 \cdot \log n)$ bits of space. Therefore, the overall runtime and space complexity are $O(R \cdot t \cdot \log n \cdot s^2 + s^3)$ and $O(M \cdot \log^2 n \cdot s^2 + s^2 \log n)$ bits, respectively. By setting $t = \frac{20 \log n}{\varphi^2}$, $R = \Theta(\frac{n}{M})$ and $s = O(\log n)$, we get that DISTINGUISH$(G, M)$ runs in $n \cdot \frac{1}{M} \cdot \text{poly}(\log n) \cdot \frac{1}{\varphi^2}$ time and uses $M \cdot \text{poly}(\log n)$ bits of space.

$\square$

# I  PROOF OF THEOREM 1.3

**Theorem I.1** (Restate of Theorem 1.3). *Any algorithm that correctly solves the $1$-cluster vs. $2$-cluster problem with error at most $1/3$ using only random walk oracles must satisfy $T \cdot S \geq \Omega(n)$, where $T$ and $S$ denote the time complexity and space complexity of the algorithm, respectively.*

Before we start the proof of Theorem 1.3, we would first introduce some basic definitions in information theory.

## I.1  BASIC DEFINITIONS

**Definition I.1** (Entropy). Given a random variable $X$ taking values in the set $\mathcal{X}$ and distributed according to $p : \mathcal{X} \to [0, 1]$, the *entropy* of $X$ is defined as

$$H(X) := -\sum_{x \in \mathcal{X}} p(x) \log p(x).$$

In the special case where $X$ has only two possible outcomes, the entropy is given by

$$H_2(X) := -p \log p - (1 - p) \log(1 - p).$$

The entropy of a random variable quantifies the average level of uncertainty or information associated with the random variable. Note that for the special case of $H_2$, we have the following property:

**Lemma I.1.**

$$1 - H_2\left(\frac{1}{2} + a\right) = \frac{1}{2 \ln 2} \sum_{l=1}^{\infty} \frac{(2a)^{2l}}{l(2l - 1)} = O\left(a^2\right).$$

Given the outcome of another random variable $Y$, we can also quantify this randomness using conditional entropy.

**Definition I.2** (Conditional entropy). Given random variables $X$ and $Y$ taking values in sets $\mathcal{X}$ and $\mathcal{Y}$, respectively, with joint distribution $p : \mathcal{X} \times \mathcal{Y} \to [0, 1]$, the *conditional entropy* of $X$ given $Y$ is defined as

$$H(X \mid Y) = H(X, Y) - H(Y) = -\sum_{x \in \mathcal{X}, y \in \mathcal{Y}} p(x, y) \log \frac{p(x, y)}{p(y)}.$$

Furthermore, the amount of information that is shared between two random variables is called mutual information.

**Definition I.3** (Mutual Information). Given random variables $X$ and $Y$ taking values in $\mathcal{X}$ and $\mathcal{Y}$, respectively, the *mutual information* between $X$ and $Y$ is defined as

$$I(X; Y) = H(X) - H(X \mid Y) = H(Y) - H(Y \mid X).$$

Similarly, given a random variable $Z$ taking values in $\mathcal{Z}$, the *conditional mutual information* of $X$ and $Y$ given $Z$ is defined as

$$I(X; Y \mid Z) = H(X \mid Z) - H(X \mid Y, Z).$$

Our proof will also use the following key properties of mutual information.

**Lemma I.2** (Data Processing Inequality). *Given random variables $X, Y$ and $Z$ taking values in sets $\mathcal{X}, \mathcal{Y}$ and $\mathcal{Z}$, respectively, such that $X \perp Z \mid Y$. Then*

$$I(X; Z) \leq I(X; Y).$$

**Lemma I.3** (Chain Rule). *Given random variables $X, Y$ and $Z$ taking values in sets $\mathcal{X}, \mathcal{Y}$ and $\mathcal{Z}$, respectively, we have*

$$I(X; Y, Z) = I(X; Z) + I(X; Y \mid Z).$$

### I.2 HARD INSTANCE I

To prove Theorem 1.3, we first consider the following Hard Instance, inspired by Diakonikolas et al. (2019) and commonly used in uniformity testing. Note that in our construction, at each time $t$, the player is allowed to pick a $W_t \in [2n]$. The proof of Theorem I.2 then follows from the proof of Theorem 23 in Diakonikolas et al. (2019).

**Definition I.4** (Hard Instance I). Let $X$ be a uniformly random bit. Based on $X$, the adversary chooses the distribution $p$ on $[2n]$ bins as follows:

- $X = 0$ : Pick $p = U_{2n}$, where $U_{2n}$ is the uniform distribution on $[2n]$.
- $X = 1$ : We construct two sets as follows: Pair the bins as $\{1, 2\}, \{3, 4\}, \cdots, \{2n-1, 2n\}$. Now on each pair $\{2i-1, 2i\}$ pick a random $Y_i \in \{\pm 1\}$. If $Y_i = 1$, we put bin $2i - 1$ to set 1 and bin $2i$ to set 2; otherwise, we put bin $2i$ to set 1 and bin $2i - 1$ to set 2. Each time, the player picks $W_t \in [2n]$. If $W_t$ belongs to set 1, we have $Z_t = 1$; otherwise, $Z_t = -1$. The distribution is then

$$(p_{2i-1}, p_{2i}) = \left( \frac{1 + Y_i Z_t}{2n}, \frac{1 - Y_i Z_t}{2n} \right).$$

We have the space-time tradeoff of this instance to be

**Theorem I.2.** *Let $\mathcal{A}$ be an algorithm that detects the Hard Instance I with error at most $1/3$. The algorithm can access the samples in a single-pass streaming fashion using $M$ bits of space and $T$ samples. Furthermore, at each step, the algorithm may choose which set to sample by specifying $W_t$. We then have $T \cdot M = \Omega(n)$.*

**Remark I.1.** *In Theorem I.2, we use $M$ to denote the space complexity because $S$ is already used in the proof to refer to a sampling-related quantity. For consistency with the rest of the paper, we will **denote the space of the algorithm by $S$** in subsequent discussions.*

*Proof of Theorem I.2.* In either case, we can think of the output of $p$ as being a pair $(C, V)$, where $C$ is an element of $[n]$ is chosen uniformly, and $V \in \{0, 1\}$ is a fair coin if $X = 0$ and has bias $Y_C Z_t$ if $X = 1$.

Let $s_1, \ldots, s_T$ be the observed samples from $p$. Let $M_t$ denote the bits stored in the memory after the algorithm sees the $t$-th sample $s_t$.

Since the algorithm $\mathcal{A}$ learns $X$ with probability at least $2/3$ after viewing $T$ samples, we know that $I(X; M_T) > \Omega(1)$. On the other hand, $M_t$ is computed from $(M_{t-1}, s_t)$ without using any information about $X$. More formally, $X \perp M_t \mid (M_{t-1}, s_t)$ and therefore we can use the data processing inequality (Lemma I.2) and chain rule (Lemma I.3) to get:

$$I(X; M_t) \leq I(X; M_{t-1}, s_t) = I(X; M_{t-1}) + I(X; s_t \mid M_{t-1}).$$

Since irrespective of $X, C$ is uniform over the pairs of bins, we note that $C$ is independent of $X$ even when conditioned on the memory $M$. Moreover, player's choice of $W_t$ is computed only from $M_{t-1}$. Thus,

$$I(X; s_t \mid M_{t-1}) = I(X; C_t V_t \mid M_{t-1}) = I(X; V_t \mid M_{t-1} C_t) = I(X; V_t \mid M_{t-1} C_t W_t).$$

Let $\alpha_{t-1} = \Pr[X = 1 \mid M_{t-1} C_t W_t]$ and thus $\Pr[X = 0 \mid M_{t-1} C_t W_t] = 1 - \alpha_{t-1}$.

We have that

$$\Pr\left[V_t = 0 \mid X = 0, M_{t-1}, C_t, W_t\right] = \frac{1}{2},$$

$$\Pr\left[V_t = 0 \mid X = 1, M_{t-1}, C_t, Z_t\right] = \frac{1 + \mathbb{E}\left[Z_t Y_{C_t} \mid M_{t-1}, W_t\right]}{2},$$

$$\Pr\left[V_t = 0 \mid M_{t-1}, C_t\right] = (1 - \alpha_{t-1})\frac{1}{2} + \alpha_{t-1}\frac{1 + \mathbb{E}\left[Z_t Y_{C_t} \mid M_{t-1}, W_t\right]}{2}$$

$$= \frac{1}{2} + \frac{\alpha_{t-1}\mathbb{E}\left[Z_t Y_{C_t} \mid M_{t-1}, W_t\right]}{2}.$$

We can calculate

$$I\left(X; V_t \mid M_{t-1}C_t W_t\right) = H\left(V_t \mid M_{t-1}C_t W_t\right) - H\left(V_t \mid M_{t-1}C_t W_t X\right)$$

$$= H_2\left(\Pr\left[V_t = 0 \mid M_{t-1}, C_t, W_t\right]\right)$$

$$\quad - \{\Pr\left[X = 1 \mid M_{t-1}C_t W_t\right] H_2\left(\Pr\left[V_t = 0 \mid X = 1, M_{t-1}, C_t, W_t\right]\right)$$

$$\quad + \Pr\left[X = 0 \mid M_{t-1}C_t W_t\right] H_2\left(\Pr\left[V_t = 0 \mid X = 0, M_{t-1}, C_t, W_t\right]\right)\}$$

$$= H_2\left(\frac{1}{2} + \frac{\alpha_{t-1}\mathbb{E}\left[Z_t Y_{C_t} \mid M_{t-1}, W_t\right]}{2}\right) - \alpha_{t-1}H_2\left(\frac{1}{2} + \frac{\mathbb{E}\left[Z_t Y_{C_t} \mid M_{t-1}, W_t\right]}{2}\right)$$

$$\quad - (1 - \alpha_{t-1}) H_2\left(\frac{1}{2}\right)$$

$$= \alpha_{t-1}\left[1 - H_2\left(\frac{1}{2} + \frac{\mathbb{E}\left[Z_t Y_{C_t} \mid M_{t-1}, W_t\right]}{2}\right)\right] - \left[1 - H_2\left(\frac{1}{2} + \frac{\alpha_{t-1}\mathbb{E}\left[Z_t Y_{C_t} \mid M_{t-1}, W_t\right]}{2}\right)\right]$$

$$= \Theta(1)\left[\alpha_{t-1}\left(\frac{\mathbb{E}\left[Z_t Y_{C_t} \mid M_{t-1}, W_t\right]}{2}\right)^2 - \left(\frac{\alpha_{t-1}\mathbb{E}\left[Z_t Y_{C_t} \mid M_{t-1}, W_t\right]}{2}\right)^2\right]$$

$$= \Theta(1)\alpha_{t-1}(1 - \alpha_{t-1})\mathbb{E}\left[Z_t Y_{C_t} \mid M_{t-1}, W_t\right]^2$$

$$\leq O(1)\mathbb{E}\left[Z_t Y_{C_t} \mid M_{t-1}, W_t\right]^2.$$

Since $C_t$ is uniformly random, we have that

$$I\left(X; V_t \mid M_{t-1}C_t W_t\right) = \frac{1}{n} \cdot \sum_{j=1}^{n} O(1)\mathbb{E}\left[Z_t Y_j \mid M_{t-1}, W_t\right]^2.$$

Now to bound this part, note that we first have $H\left(M_{t-1}, W_t\right) \leq M$ that $I\left(Z_t Y_1 \ldots Z_t Y_n; M_{t-1}, W_t\right) \leq M$. At the same time, notice that $Z_t$ is just flipping the value of $Y_1, \ldots, Y_n$ and thus $H\left(Z_t Y_1 \ldots Z_t Y_n\right) = H\left(Y_1 \ldots Y_n\right) = n$. Thus we have

$$H\left(Z_t Y_1 \ldots Z_t Y_n \mid M_{t-1}, W_t\right) = H\left(Z_t Y_1 \ldots Z_t Y_n\right) - I\left(Z_t Y_1 \ldots Z_t Y_n; M_{t-1}, W_t\right) \geq n - M.$$

On the other hand, we have that

$$\sum_{i=1}^{n} H\left(Z_t Y_i \mid M_{t-1}, W_t\right) \geq H\left(Z_t Y_1 \ldots Z_t Y_n \mid M_{t-1}, W_t\right) \geq n - M.$$

Thus,

$$M \geq \sum_{i=1}^{n}\left[1 - H\left(Z_t Y_i \mid M_{t-1}, W_t\right)\right] = \Theta\left(\sum_{i=1}^{n}\mathbb{E}\left[Z_t Y_i \mid M_{t-1}, W_t\right]^2\right),$$

where the equality comes from the fact that if $\Pr\left[Z_t Y_i = 1 \mid M_{t-1}, W_t\right] = \frac{1}{2} + \beta$, then

$$\mathbb{E}\left[Z_t Y_i \mid M_{t-1}, W_t\right] = \Pr\left[Z_t Y_i = 1 \mid M_{t-1}, W_t\right](+1) + \Pr\left[Z_t Y_i = -1 \mid M_{t-1}, W_t\right](-1)$$

$$= \left(\frac{1}{2} + \beta\right) - \left(\frac{1}{2} - \beta\right) = 2\beta.$$

We finally have that

$$
\begin{aligned}
\Omega(1) \leq I\left(M_T; X\right) &= \sum_{t=0}^{T-1} I\left(M_{t+1}; X\right) - I\left(M_t; X\right) \\
&= \sum_{t=0}^{T-1} I\left(M_t, S_{t+1}; X\right) - I\left(M_t; X\right) \\
&= \sum_{t=0}^{T-1} I\left(S_{t+1}; X \mid M_t\right) \\
&= \sum_{t=0}^{T-1} I\left(V_{t+1}; X \mid M_t, C_{t+1}, W_{t+1}\right) \\
&= O(1)\frac{T \cdot M}{n}.
\end{aligned}
$$

We conclude that $T \cdot M \geq \Omega(n)$. $\qquad \square$

## I.3  HARD INSTANCE II

For the graph problems, we would consider the following Hard Instance.

**Definition I.5** (Hard Instance II). *Let $X$ be a uniformly random bit. Let $\varphi \in (0,1)$ with $\varphi = \Omega(1)$, and let $d = O(1)$. Based on $X$, the adversary chooses a $d$-regular graph $G$ on $2n$ vertices as follows:*

- *$X = 0$ : Pick the graph to be a $\varphi$-expander on $2n$ vertices.*
- *$X = 1$ : We construct two sets as follows: Pair bins the as $\{1,2\}, \{3,4\}, \cdots, \{2n-1, 2n\}$. Now on each pair $\{2i-1, 2i\}$ pick a random $Y_i \in \{\pm 1\}$. If $Y_i = 1$, we put vertex $2i-1$ to set $1$ and vertex $2i$ to set $2$; otherwise, we put vertex $2i$ to set $1$ and vertex $2i-1$ to set $2$. The graph is then composed of two identical $\varphi$-expanders over set $1$ and set $2$.*

We would assume that the algorithm has access to the graph only via the random walk queries.

**Definition I.6** (Random walk queries). For any specified starting vertex $x$, a random walk query returns the endpoint of an $O(\log n)$-step random walk starting from $x$.

We have the properties of a random walk for a $\varphi$-expander as follows:

**Lemma I.4.** *Assume $G = (V, E)$ is a $d$-regular $\varphi$-expander on $n$ vertices. Let $M$ be the lazy random walk transition matrix of $G$. Let $M^t \mathbb{1}_x$ be the probability distribution of a random walk with length $O(\frac{\log n}{\varphi^2})$ starting from vertex $x \in V$. Let $\pi = (\frac{1}{n}, \ldots, \frac{1}{n})^T \in \mathbb{R}^n$ be the uniform distribution over $n$ vertices. We have that $d_{\mathrm{TV}}(M^t \mathbb{1}_x, \pi) \leq \frac{0.01}{n^2}$.*

To prove Lemma I.4, we first introduce the definition of mixing time.

**Definition I.7** (Mixing time). Let $G = (V, E)$ be a $d$-regular graph on $n$ vertices. Let $M$ be the lazy random walk transition matrix of $G$. Let $m_t = M^t m_0$, where $m_0$ is a distribution over $[n]$. Let $\pi = (\frac{1}{n}, \ldots, \frac{1}{n})^T$ be the stationary distribution of $G$. Then the *mixing time* $\tau_\varepsilon(M)$ is defined to be the smallest $t$ such that for any $m_0$, $d_{\mathrm{TV}}(m_x, \pi) \leq \varepsilon$.

*Proof of Lemma I.4.* Note that $\pi = (\frac{1}{n}, \ldots, \frac{1}{n})^T \in \mathbb{R}^n$ is the stationary distribution of $G$. According to spectral graph theory, we have $\tau_\varepsilon(M) = O(\frac{1}{\phi(G)^2}) \log(\frac{n}{\varepsilon})$. Let $\varepsilon = \frac{0.01}{n^2}$. Note that $G$

is a $\varphi$-expander, we have that $\phi(G) = \varphi$ (see Definition 1.1). Therefore, according to the definition of mixing time, we get that for $t = \tau_\varepsilon(M) = O(\frac{1}{\varphi^2}\log(\frac{n}{\frac{0.01}{n^2}})) = O(\frac{\log n}{\varphi^2})$, we have that $d_{\mathrm{TV}}(M^t \mathbb{1}, \pi) \leq \frac{0.01}{n^2}$. $\qquad\square$

With the above results, we would show the space-time trade-off of identifying Hard Instance II.

**Theorem I.3** (Variant of Theorem 1.3). *Let $\mathcal{A}$ be an algorithm which detects the Hard Instance II with error probability at most $1/3$. The algorithm can perform $T$ random walk queries using $M$ bits of space. We have $M \cdot T = \Omega(n)$.*

**Remark I.2.** *In Theorem I.3, we use $M$ to denote the space complexity because $S$ is already used in the proof to refer to a sampling-related quantity. For consistency with the rest of the paper, we will **denote the space of the algorithm by $S$** in subsequent discussions.*

*Proof of Theorem I.3.* We would reduce this problem to the Hard Instance I. Assume we have an algorithm $\mathcal{A}$ that solves the Hard Instance II. We would show how it can be used to solve Hard Instance I. At each time, the algorithm would choose to make a random walk query starting from vertex $i$. We would then set $W_t$ to the Hard Instance I and get the feedback sample $s_t$. We would feed $s_t$ to the algorithm $\mathcal{A}$ and then to the next round. Finally, after $T$ rounds, we would output the results of $\mathcal{A}$.

To prove the correctness, we need to show that the total variation distance is $O(1)$ between the history generated by Hard Instance I: $(s_1, m_1, \ldots, s_T, m_T)$ and the history generated by Hard Instance II: $(s_1', m_1', \ldots, s_T', m_T')$. We would prove by math induction.

Now for $d_{\mathrm{TV}}((m_t, s_t), (m_t', s_t'))$, we consider any fixed $x \in [2n], m \in [M]$ that

$$
\begin{aligned}
&|p(m_t = m, s_t = x) - p(m_t' = m, s_t' = x)| \\
&= \Big| \sum_{(\widetilde{m}, \widetilde{x})} p(m_t = m, s_t = x | m_{t-1} = \widetilde{m}, s_{t-1} = \widetilde{x}) \cdot (m_{t-1} = \widetilde{m}, s_{t-1} = \widetilde{x}) \\
&\quad - \sum_{(\widetilde{m}, \widetilde{x})} p(m_t' = m, s_t' = x | m_{t-1}' = \widetilde{m}, s_{t-1}' = \widetilde{x}) \cdot p(m_{t-1}' = \widetilde{m}, s_{t-1}' = \widetilde{x}) \Big| \\
&\leq \Big| \sum_{(\widetilde{m}, \widetilde{x})} p(m_t = m, s_t = x | m_{t-1} = \widetilde{m}, s_{t-1} = \widetilde{x}) \\
&\quad \cdot \big( p(m_{t-1} = \widetilde{m}, s_{t-1} = \widetilde{x}) - p(m_{t-1}' = \widetilde{m}, s_{t-1}' = \widetilde{x}) \big) \Big| \\
&\quad + \Big| \sum_{(\widetilde{m}, \widetilde{x})} p(m_{t-1}' = \widetilde{m}, s_{t-1}' = \widetilde{x}) \\
&\quad \cdot \big( p(m_t = m, s_t = x | m_{t-1} = \widetilde{m}, s_{t-1} = \widetilde{x}) - p(m_t' = m, s_t' = x | m_{t-1}' = \widetilde{m}, s_{t-1}' = \widetilde{x}) \big) \Big|.
\end{aligned}
$$

Now for the first part, we have

$$
\begin{aligned}
&\sum_{(m,x)} \Big| \sum_{(\widetilde{m}, \widetilde{x})} p(m_t = m, s_t = x | m_{t-1} = \widetilde{m}, s_{t-1} = \widetilde{x}) \\
&\quad \cdot \big( p(m_{t-1} = \widetilde{m}, s_{t-1} = \widetilde{x}) - p(m_{t-1}' = \widetilde{m}, s_{t-1}' = \widetilde{x}) \big) \Big| \\
&\leq \sum_{(m,x)} \sum_{(\widetilde{m}, \widetilde{x})} \Big( p(m_t = m, s_t = x | m_{t-1} = \widetilde{m}, s_{t-1} = \widetilde{x}) \\
&\quad \cdot \big| p(m_{t-1} = \widetilde{m}, s_{t-1} = \widetilde{x}) - p(m_{t-1}' = \widetilde{m}, s_{t-1}' = \widetilde{x}) \big| \Big) \\
&= \sum_{(\widetilde{m}, \widetilde{x})} \Big( \big| p(m_{t-1} = \widetilde{m}, s_{t-1} = \widetilde{x}) - p(m_{t-1}' = \widetilde{m}, s_{t-1}' = \widetilde{x}) \big|
\end{aligned}
$$

$$\cdot \sum_{(m,x)} p(m_t = m, s_t = x | m_{t-1} = \widetilde{m}, s_{t-1} = \widetilde{x}) \Big)$$

$$= \sum_{(\widetilde{m}, \widetilde{x})} \Big| p(m_{t-1} = \widetilde{m}, s_{t-1} = \widetilde{x}) - p(m'_{t-1} = \widetilde{m}, s'_{t-1} = \widetilde{x}) \Big|$$

$$= 2 d_{\text{TV}}((m_{t-1}, s_{t-1}), (m'_{t-1}, s'_{t-1})).$$

For the second part, we notice that

$$p(m_t = m, s_t = x | m_{t-1} = \widetilde{m}, s_{t-1} = \widetilde{x}) - p(m'_t = m, s'_t = x | m'_{t-1} = \widetilde{m}, s'_{t-1} = \widetilde{x})$$
$$= p(m_t = m | s_t = x, m_{t-1} = \widetilde{m}, s_{t-1} = \widetilde{x}) \cdot p(s_t = x | m_{t-1} = \widetilde{m}, s_{t-1} = \widetilde{x})$$
$$- p(m'_t = m | s'_t = x, m'_{t-1} = \widetilde{m}, s'_{t-1} = \widetilde{x}) \cdot p(s'_t = x | m'_{t-1} = \widetilde{m}, s'_{t-1} = \widetilde{x}).$$

Note that since we are using the same algorithm, when fixing $m_{t-1}$ and $s_t$, the update of $m_t$ and $m'_t$ is the same, and thus

$$p(m_t = m, s_t = x | m_{t-1} = \widetilde{m}, s_{t-1} = \widetilde{x}) - p(m'_t = m, s'_t = x | m'_{t-1} = \widetilde{m}, s'_{t-1} = \widetilde{x})$$
$$= p(m_t = m | s_t = x, m_{t-1} = \widetilde{m}) \cdot \big( p(s_t = x | m_{t-1} = \widetilde{m}) - p(s'_t = x | m'_{t-1} = \widetilde{m}) \big).$$

Moreover, by the property of lazy random walk (Lemma I.4), we should have that for any $\widetilde{m}$,

$$\frac{1}{2} \sum_x \big| p(s_t = x | m_{t-1} = \widetilde{m}) - p(s'_t = x | m'_{t-1} = \widetilde{m}) \big| \leq \frac{0.01}{n^2}.$$

Summing over all $(m, x)$, we have the second part is bounded by

$$\sum_{(m,x)} \Big| \sum_{(\widetilde{m}, \widetilde{x})} p(m'_{t-1} = \widetilde{m}, s'_{t-1} = \widetilde{x}) \cdot p(m_t = m | s_t = x, m_{t-1} = \widetilde{m})$$

$$\cdot \big( p(s_t = x | m_{t-1} = \widetilde{m}) - p(s'_t = x | m'_{t-1} = \widetilde{m}) \big) \Big|$$

$$\leq \sum_{(m,x,\widetilde{m},\widetilde{x})} p(m'_{t-1} = \widetilde{m}, s'_{t-1} = \widetilde{x}) \cdot p(m_t = m | s_t = x, m_{t-1} = \widetilde{m})$$

$$\cdot \big| p(s_t = x | m_{t-1} = \widetilde{m}) - p(s'_t = x | m'_{t-1} = \widetilde{m}) \big|$$

$$= \sum_{(x,\widetilde{m},\widetilde{x})} p(m'_{t-1} = \widetilde{m}, s'_{t-1} = \widetilde{x}) \big| p(s_t = x | m_{t-1} = \widetilde{m}) - p(s'_t = x | m'_{t-1} = \widetilde{m}) \big|$$

$$\cdot \sum_m p(m_t = m | s_t = x, m_{t-1} = \widetilde{m})$$

$$= \sum_{(x,\widetilde{m},\widetilde{x})} p(m'_{t-1} = \widetilde{m}, s'_{t-1} = \widetilde{x}) \big| p(s_t = x | m_{t-1} = \widetilde{m}) - p(s'_t = x | m'_{t-1} = \widetilde{m}) \big|$$

$$= \sum_{(\widetilde{m},\widetilde{x})} p(m'_{t-1} = \widetilde{m}, s'_{t-1} = \widetilde{x}) \cdot \sum_x \big| p(s_t = x | m_{t-1} = \widetilde{m}) - p(s'_t = x | m'_{t-1} = \widetilde{m}) \big|$$

$$\leq 2 \times \frac{0.01}{n^2} \sum_{(\widetilde{m},\widetilde{x})} p(m'_{t-1} = \widetilde{m}, s'_{t-1} = \widetilde{x})$$

$$= 2 \times \frac{0.01}{n^2}.$$

Combining the results, we have

$$d_{\text{TV}}((m_t, s_t), (m'_t, s'_t)) = \frac{1}{2} \sum_{(m,x)} |p(m_t = m, s_t = x) - p(m'_t = m, s'_t = x)|$$

$$\leq d_{\mathrm{TV}}((m_{t-1}, s_{t-1}), (m'_{t-1}, s'_{t-1})) + \frac{0.01}{n^2}.$$

Moreover, for the initial points, we have that

$$d_{\mathrm{TV}}(s_1, s'_1) \leq \frac{0.01}{n^2}.$$

Since $m_1, m'_1$ are merely a function of $s_1, s'_1$, we have that

$$d_{\mathrm{TV}}(m_1, m'_1) \leq \frac{0.01}{n^2}.$$

Therefore

$$d_{\mathrm{TV}}((m_1, s_1), (m_1', s'_1)) \leq d_{\mathrm{TV}}(s_1, s'_1) + d_{\mathrm{TV}}(m_1, m'_1) \leq \frac{0.02}{n^2},$$

$$d_{\mathrm{TV}}((m_t, s_t), (m_t', s'_t)) \leq \frac{0.01(1+t)}{n^2}.$$

This means that

$$d_{\mathrm{TV}}(m_T, m'_T) \leq d_{\mathrm{TV}}((m_T, s_T), (m_T', s'_T)) \leq \frac{0.01(1+T)}{n^2} \leq 0.01,$$

where we use the fact that $T \leq O(n^2)$ since otherwise we can get the output using constant space.

Now note that the output result is only the function of $m_T$. Since the total variation distance of $m_T$ is bounded, the correctness can still be guaranteed using the uniform distribution rather than the random walk distribution. $\square$

## J EXPERIMENTAL DETAILS

**Accuracy**  Let $C_1, \ldots, C_k$ be the ground-truth clustering and let $\widehat{C}_1, \ldots, \widehat{C}_k$ be the clusters produced by the oracle, where $\widehat{C}_i = \{x \in V | \mathrm{WHICHCLUSTER}(G, x) = i\}$. The accuracy is defined as $\frac{1}{n} \cdot \max_\pi \sum_{i=1}^k |C_i \cap \widehat{C}_{\pi(i)}|$, where $\pi : [k] \to [k]$ is a permutation.

**Implementation details**  In our experiments, we implemented three main components: (i) the new dot product oracle proposed in this paper (Alg. 3 and Alg. 4), (ii) the original dot product oracle in Gluch et al. (2021), and (iii) the spectral clustering oracle relies on a poly($k$) conductance gap itself. The clustering oracle relies on accurate dot product estimates to function correctly; hence, we first needed to identify parameters that ensure reliable dot product estimation performance. These parameters include (i) $s_{\mathrm{dot}}$, the number of sampled vertices in dot product oracle, (ii) $t$, the random walk length and (iii) $l$, the number of repetitions in the median trick, and a set of space-time-related parameters.

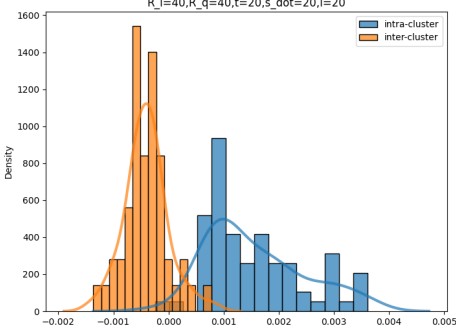 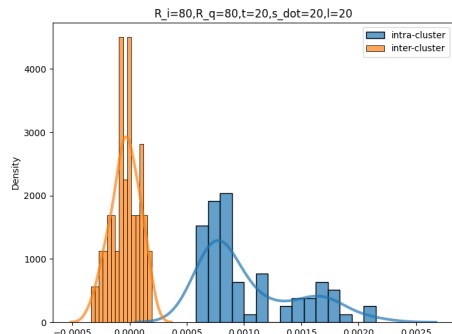

(a) unsuitable parameter values: $R_{\mathrm{init}} = R_{\mathrm{query}} = 40$    (b) suitable parameter values: $R_{\mathrm{init}} = R_{\mathrm{query}} = 80$

Figure 2: Effect of parameter settings on the original dot product oracle. (a): an unsuitable configuration where the estimated spectral dot products for intra-cluster and inter-cluster pairs overlap. (b): a suitable configuration where a clear gap emerges between the two distributions.

For the original dot product oracle in Gluch et al. (2021), $R_{\text{init}}, R_{\text{query}}$ are the space-time-related parameters. We set $R_{\text{init}}$ and $R_{\text{query}}$ according to the theoretical guarantee, which states that the oracle works when $R_{\text{init}} = R_{\text{query}} = O(\sqrt{n})$. Following the implementation details in Shen & Peng (2023), we explored multiple parameter configurations for $s_{\text{dot}}, t, l, R_{\text{init}} = R_{\text{query}}$. For each configuration, we initialized the dot product oracle with the corresponding parameters, sampled a subset of vertex pairs, computed their estimated spectral dot products, and plotted the density graphs (see Figure 2). The presence of a clear gap (see Figure 2b) in the density graph was used as the criterion for selecting suitable parameter values. In fact, for a graph with parameters $n = 3000$, $k = 3$, $p = 0.07$, and $q = 0.002$, we found that $s_{\text{dot}} = 20$, $t = 20$, $l = 20$, and $R_{\text{init}} = R_{\text{query}} \geq 80$ provided reliable estimates. And we make $80 \times 80$ a concrete instantiation of $O(\sqrt{n}) \times O(\sqrt{n}) = O(n)$.

For the new dot product oracle, we set $s_{\text{dot}} = 20, t = 20$ and $l = 20$ like above. The space-time-related parameters $M_{\text{init}} = M_{\text{query}}$ serve as inputs, corresponding to $R_{\text{init}}^{\text{our}} = R_{\text{query}}^{\text{our}} = \frac{80 \times 80}{M_{\text{init}}} = \frac{6400}{M_{\text{init}}}$ (see line 2 of Alg. 3 and Alg. 4). In our experiments, we varied $M_{\text{init}} = M_{\text{query}}$ in the range $[30, 80]$.

Finally, for the clustering oracle itself, we determined the number of sampled vertices $s$ (see line 3 of Alg. 12) through extensive testing of multiple candidate values, and selected $s = 21$ for all experiments. Additionally, we set a threshold $\theta$ (see line 8 of Alg. 12) to construct similarity graph; based on the density plots of estimated dot products (see Figure 2b), we chose $\theta \approx 0.0005$.

