# OpenReview forum: "Sublinear Spectral Clustering Oracle with Little Memory"
_ICLR.cc/2026/Conference — ICLR 2026 Poster_

### Official Review · Reviewer_S7oN · 2025-10-28

**Soundness:** 3
**Presentation:** 4
**Contribution:** 3
**Rating:** 8
**Confidence:** 3

**Summary:**

This paper considers the task of community detection in well-clusterable graphs with sublinear space. The goal is to design a data structure D that fits in sublinear memory, and that enables one to query the cluster assignment for each node in sublinear time. Previous approaches all require $\Omega(\sqrt{n})$ space for such a datastructure D, but this paper overcomes that. In particular, it is able to design a data structure with a much smaller memory requirement that still allow for sublinear time which-cluster queries. The paper also provides new insights into the time-space tradeoff for this problem, by designing oracles there memory usage S and query time $T$ satisfy $S\cdot T \approx \tilde{O}(n)$. Again, this holds for a class of graphs with good clustering structure. The paper also proves that this is optimal up to logarithmic factors for a certain class of techniques.

The notion of well-clusterable graphs corresponds (roughly) to graphs that have a k partition where clusters are roughly balanced in size and have small conductance (and large inner conductance, which measures internal connectivity of clusters).

The paper also proves new results (sublinear algorithms and lower bounds) for the 1-cluster/2-cluster problem, which seeks to tell the difference between graphs that are expenders on n nodes or that are disjoint unions of two identical expanders on n/2 nodes.

For the clustering results, the key technical advance is to provide a new way to estimate the dot product between the spectral embedding of two nodes in sublinear space and time (the spectral embedding for a node comes from the node's entries in the first few eigenvectors of the normalized Laplacian). This primitive is combined with a previously observation that if two nodes are from the same cluster (under the well-clusterability assumptions), then the dot product of their embeddings with be large, and otherwise the dot product will be zero.

**Strengths:**

* This paper provides good motivation and justification for sublinear time and space algorithms, and tackles and interesting problem in graph analysis.
* The main theoretical result is interesting and represents a substantial improvement over previous techniques for sublinear space clustering oracles. In particular, it breaks the $\Omega(\sqrt{n})$ space barrier limiting previous approaches. The results on algorithms and the lower bound for the 1-cluster/2-cluster problem are a nice bonus addition to the paper. The theoretical results are highly non-trivial.
* The structure and writing of the paper are excellent; about as good as one could hope for a paper whose core contribution is so dense and complicated. The paper did a very good job motivating the problem, explaining the key contributions and their significance, and communicating the main technical components that made them work. I learned a lot by reading this paper, despite how theoretically dense it is. The presentation is very, very good.
* I really appreciate that the authors have even provided an implementation of the algorithm and accompanied their theory with numerical experiments. Even if it's only on synthetic data, it is impressive to have an implementation of any kind for an algorithm that is so detailed and complicated.

**Weaknesses:**

The main downsides of this paper, when considering it for publication at ICLR specifically are: (1) the algorithm is extremely complicated and detailed, and (2) the algorithm is very impractical for real-world clustering problems. I don't this at all disqualifies the article from being accepted and published somewhere. The contribution here seems very impressive. I do wonder though whether ICLR is the right venue for this work, and instead of a core theory conference. For example, the main work this improves upon (Peng 2020; Gluche et al 2021) are both SODA papers.

Expounding more on the two weaknesses above:

(1) Although the writing does an impressive job making the pieces of the algorithm make sense at a high level, there is still no way around how intricate the statements of the results are. Many many different functions and parameters need to be defined---all with complicated dependencies on each other---even just to present the statement of the results, without even considering what it takes to prove the results.

(2) The theory assumes d-regular graphs with a very specific clustering structure that is not going to be satisfied by pretty much any real world graph. Expanding this to d-bounded graphs doesn't make it much more practical. Even if a graph does satisfy the well-clusterability assumptions for certain choices of $k, \varepsilon, \varphi$, we wouldn't know these a priori, and the algorithm make strong assumptions about these parameter (just one among many examples: the specific need in Theorem 3.1 for $\varepsilon /\varphi \leq 1/10^5$).

This makes it all the more impressive that the paper includes an experimental result of any kind, but in order for this to work (even in a very carefully controlled synthetic setting), one needs to try out many different parameters for the algorithms.

I'm still overall in favor of seeing this paper accepted, given the significance of its core technical contribution and its many other strengths. Graph clustering in general seems well within scope at the conference, and the fact that there are at least some numerical experiments is a plus.

**Questions:**

Is there hope for something like this to be practical for real world graph clustering? What would it take to make that happen?

How long did it take to construct $\mathcal{D}$ in practice for your numerical experiments?

---

> ### Author Response · Authors · 2025-11-20
> **Official Comment by Authors (1/2)**
>
> We thank the reviewer for the supportive feedback and positive evaluation of our work. We address your concerns in detail below.
>
> **W1: I do wonder though whether ICLR is the right venue for this work, and instead of a core theory conference. … Graph clustering in general seems well within scope at the conference, and the fact that there are at least some numerical experiments is a plus.**
>
> Thank you for raising this point. we believe there are several reasons why ICLR is indeed appropriate:
> * **Graph clustering lies well within ICLR’s scope.** As you noted, graph clustering is broadly relevant to representation learning and machine learning, and many clustering-oriented works have appeared at ICLR.
> * **Our time-space trade-off results address resource‑efficiency questions.** One of our main contributions is a time-space trade-off for sublinear spectral clustering algorithms. Given the communitn by’s strong interest in efficient learning and representation algorithms (especially under constrained computational resources), we believe ICLR is a fitting venue for such a contribution.
> * **Balance of theory and experiment.** Although our work has a strong theoretical component, we also include numerical experiments, which resonate with ICLR’s balance of theory and empirical validation.
> * **There is precedent for theoretical graph clustering work at top ML conferences.** For example,
>     * *Coreset Spectral Clustering* was published at ICLR 2025,
>     * *A Sublinear-Time Spectral Clustering Oracle with Improved Preprocessing Time* was published at NeurIPS 2023 and
>     * *Sublinear-Time Clustering Oracle for Signed Graphs* was published at ICML 2022.
>
> **W2: The presentation of the results is intricate.**
>
> We appreciate this observation.
> * We agree that the statements of the main results inevitably involve many parameters with nontrivial dependencies. This complexity is largely inherent to the problem setting, and we have already attempted to mitigate it. For instance, Theorem 1.1 provides an informal and simplified version of Theorem 3.1 precisely to improve readability.
> * Moreover, we acknowledge that further clarification would benefit readers. In the next version, we plan to include an additional illustrative special case (e.g., a $2$-cluster graph) that captures the essence of our guarantees without the full technical overhead. We believe this will make the results easier to understand while preserving the full formal statements.
>
> **W3: The theory assumes $d$-regular graphs with a very specific clustering structure that is not going to be satisfied by pretty much any real world graph. Expanding this to $d$-bounded graphs doesn't make it much more practical. Even if a graph does satisfy the well-clusterability assumptions for certain choices of $k,\varepsilon,\varphi$, we wouldn't know assumptions about these parameter (just one among many examples: the specific need in Theorem 3.1 for $\varepsilon/\varphi^2\le 1/10^5$).**
>
> These assumptions reflect the inherent structure of the sublinear clustering oracle problem and are standard in the literature; previous works rely on similar conditions. Moreover, they still capture meaningful real-world settings—for example, certain European social networks satisfy these properties (each individual naturally has a limited number of connections, making the network effectively $d$-bounded; moreover, the population tends to form geographically or culturally cohesive communities (e.g., users from the same country), which yields a well-clusterable structure). For general graphs, one can instead use a standard lazy random walk, and our algorithms extend naturally to that setting (see below).

---

> ### Author Response · Authors · 2025-11-20
> **Official Comment by Authors (2/2)**
>
> **Q1: Is there hope for something like this to be practical for real world graph clustering? What would it take to make that happen?**
>
> Thank you for raising this question. We agree that many real world graphs may not strictly satisfy the assumptions required by our algorithm (e.g., $d$-bounded and $\varepsilon/\varphi^2<1/10^5$). Nevertheless, we believe that there is meaningful potential for making such techniques applicable in practice.
> * **For bounded degree assumption:** many real world graphs can be transformed into effectively bounded-degree graphs through standard and widely used preprocessing techniques, including:
>     * bounded-degree reduction tricks:
>         * replacing a high-degree node with a sampled subset of its incident edges while preserving its essential spectral or clustering properties;
>         * removing a very small fraction of extremely high-degree nodes.
>     * direct operation on general graphs:
>         * using a **lazy** random walk, whose transiom matrix is $M=\frac{1}{2}(I+AD^{-1})$, which reduces the bias introduced by large-degree vertices and stabilizes the walk on graphs with highly skewed degree distributions;
>         * analyzing the **symmetrized transition matrix**: $\bar{M}=D^{-1/2}MD^{1/2}=\frac{1}{2}(I+D^{-1/2}AD^{-1/2})$, which is a similarity transform of the lazy walk matrix and naturally controls the influence of high-degree vertices. This symmetrized operator is standard in prior work on graph property testing for general graphs [1].
>     * In fact, these tools are standard in sublinear-time graph algorithms and property testing, where the bounded-degree assumption mainly serves to simplify the analysis.
> * **For conductance assumption ($\varepsilon/\varphi^2<1/10^5$):** The condition $\varepsilon/\varphi^2$ in our analysis is a sufficient (and conservative) requirement used to simplify certain concentration and approximation arguments. It is not indeed to be a sharp necessary condition for the algorithm to perform well in practice. Empirically, our algorithms **perform well far beyond the conservative regimes** required by the worst-case proof.
>     * In our experiments, we evaluated a graph generated by an SBM with $k=3,p=0.07,q=0.002$ and $1000$ vertices per cluster. Under this parameter setting, the expected outer conductance and inner conductance are approximately $\varepsilon\approx 0.0541$ and $\varphi\approx 0.4734$, leadign to $\varepsilon/\varphi^2\approx 0.2414$, which is significantly larger than $1/10^5$. Nevertheless, our algorithms still perform effectively on these instances.
>
> Overall, while our theoretical guarantees require certain structural assumptions, we believe the underlying techniques have real potential for practical graph clustering, especially with standard preprocessing or more robust random-walk constructions.
>
> **Q2: How long did it take to construct $\mathcal{D}$ in practice for your numerical experiments?**
>
> * In our experiments, constructing $\mathcal{D}$ required between $6.37$ and $27.65$ seconds, depending on the space parameter in the space-time trade-off setting. We did not optimize our implementation for speed; our primary goal was to validate the theoretical guarantees—namely, the space efficiency and the space-time trade-off—rather than to benchmark runtime.
> * In particular, the random-walk simulation was implemented in plain Python with simple for-loops, which are significantly slower than compiled languages due to interpretation overhead and dynamic typing. No low-level optimizations were applied. As a result, the measured running time does not reflect the algorithm’s full performance potential. A more optimized implementation (e.g., in C++ or a similar compiled language) could substantially reduce the construction time.
>
> **Reference**:
>
> [1] Testing graph clusterability: Algorithms and lower bounds. FOCS 2018.
>
> **We thank you again for your valuable feedback. We hope the clarifications above address your concerns. If there are any remaining concerns, we would be glad to provide additional details.**

---

> > ### Comment · Reviewer_S7oN · 2025-11-25
> > **Thanks**
> >
> > Thanks for your responses. I am satisfied by the clarification and in any case these were not major concerns to me and did not affect my overall positive view of the paper. Your point about some theoretical conditions being sufficient conditions for theory but not strictly necessary for good performance in practice is, I think, an especially meaningful point. There are certainly plenty of practical algorithms where this is the case. Overall I maintain a positive view of the paper.

---

> > > ### Author Response · Authors · 2025-11-27
> > >
> > > We sincerely thank you for your positive and encouraging feedback, and we are glad that our clarifications addressed your concerns. We remain happy to clarify anything further if needed.

---

### Official Review · Reviewer_64dM · 2025-10-30

**Soundness:** 3
**Presentation:** 3
**Contribution:** 3
**Rating:** 6
**Confidence:** 3

**Summary:**

The paper studied the construction of spectral clustering oracles on well-clustered graphs with limited memory. The problem has recently attracted a flurry of work due to its applications in sublinear clustering algorithms. Here, we are given a graph that could be partitioned into $k$ clusters where the conductance between the clusters are high and the conductance inside the clusters are low. As such, we could label the vertices to generate a ‘ground truth’ clustering. The goal for the algorithm is to compute a data structure such that upon querying a vertex $x$, the algorithm can answer the cluster label of $x$ with high efficiency. The metrics for good algorithms in this application include:
- Pre-processing time: the time to construct the data structure
- Querying time: the time complexity needed to return the answer for each cluster
- Accuracy: The answer for most of the vertex queries should be correct
- Memory efficiency: the memory used by the data structure should be small

The last aspect was the main contribution of this paper. The paper discussed that all previous algorithms require $\Omega(\sqrt{n})$ space for such applications; in contrast, this algorithm is able to design an algorithm with only $n^{O(\varepsilon/\phi^2)}$ space, where $\varepsilon$ and $\phi$ are parameters that characterize the clusterability of the graph. The query time will be affected, which is now $n^{1+O(\varepsilon/\phi^2)}$ time. In fact, the trade-off could be made general with $n^{O(\varepsilon/\phi^2)}M$ space and $n^{1+O(\varepsilon/\phi^2)}/M$ time.

**Main techniques.** The main techniques of the paper follow from the construction in Shen and Peng [NeurIPS’23]. In a nutshell, this line of techniques reduces the algorithm for the spectral clustering oracle to the approximation of the dot products of vertex embeddings. The previous space lower bound is due to the computation of the approximation dot product using random walks, and this paper adopted the simple idea to conduct the walk in batches to trade time efficiency for space efficiency.

**Strengths:**

I’m generally supportive of the paper. The spectral clustering oracle problem has attracted quite some attention over the past few years, and it is great that the space aspect is taken into consideration in this paper. I did not get the time to verify the correctness of the results, but the technique overview provided some good justifications for first-time readers to believe the correctness. Some experiments are also provided besides the theoretical results.

**Weaknesses:**

On the flip side, I think the paper could do a better job in terms of the comparison between their results and existing algorithms.

Judging from the presentation of the main results, it is not entirely clear whether certain restrictions (e.g., $d$-bounded graph) are also used in previous results, and how the misclustering error would compare with oracles with no space limits.

Furthermore, it is not always clear which part of the algorithm follows from existing work, and which part is the contribution of this paper. The techniques are not thoroughly compared with similar papers. Since I do not know the techniques in those papers, it is harder for me to evaluate the technical novelty.

Overall, I think it’s a solid paper with a good set of results. However, the writing issues and the fact that I’m not very familiar with related techniques make it hard for me to champion for it.

**Questions:**

Most of the questions are embedded in the weakness comments. Some additional questions and comments:

Line 150: The notion of ‘random walk queries’ is not defined. Also, what does Theorem 1.3 mean? The lower bound works only against algorithms that rely exclusively on random walk queries, and cannot make, e.g., degree or neighborhood queries, on adjacency lists? This family of algorithms seems to be extremely restrictive.

Line 262-263: We use $O_\phi$ *to suppress*; similarly, $\tilde{O}$ *to hide*.

The leading constants in $n^O(\varepsilon)$ appear to be crazily large for any interesting applications. In your experiment, did you conduct some type of algorithm engineering to bring down the actual time and space?

---

> ### Author Response · Authors · 2025-11-20
> **Official Comment by Authors (1/2)**
>
> We thank the reviewer for the supportive feedback and positive evaluation of our work. We fixed the typos in the updated manuscript. We address your concerns in the following.
>
> **W1: I think the paper could do a better job in terms of the comparison between their results and existing algorithms.**
>
> Thank you for this suggestion. We agree that the comparison with existing algorithms could be clearer, and the current presentation is compressed due to space constraints. We have **added a new table (Table $1$)** in our updated manuscript which is highlighted in blue to make the distinctions and improvements more explicit.
>
> For the reviewer's convenience, we also reproduce the content of Table $1$ below, focusing on space usage, query time, misclassification error (# of misclassified vertices), and the conductance gap required by the clustering oracle:
>
> | work | space usage | query time | misclassification error | conductance gap |
> |----------|----------|----------|----------|----------|
> | [1]   | $\widetilde{O}_\varphi(\sqrt{n}\cdot \textup{poly}(\frac{k}{\varepsilon}))$   | $\widetilde{O}_\varphi(\sqrt{n}\cdot \textup{poly}(\frac{k}{\varepsilon}))$   | $O(kn\sqrt{\varepsilon})$   | $\textup{poly}(k)\log n$   |
> | [2]  |  $\widetilde{O}_\varphi(n^{1-\delta+O(\varepsilon)}\cdot \textup{poly}(\frac{k}{\varepsilon}))$  | $\widetilde{O}_\varphi(n^{\delta+O(\varepsilon)}\cdot \textup{poly}(\frac{k}{\varepsilon}))$   | $O(\log k\cdot \varepsilon)\|C_i\|$ for each cluster $C_i$   | $\log k$   |
> | **this work (1)**   |  $\widetilde{O}_\varphi(n^{O(\varepsilon)}\cdot M\cdot \textup{poly}(\frac{k}{\varepsilon}))$  | $\widetilde{O}_\varphi(n^{1+O(\varepsilon)}\cdot\frac{1}{M}\cdot \textup{poly}(\frac{k}{\varepsilon}))$   | $O(\log k\cdot \varepsilon)\|C_i\|$ for each cluster $C_i$   | $\log k$   |
> | [3] | $\widetilde{O}_\varphi(n^{1-\delta+O(\varepsilon)}\cdot \textup{poly}(k))$   | $\widetilde{O}_\varphi(n^{\delta+O(\varepsilon)}\cdot \textup{poly}(k))$   | $O(\textup{poly} (k)\cdot \varepsilon^{1/3})\|C_i\|$ for each cluster $C_i$   | $\textup{poly}(k)$   |
> | **this work (2)**  | $\widetilde{O}_\varphi(n^{O(\varepsilon)}\cdot M\cdot \textup{poly}(k))$   | $\widetilde{O}_\varphi(n^{1+O(\varepsilon)}\cdot \frac{1}{M}\cdot \textup{poly}(k))$   | $O(\textup{poly} (k)\cdot \varepsilon^{1/3})\|C_i\|$ for each cluster $C_i$   | $\textup{poly}(k)$   |
>
> where $\boldsymbol{\delta\in(0,1/2]}$ is a constant and $n^{\Theta(\varepsilon)}\le M\le O(\frac{n^{1/2-O(\varepsilon)}}{k})$ is a trade-off parameter.
>
> As shown above,
> * our work is the first to break the $\Omega(\sqrt{n})$ space barrier for sublinear spectral clustering oracles, achieving **space usage well below $\sqrt{n}$**. In addition, our approach offers a flexible time-space trade-off.
> * Our space-limited clustering oracle achieves the **same misclassification error** and relies on the **same conductance gap** as oracles without space constraints; in other words, our oracle does not worsen performance in these two aspects.
>
> **W2: Judging from the presentation of the main results, it is not entirely clear whether certain restrictions (e.g., $d$-bounded graph) are also used in previous results, and how the misclustering error would compare with oracles with no space limits.**
>
> * We would like to stress that the $d$-boundedness assumption is primarily a theoretical tool that allows us to analyze the behavior of random-walk-based algorithms in a clean and controlled setting. Similar bounded-degree assumptions are **commonly adopted** in the literature on graph clustering algorithms [2,3,4] and property testing [5,6,7].
> * As shown in the table above, our space-limited clustering oracle attains the **same misclassification error** as clustering oracles with no space limits.
>     * In addition, we have added an explicit clarification in the revised manuscript (highlighted in blue) to emphasize that the misclassification error remains unchanged.
>
> **Referece:**
>
> [1] Robust clustering oracle and local reconstructor of cluster structure of graphs. SODA 2020.
>
> [2] Spectral clustering oracles in sublinear time. SODA 2021.
>
> [3] A sublinear-time spectral clustering oracle with improved preprocessing time. NeurIPS 2023.
>
> [4] Learning hierarchical cluster structure of graphs in sublinear time. SODA 2023.
>
> [5] Testing expansion in bounded-degree graphs. FOCS 2007.
>
> [6] Testing cluster structure of graphs. STOC 2015.
>
> [7] Testing graph clusterability: Algorithms and lower bounds. FOCS 2018.
>
> **Please see below for W3 and Q1~Q3.**

---

> ### Author Response · Authors · 2025-11-20
> **Official Comment by Authors (2/2)**
>
> **W3: It is not always clear which part of the algorithm follows from existing work, and which part is the contribution of this paper. The techniques are not thoroughly compared with similar papers.**
>
> We thank the reviewer for pointing this out. To clarify our contributions relative to prior techniques, we have made the following novel aspects more clearly stated in the updated manuscript, and the corresponding changes are highlighted in blue:
> * **Upper bound:** The main novelty lies in adapting ideas previously used to analyze the sample-space trade-off in distribution testing under the streaming model to the study of random-walk behavior on graphs. While the underlying technique is inspired by prior work, we are the first to apply this idea in the graph setting to rigorously analyze random walks.
> * **Lower bound:** Our main novelty is a new reduction that connects random-walk-based graph clustering to space-bounded distribution testing. We construct paired hard instances and show how any random-walk algorithm for distinguishing $1$-cluster vs. $2$-cluster instances can be simulated in the distribution-testing setting. The key technical contribution is an inductive coupling argument ensuring that the random-walk histories remain indistinguishable in total-variation distance. This reduction is new and is what enables our space-time lower bound.
>
> **Q1: The notion of 'random walk queries' is not defined.**
>
> Thank you for pointing this out.
> * Due to space limitations, the formal definition of the random walk query model is previously provided only in Appendix I, and the main text gives only a brief description (line 147): "for each queried vertex $x$, the oracle returns the endpoint of a random walk of length $O(\log n)$ starting from $x$".
> * Now, we have moved the formal definition into the main body for clarity (line 172).
>
> **Q2: What does Theorem 1.3 mean? The lower bound works only against algorithms that rely exclusively on random walk queries, and cannot make, e.g., degree or neighborhood queries, on adjacency lists?**
> * Theorem 1.3 establishes a space-time trade-off lower bound for algorithms that solve the $1$-cluster vs. $2$-cluster problem using only random-walk queries. Specifically, it states that $S\cdot T\ge \Omega(n)$, where $S$ and $T$ denote space and time complexities.
> * Our lower bound does not extend to models that allow adjacency list, neighborhood, or degree queries. Here is the **bottleneck**:
>     * The main difficulty lies in the fact that our current lower bound critically relies on the ability to interpret each random-walk query as providing one independent sample from a well-defined probability distribution (namely, the endpoint distribution of a $O(\log n)$-length walk). This allows us to reduce the clustering problem to a distribution-testing problem and via the information-theoretic framework to get a space-time lower bound. In contrast, neighborhood queries or adjacency list queries do not provide such samples: each query only reveals a local piece of the graph (e.g., a neighbor of a vertex), which does not correspond to a draw from the random-walk endpoint distribution. Because of this mismatch, the core reduction used in our proof does not carry over.
>     * Following the seminal work of Raz [1], almost all previous results in this area focused on the **sample**-space trade-off, where each sample corresponds to an independent draw from a distribution over some domain, rather than **query**-space trade-off. Here, we use random-walk query to simulate the "sample" setting.
>     * Establishing a comparable lower bound in the adjacency list query model would likely require constructing fundamentally new hard instances or techniques. We view this as an interesting direction for future work.
>
> **Q3: The leading constants in $n^{O(\varepsilon)}$ appear to be crazily large for any interesting applications. In your experiment, did you conduct some type of algorithm engineering to bring down the actual time and space?**
>
> Thank you for this question.
> * In our experiments, we did not apply any algorithmic or low-level engineering optimizations; the implementation is a straightforward, "naive" version of the algorithm written in plain Python.
> * The focus of our experiments was to validate the theoretical result of the algorithm, in particular the space efficiency and the time-space trade-off.
> * For larger-scale or performance-critical applications, implementing the algorithm in a compiled language (e.g., C++) or applying standard engineering optimizations may substantially improve efficiency.
>
> **Reference:**
>
> [1] A time-space lower bound for a large class of learning problems. FOCS 2017.
>
> **We thank you again for your valuable feedback. We hope the clarifications above address your concerns. If there are any remaining concerns, we would be glad to provide additional details.**

---

> ### Author Response · Authors · 2025-11-27
> **We would love to hear back from Reviewer 64dM**
>
> Dear Reviewer 64dM,
>
> Thank you very much for your careful reading and constructive comments. We have provided point-by-point responses to all of your concerns and have updated the manuscript accordingly, with all changes highlighted in blue for easy reference.
>
> We would be grateful if you could confirm whether our responses have addressed your concerns. If there is any further clarification or additional information that you need, please feel free to let us know. We would be glad to provide any further details.
>
> Best regards,
> Submission 16919 Authors

---

### Official Review · Reviewer_hd45 · 2025-11-03

**Soundness:** 2
**Presentation:** 2
**Contribution:** 2
**Rating:** 4
**Confidence:** 4

**Summary:**

This paper considered the problem of designing sublinear spectral clustering oracles for well-clustered graphs. The authors assume query access to the adjacency list of the graph. They have given a space-time tradeoff for this problem, and also showed this tradeoff is tight for approaches using only random walk oracles. One of the interesting feature of this work is that their algorithm has space complexity of $o(\sqrt{n})$, in contrast to previous algorithms which require $\Omega(\sqrt{n})$ space.

**Strengths:**

Strengths:

1. They have given a space-time tradeoff for designing sublinear spectral clustering oracles.

2. They have shown that this tradeoff is tight for approaches that use only random walk oracles

**Weaknesses:**

1. (Upper bound) The main technical contribution of this work is the construction of an efficient inner-product oracle (Theorem 3.2) for spectrally embedded vertex vectors. However, the details of this construction are not presented in the main text. The authors only sketch those proofs that primarily follow the approach of Shen & Peng (2023).

2. (Lower bound) While the proof of the space–time tradeoff lower bound is nice, it would have been much more interesting if the lower bound had been established under adjacency list query access, which is the more natural and standard setting.

3. I suggest moving the experimental section to the appendix and instead including a more detailed proof outline for Theorem 3.2 in the main text.

4. Although the tradeoff result is nice, it is unclear whether the resolved question was explicitly open in the prior literature. Clarification on this point would be helpful.

5. The authors did not adequately highlight the novel ideas distinguishing their techniques from previous works.

**Questions:**

1. The authors should highlight the new ideas in their technique compared to previous works.

2. What is the main bottleneck to extending the lower bound to adjacency oracle query access?

3. Was the tradeoff an explicit open problem in the prior literature?

---

> ### Author Response · Authors · 2025-11-20
> **Official Comment by Authors (1/2)**
>
> Thank you very much for taking the time to review our paper. We appreciate the constructive suggestions. We slightly summarized your concerns and provided detailed answers below.
>
> **W1: The main technical contribution of this work is the construction of an efficient inner-product oracle (Theorem 3.2) for spectrally embedded vertex vectors. However, the details of this construction are not presented in the main text. The authors only sketch those proofs that primarily follow the approach of Shen & Peng (2023).**
>
> * We would like to clarify that the construction details of the inner-product oracle (Theorem 3.2) are actually presented in the main text, specifically in Algorithms $1\sim 4$. Due to space constraints, we moved the full proof of Theorem 3.2 to the appendix, but the complete description of the oracle itself is contained in the main body.
> * The proof sketch that primarily follows the approach of Shen & Peng (2023) refers to Item 2 of Theorem 3.1 rather than Theorem 3.2. We apologize for the confusion caused by this arrangement, and we will revise the presentation to make this distinction clearer in the final version.
>
> **W2 & Q2: While the proof of the space-time tradeoff lower bound is nice, it would have been much more interesting if the lower bound had been established under adjacency list query access, which is the more natural and standard setting. What is the main bottleneck to extending the lower bound to adjacency oracle query access?**
>
> Thank you for this insightful comment.
> * Our space-time trade-off lower bound is indeed established under the random walk query model. We focus on this model as random-walk-based methods constitute one of the most natural and widely used approaches for sublinear graph clustering; to the best of our knowledge, all the previous sublinear spectral  clustering oracles are based on the random-walk approaches [1,2,3]. For this class of algorithms, our result gives an almost-tight lower bound (up to $\textup{poly}(\log n)$ factors). Nonetheless, we fully agree that obtaining a comparable lower bound under adjacency list query access would be even more interesting, and we view this as an interesting direction for future work.
> * **Bottleneck:**
>     * The main difficulty lies in the fact that our current lower bound critically relies on the ability to interpret each random-walk query as providing one independent sample from a well-defined probability distribution (namely, the endpoint distribution of a $O(\log n)$-length walk). This allows us to reduce the clustering problem to a distribution-testing problem and via the information-theoretic framework to get a space-time lower bound. In contrast, adjacency list queries do not provide such samples: each query only reveals a local piece of the graph (e.g., a neighbor of a vertex), which does not correspond to a draw from the random-walk endpoint distribution. Because of this mismatch, the core reduction used in our proof does not carry over.
>     * Following the seminal work of Raz [4], almost all previous results in this area focused on the **sample**-space trade-off, where each sample corresponds to an independent draw from a distribution over some domain, rather than **query**-space trade-off. Here, we use random-walk query to simulate the "sample" setting.
>     * Establishing a comparable lower bound in the adjacency list query model would likely require constructing fundamentally new hard instances or techniques.
>
> **W3: I suggest moving the experimental section to the appendix and instead including a more detailed proof outline for Theorem 3.2 in the main text.**
>
> Thank you for this helpful suggestion. We agree that providing a more detailed proof outline for Theorem 3.2 would improve the clarity of the presentation. In the final version of the paper, we will reorganize the exposition and move additional proof details of Theorem 3.2 into the main text, while adjusting the placement of the experimental section accordingly.
>
> **Reference:**
>
> [1] Testing cluster structure of graphs. STOC 2015.
>
> [2] Spectral clustering oracles in sublinear time. SODA 2021.
>
> [3] A sublinear-time spectral clustering oracle with improved preprocessing time. NeurIPS 2023.
>
> [4] A time-space lower bound for a large class of learning problems. FOCS 2017.
>
> **Please see below for W4 & Q3 and W5 & Q1.**

---

> ### Author Response · Authors · 2025-11-20
> **Official Comment by Authors (2/2)**
>
> **W4 & Q3: Although the tradeoff result is nice, it is unclear whether the resolved question was explicitly open in the prior literature. Clarification on this point would be helpful. Was the tradeoff an explicit open problem in the prior literature?**
>
> * To the best of our knowledge, the space-time trade-off for sublinear graph problems has not been explicitly studied before; our work is the *first* to consider this question in the context of sublinear graph algorithms. Thus, instead of resolving a previous open question, we are the first to ask this question and provide a non-trivial answer through matching (up to $\textup{poly}(\log n)$ factors) upper and lower bounds. Prior work on space limitations mainly focused on streaming models, which is quite different from our setting.
> * As noted in the strengths section by another reviewer (64dM), **The spectral clustering oracle problem has attracted quite some attention over the past few years, and it is great that the space aspect is taken into consideration in this paper**, which further underscores the importance of our contribution in exploring the space-time trade-off.
> * We have made this novelty and clarification more explicit in the revised manuscript, and the corresponding changes are highlighted in blue.
>
> **W5 & Q1: The authors did not adequately highlight the novel ideas distinguishing their techniques from previous works. The authors should highlight the new ideas in their technique compared to previous works.**
>
> Thank you for this comment. We appreciate the reviewer's suggestion to more clearly emphasize the novel ideas behind our techniques. In the revised manuscript, we have highlighted the novel ideas in blue, focusing on two main aspects:
>
> * **Upper bound:** The main novelty lies in adapting ideas previously used to analyze the sample-space trade-off in distribution testing under the streaming model to the study of random-walk behavior on graphs. While the underlying technique is inspired by prior work, we are the first to apply this idea in the graph setting to rigorously analyze random walks.
> * **Lower bound:** Our main novelty is a new reduction that connects random-walk-based graph clustering to space-bounded distribution testing. We construct paired hard instances and show how any random-walk algorithm for distinguishing $1$-cluster vs. $2$-cluster instances can be simulated in the distribution-testing setting. The key technical contribution is an inductive coupling argument ensuring that the random-walk histories remain indistinguishable in total-variation distance. This reduction is new and is what enables our space-time lower bound.
>
> **We thank you again for your valuable feedback. We hope the clarifications above address your concerns. If there are any remaining concerns, we would be glad to provide additional details.**

---

> ### Author Response · Authors · 2025-11-27
> **We would love to hear back from Reviewer hd45**
>
> Dear Reviewer hd45,
>
> Thank you very much for your careful reading and constructive comments. We have provided point-by-point responses to all of your concerns. Due to page limits and considerations of other reviewers' feedback, we have updated the manuscript where possible, with the corresponding changes highlighted in blue for easy reference.
>
> We would be grateful if you could confirm whether our responses have addressed your concerns. If there is any further clarification or additional information that you need, please feel free to let us know. We would be glad to provide any further details.
>
> Best regards, Submission 16919 Authors

---

### Author Response · Authors · 2025-12-02
**Summary to Area Chair**

We sincerely thank the Area Chair for taking the time to handle our submission during this exceptional review process. We provide the following brief summary to assist your evaluation.

**Our Contributions**

Our work addresses the problem of designing sublinear spectral clustering oracles for well-clusterable graphs, and makes the following contributions:
1. **Breaking the Memory Barrier:** We design sublinear clustering oracles that require significantly less memory than existing approaches, thus breaking the $\Omega{(\sqrt{n})}$ barrier.
2. **Memory-Time Trade-off:** We establish the trade-off between memory usage $S$ and query time $T$, showing that $S\cdot T\approx \widetilde{O}(n)$.
3. **Tightness via Lower Bound:** We prove that this trade-off is nearly tight for approaches based on random-walk oracles by providing a matching lower bound.
4. **Experimental Validation:** We validate the performance of our oracles through experiments on synthetic networks.

**Overall Reviewer Feedback**

* Our submission received three reviews, with scores of 8 (``S7oN``), 6 (``64dM``) and 4 (``hd45``).
* Two reviewers explicitly expressed supportive overall assessments:
    * ``S7oN`` (score 8): "I'm still overall in favor of seeing this paper accepted."
    * ``64dM`` (score 6): "I’m generally supportive of the paper."
* During the rebuttal phase, we received a response from only one reviewer—``S7oN`` (score 8), who further reaffirmed the positive assessment, stating:
    * "I am satisfied by the clarification and in any case these were not major concerns to me and did not affect my overall positive view of the paper. ... Overall I maintain a **positive** view of the paper."

**Strengths Highlighted by the Reviewers**

* The paper is well-motivated and tackles an interesting problem in graph analysis. (``S7oN, 64dM``)
* The paper provides strong and meaningful theoretical contributions to sublinear spectral clustering oracles: (``S7oN, 64dM, hd45``)
    * breaks $\Omega(\sqrt{n})$ space barrier, (``S7oN, 64dM``)
    * provides a tight space-time trade-off. (``S7oN, 64dM, hd45``)
* This paper provides experimental results to validate theoretical results. (``S7oN, 64dM``)
* This paper has well-structured presentation and high-quality writing. (``S7oN, 64dM``)

**Main Concerns and Our Responses**

We have provided point-by-point responses to all weaknesses and questions raised by the reviewers, and we have updated the manuscript accordingly, with all changes *highlighted in blue*. Below, we summarize the main concerns and briefly outline our responses.

1. Reviewer ``hd45`` (score 4)
* About extending the lower bound to adjacency oracle query access.
    * We have demonstarted the **fundamental bottleneck**. Establishing a comparable lower bound in the adjacency list query model is highly challenging and would likely require constructing fundamentally new hard instances or techniques.
* Whether the space-time tradeoff we resolve was previously stated as an explicit open problem in the literature?
    * Our work is the *first* to consider this question in the context of sublinear graph algorithms (see line 70 and 79).
    * Reviewer ``64dM``: "...it is great that the space aspect is taken into consideration in this paper."
    * Reviewer ``S7oN``: "...provides good motivation...and tackles and interesting problem in graph analysis."
* About highlighting the new ideas in our technique.
    * We have summarized the key new ideas in our responses and have updated the manuscript accordingly to make these contributions clearer (see lines 210$\sim$212 and 237$\sim$241).

2. Reviewer ``64dM`` (score 6)
* About doing a better job in terms of the comparison between our results and existing algorithms.
    * We have improved the comparison between our results and existing algorithms in the response. Additionally, we added a new table (see Table 1) in the manuscript, which provides a clearer summary and a more comprehensive comparison (see lines 152$\sim$154).
* About clarify our contributions relative to prior techniques.
    * Similar to the concern raised by reviewer ``hd45``.
* About the bottleneck to extend the lower bound to adjacency oracle query access.
    * Similar to the concern raised by reviewer ``hd45``.

3. Reviewer ``S7oN`` (score 8)
* About the practicality of our approach on real-world graphs, given that many theoretical assumptions may not hold, and asks what would be required to achieve this.
    * We clarified that some assumptions are standard tools used to simplify the analysis, while others are sufficient (and conservative) conditions rather than strictly necessary.
    * We also provides techniques to address these limitations.
    * The reviewer responded: **"Your point about some theoretical conditions being sufficient conditions for theory but not strictly necessary for good performance in practice is, I think, an especially meaningful point."**

---

### Meta-Review · Area_Chair_VQPN · 2025-12-28

**Summary:**

The paper is the first to study the memory-time tradeoff, and this is conceptually new. The reviewers are overall positive about this paper. However, reviewers also raised concerns. I feel that the issues seem to be minor compared with the merit of the paper, and that the issues are to some extent addressed in the rebuttal. Overall, I suggest an acceptance.

**Reviewer Concerns:**

I think that the main concerns of the reviewers are as follows.

1. The algorithm/analysis is quite complicated, and the paper did not do a good introduction in the main text.
2. The novelty compared with previous works, especially the techniques, is not very clear form the paper.
3. The lower bound seems to be weak.
4. The experiments can be improved.

All of these have been discussed in the rebuttal. I think 1 and 2 are adequately addressed, or at least the authors suggested promising ways to edit the paper. Item 3 is a more fundamental weakness, and the authors did not resolve it, and left as open question. The authors also discussed the difficulty to improve the lower bound. Item 4 is not properly addressed, because no new experiments are added, or even promised. Nonetheless, some more explanations about their experiment details are given, which seems to be useful.

**Reviewer Scores:**

Reviewer hd45 may slightly raise the score from 4 to maybe 5, because the authors partially addressed the issues. Reviewer 64dM may keep the positive score 6 -- although the authors try to respond to the major concerns, the concerns seem to be relatively minor, and that the respond does not provide fundamentally new messages. Reviewer S7oN will keep the score 8, as the reviewer indicated in a follow-up comment.

---

### Decision · Program_Chairs · 2026-01-26

Accept (Poster)